# Schwarz–Schur Involution: Lightspeed Differentiable Sparse Linear Solvers

**Yu Wang** [1 2 3]  **S. Mazdak Abulnaga** [1 2]  **Yaël Balbastre** [1 4]  **Bruce Fischl** [1 2]

## Abstract

Sparse linear solvers or generalized deconvolution are fundamental to science and engineering, applied in partial differential equations (PDEs), scientific computing, computer vision, and beyond. Indirect solvers possess characteristics that make them undesirable as stable differentiable modules; existing direct solvers, though reliable, are too expensive to be adopted in neural architectures. We substantially accelerate *direct* sparse solvers by up to three orders of magnitude, violating common assumptions that direct solvers are too slow. We "condense" a sparse Laplacian matrix into a dense tensor, a compact data structure that batchwise stores the Dirichlet-to-Neumann matrices, reducing the sparse solving to recursively merging pairs of dense matrices that are much smaller. The batched small dense systems are sliced and inverted in parallel to take advantage of dense GPU BLAS kernels, highly optimized in the era of deep learning. Our method is efficient, qualified as a strong zero-shot baseline for AI-based PDE solving, and a reliable differentiable module integrable into machine learning pipelines. Our code is available at https://github.com/wangyu9/Schwarz_Schur_Involution.

## 1. Introduction

In this paper, we propose a conceptually simple approach that reduces solving certain sparse linear system $(\mathbf{A}, \mathbf{b})$ to "involuting" tensors $(\boldsymbol{\alpha}, \boldsymbol{\beta})$, leading to significant improvements. Global linear systems in computer vision and scientific computing, such as the Laplacian or Hessian systems on image domains, are usually very large yet sparse, as induced by pixel affinity. For a sparse $\mathbf{A}$, *direct* solvers find the *exact* solution: $\mathbf{x} = \mathbf{A}^{-1}\mathbf{b}$, for which we present the first

Table 1. Our substantially faster sparse solver (including the factorization and substitution stages) is the first direct method to run at interactive rates. It takes our method 10.9 ms (resp. 220 ms) to solve a Laplacian system (Dirichlet) on an image of $513 \times 513$ (resp. $2561 \times 2561$). Runtime is reported in milliseconds.

| Example | CUDA | SciPy | ours | speedup |
|---------|------|-------|------|---------|
| $2561^2$ | 36926 | 253318 | 220.1 | **168X 1151X** |
| $2049^2$ | 21354 | 143100 | 158.5 | **135X 903X** |
| $1025^2$ | 4710 | 16512 | 36.45 | **129X 453X** |
| $513^2$ | 1036 | 2051 | 10.90 | **95.0X 188X** |
| $257^2$ | 234 | 355 | 5.82 | **40.2X 61.0X** |

interactive rate algorithm on common-size images. Solving the linear system $\mathbf{A}^{-1}\mathbf{b}$ is equivalent to the generalized deconvolution of an image $\mathbf{b}$ with a spatially varying kernel (in rows of) $\mathbf{A}$. Sparse linear solvers underpin modern image processing, computer vision, graphics, as well as numerical methods for PDEs such as finite difference/element methods (FD/FEM) used in electrical/mechanical engineering and computational sciences. Users in these areas can also benefit from our approach if efficiency is a primary concern. Our direct solver—even naïvely prototyped in PyTorch—is 60 to 1000× faster than SciPy and 40 to 170× faster than CUDA (cuDSS), which are highly optimized (Table 1).

Successes of modern deep learning are largely built on *differentiable numerical linear algebras*, particularly matrix multiplications of dense—such as attentions (Vaswani et al., 2017)—and sparse matrices—notably convolutions (LeCun et al., 1989). High performance AI systems rely on efficient implementation of these operations, including dense (Strassen, 1969; Blahut, 2010; Dao et al., 2022; Dao, 2023) and sparse multiplications (Winograd, 1980; Cook, 1966; Lavin & Gray, 2016). In contrast, matrix inversions are almost never employed in neural architectures. We believe that inversions are underexplored and can play a prominent role, since (sparse) matrix inversions encompass a large range of operations such as generalized deconvolution, geometry representation, results of physical simulations, decorrelation, and equilibrium states of localized interactions.

As detailed in §F.2, a key obstacle preventing neural architectures from adopting linear solvers is the lack of a sparse solver like ours that is consistent-performance, lightspeed, in-the-wild, accurate, robust, and zero-parameter. Indirect, a.k.a. iterative, solvers fall short of these goals due to their strong problem dependency, parameter sensitivity, and un-

[1] Athinoula Martinos Center, Massachusetts General Hospital, Harvard Medical School [2] Massachusetts Institute of Technology, MIT CSAIL, USA [3] Sony, USA [4] University College London, UK. Correspondence to: <{ywang155,bfischl}@mgh.harvard.edu>.

*Proceedings of the 42nd International Conference on Machine Learning*, Vancouver, Canada. PMLR 267, 2025. Copyright 2025 by the author(s).

predictable runtime. The large diversity of PDEs results in laborious workflows requiring expert users in the loop to tweak parameters or preconditioners to employ indirect solvers. Iterative solvers struggle to deconvolute indefinite kernels, such as stencils discretizing the Helmholtz PDE (Ernst & Gander, 2011). While existing direct solvers are promising candidates for meeting the desiderata, they are too slow—a limitation that our method overcomes. Catalyzed by the surge of interest in physics-informed machine learning and AI for PDE/Science (Berens et al., 2023), identification or learning of unknown physical systems open up a unique setting to solve **matrix A whose properties are unknown** in advance to apply appropriate iterative schemes, necessitating a general-purpose solver that is deployable **for *any* A** and/or a large varieties of matrices from some training batch. This is similar to the generalized deconvolution with unknown kernels in computer vision. Applying unsuitable indirect solvers can yield catastrophic failure.

Our improvement is due to "condensing"—transfer GPU's capacity of dense BLAS (Dongarra et al., 1988; 2018) to sparse computation, through design choices with a compact data structure—a tensor storing Dirichlet-to-Neumann matrices or sub-systems in batches, and a procedure we coin as "Schwarz-Schur involution"—recursively applying the Schur complement formula to block-wisely contract nodes and keep track of sparsity explicitly. Our approach amounts to a parallel implementation of Gaussian elimination under nested dissections (George, 1973) and the induced multifrontal solvers (Duff & Reid, 1983), recursively canceling variables. We take advantage of the regularity of image grids to aggressively explore parallelisms, leveraging advances in GPUs sparked by deep learning that shift the best practice towards algorithms better exploiting parallelisms.

Our sparse solvers, with speeds exceeding iterative solvers, retain the accuracy and reliability of direct solvers, efficient enough to be integrated within neural architectures. Owing to the central role of linear solvers, our method enables efficient implementations including but not limited to:
• generalized deconvolution of spatially varying kernels.
• identify/reduce gaps between AI & conventional methods.
• zero-shot baselines for learning-based PDE solvers.
• exact Newton solvers made tractable on image domains.
• mathematical optimization layers embedded in neural nets (Amos & Kolter, 2017), (physics) solver-in-the-loop.
• geometric deep learning & algorithms (Litany et al., 2017), shape/deformation representation, graph/mesh/FEM neural networks (Wang et al., 2019; Pfaff et al., 2020).
• eigenbases for spectral neural nets (Bruna et al., 2013), spectral clustering & segmentation (Shi & Malik, 2000).

### 1.1. Related Work

We refer interested readers to standard texts on direct solvers of dense (Davis, 2006) and sparse systems (Duff et al., 2017) for extensive surveys. Direct solvers first perform the numerical factorization—Cholesky or LDLT factorization for symmetric **A** and LU factorization for asymmetric systems—followed by a back substitution to spread out **b** to yield **x**. Major backends for direct sparse solvers include SuperLU (Li, 2005), UMFPACK (Davis, 2004), CUDA (Nickolls et al., 2008), Pardiso (Schenk & Gärtner, 2004), Eigen (Guennebaud et al., 2014), and Cholmod/-SuiteSparse (Chen et al., 2008). When **A** comes from a spatially constant kernel or point spread function (PSF), the problem becomes an image deconvolution, with efficient frequency domain solvers (Sezan & Tekalp, 1990; Hansen et al., 2006); spectral solvers like FFT/IFFT (Fast Fourier Transform) (Frigo & Johnson, 2005) are limited to solving homogeneous Laplace systems. Instead, we attack PDEs or deconvolution with spatially varying coefficients/kernels.

We consider solving indefinite and nonsymmetric square linear systems (Trefethen & Bau, 1997), so methods requiring symmetric or positive-definite **A** do not generally apply, such as CG (Conjugate Gradient) (Hestenes et al., 1952), MINRES (Paige & Saunders, 1975; Liu & Roosta, 2022). Methods apply to our setting include: CGS (Conjugate Gradient Squared) (Sonneveld, 1989) and the improved variant biCGstab (Bi-Conjugate Gradient Stabilized) (Van der Vorst, 1992; Sleijpen & Fokkema, 1993), iterative sparse least-squares LSQR (Paige & Saunders, 1982b;a) that amounts to CG applied to $(\mathbf{A}^\intercal \mathbf{A})^{-1} \mathbf{A}^\intercal \mathbf{b}$ and a recent improvement LSMR (Fong & Saunders, 2011), GMRES (generalized minimal residual) (Saad & Schultz, 1986) and improvements DQGMRES (Saad & Wu, 1996) and FGMRES (Saad, 1993). Other methods include BILQ (Montoison & Orban, 2020; Fletcher, 1976), QMR (Freund & Nachtigal, 1991; 1994), FOM (Saad, 1981), and DIOM (Saad, 1984). The multigrid methods (Bramble, 1993) apply a hierarchical coarse-to-fine strategy, effective as iterative solvers or the preconditioners in CG to yield PCG, especially for elliptic PDEs. We defer discussions on solvers in vision & graphics (Barron & Poole, 2016; Krishnan et al., 2013; Bolz et al., 2003; Jeschke et al., 2009; Horvath & Geiger, 2009; Liu et al., 2016) to §A.

## 2. Mathematical Preliminaries

**Sparse linear solver = exact generalized deconvolution.**
In the paper, for an $H \times W$ image with $n$ pixels, $n := HW$, whose boundary has $b := 2W + 2H - 4$ pixels, we consider a sparse matrix $\mathbf{A} \in \mathbb{R}^{n \times n}$ that encodes the affinity of pixels: each row/column of $\mathbf{A}$ corresponds to one pixel and $\mathbf{A}_{ij}$ is nonzero only if pixels $i, j \in \{1, ..., n\}$ are adjacent in the image, namely their integer coordinates $(x_i, y_i), (x_j, y_j) \in \mathbb{N}^2$ satisfying that $(x_i - x_j)^2 + (y_i - y_j)^2 \leq 2$. For a function $u(x, y)$ on the image grid, consider the linear operator or a $3 \times 3$ convolution centered at $(x, y)$ with $a^{(x,y)} \in \mathbb{R}^{3 \times 3}$:

$$v(x, y) \leftarrow \sum \sum\nolimits_{\delta_x, \delta_y \in \{-1, 0, 1\}} a^{(x,y)}(\delta_x, \delta_y) u(x + \delta_x, y + \delta_y)$$

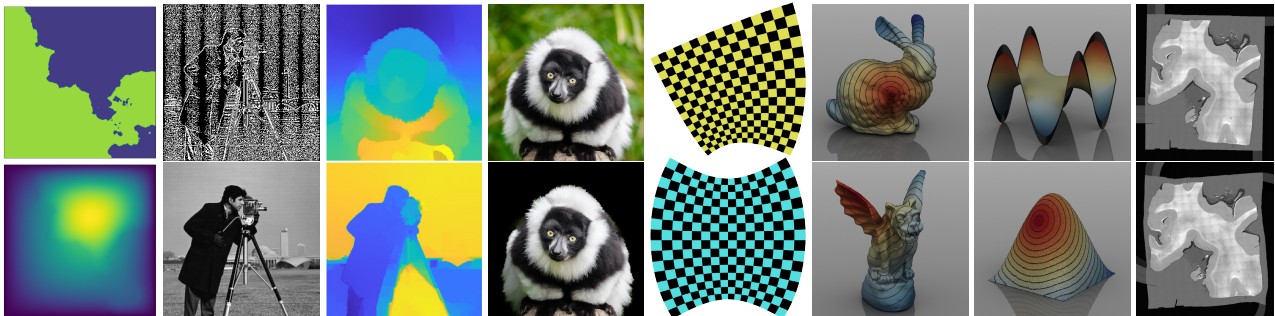

*Figure 1.* Our method immediately accelerates direct solvers in a large variety of tasks by orders of magnitude, including: FEM for solving PDEs, deconvolution (generalized to spatially varying kernels), eigen solver and spectral method, image segmentation and matting, physical simulation, deformation, geometry processing, shape optimization, Newton's method, and diffeomorphic image registration.

using a 9-point stencil $a^{(x,y)}$—a spatially varying kernel:

$$a^{(x,y)} := \begin{bmatrix} a^{(x,y)}(-1,-1) & a^{(x,y)}(-1,0) & a^{(x,y)}(-1,1) \\ a^{(x,y)}(0,-1) & a^{(x,y)}(0,0) & a^{(x,y)}(0,1) \\ a^{(x,y)}(1,-1) & a^{(x,y)}(1,0) & a^{(x,y)}(1,1) \end{bmatrix}. \quad (1)$$

By flattening the 2-dim array $u, v$ into the vector $\mathbf{u}, \mathbf{v} \in \mathbb{R}^{n \times 1}$ (see Figure 25), for $\mathbf{A}_{ij} = a^{(x_j, y_j)}(x_i - x_j, y_i - y_j)$, the convolution with $a^{(x,y)}$ can be written as the matrix-vector product $\mathbf{v} \leftarrow \mathbf{A}\mathbf{u}$. Thus, the linear solve $\mathbf{A}^{-1}\mathbf{v}$ *generalizes deconvolution* to the case with a spatially varying kernel $a^{(x,y)}$. Theoretically, our approach generalizes to larger stencils and 3D, which we leave for future work.

**Differentiable and repetitive linear solvers.** In the learning setting, the system $\mathbf{A}, \mathbf{b}$ come from some learnable parameters $\boldsymbol{\theta}$ to be identified with gradient descent. Thus, in addition to solving $\mathbf{x} = \mathbf{A}^{-1}\mathbf{b}$, we need to obtain $\partial\mathbf{x}/\partial\mathbf{b}$ and $\partial\mathbf{x}/\partial\mathbf{A}$, which have closed-form expressions that can be derived using the adjoint method; evaluating these gradients requires solving the transposed system $\mathbf{A}^\mathsf{T}$, see §C.3.

It is common to sequentially solve multiple pairs $(\mathbf{A}, \mathbf{b}_1)$, $(\mathbf{A}, \mathbf{b}_2)$,..., $(\mathbf{A}, \mathbf{b}_k)$, i.e., the same left-hand side $\mathbf{A}$ but different right-hand sides. Direct solvers, including ours, stand out for allowing **separating numerical factorization from back substitution and reuse** the former. Numerical factorization is the computation involving $\mathbf{A}$ only, which is the *dominant cost that can be reused* for direct solvers. This is not possible for indirect solvers, which have to start over for each right-hand side. The typical value of $k$ is 2 in differentiating linear solvers, $k > 20$ when being applied in eigen solvers, or $k > 100$ in time steppings. Thus, compared to iterative solvers in these settings, computationally, it means that iterative solvers get an extra $\times k$ slowdown.

**PDE and FEM preliminaries.** Our method applies to any invertible $\mathbf{A}$ with the aforementioned sparsity. But first let us examine the $\mathbf{A}$ that arises from discretizing elliptic PDEs (Gilbarg et al., 1977), a situation that motivates our initial development. Surprisingly, the strategy generalizes beyond elliptic PDEs without adaptations to other linear systems in practice, even non positive-semidefinite (PSD). Let us consider solving the Laplace equation defined using

the anisotropic diffusion coefficient $\mathbf{C}(\mathbf{x}) \in \mathbb{R}^{2 \times 2}, \forall \mathbf{x} \in \Omega$ on the domain $\Omega$, subject to either the Dirichlet condition (2) or the Neumann condition (3) at the boundary $\partial\Omega$:

$$-\nabla \cdot [\mathbf{C}(\mathbf{x})\nabla u(\mathbf{x})] = f(\mathbf{x}), \quad u(\mathbf{x})|_{\partial\Omega} = g(\mathbf{x}). \quad (2)$$

$$-\nabla \cdot [\mathbf{C}(\mathbf{x})\nabla u(\mathbf{x})] = f(\mathbf{x}), \quad \mathbf{n}^\mathsf{T}(\mathbf{x})\mathbf{C}(\mathbf{x})\nabla u(\mathbf{x})|_{\partial\Omega} = h(\mathbf{x}). \quad (3)$$

Discretizing with a first-order piecewise linear FEM yields:

$$\mathbf{L} = \mathbf{G}^\mathsf{T}\mathbf{C}\mathbf{G}, \quad [\mathbf{L}\mathbf{u}]|_{b+1:n} = \mathbf{f}, \quad \mathbf{u}|_{1:b} = \mathbf{g} \text{ or } [\mathbf{L}\mathbf{u}]|_{1:b} = \mathbf{h},$$

following the notation and discretization (Wang et al., 2023), in which $\mathbf{u} \in \mathbb{R}^n, \mathbf{f} \in \mathbb{R}^{n-b}, \mathbf{g}, \mathbf{h} \in \mathbb{R}^b$ discretize the solution $u$ and right-hand side functions $f, g, h$; $\mathbf{n}(\mathbf{x})$ is the normal direction at the boundary point $\mathbf{x}$; $\cdot|_{1:b}$ or $\cdot|_{b+1:n}$ selects rows for boundary or interior pixels, resp.; the matrices $\mathbf{G}, \mathbf{C}$ discretize the gradient operator $\nabla$ and $\mathbf{C}(\mathbf{x})$. Under Dirichlet boundary condition $g(\mathbf{x})$, the solution to (2) $u(\mathbf{x})$ is unique, which determines the Neumann boundary condition $h(\mathbf{x})$, inducing the *Dirichlet-to-Neumann* (DtN) map: $g(\partial\Omega) \to h(\partial\Omega)$. After discretization, the DtN map becomes the Schur complement of the matrix $\mathbf{L}$. Thus, the *coefficient-to-solution* operator: $\mathbf{C}(\mathbf{x}) \to u(\mathbf{x})$, under FD/FEM, can be simply viewed as a linear solver.

What exactly $\mathbf{A}$ represents—which helps to motivate—does not make any difference to our method. Our method does not assume $\mathbf{A}$ is symmetric, though this is often the case. Solving Neumann boundary value problems amounts to setting $\mathbf{A} = \mathbf{L}$; solving parabolic PDEs amounts to letting $\mathbf{A} = t\mathbf{L} + \mathbf{M}$, and choosing $\mathbf{A} = \mathbf{L} - \kappa^2 \mathbf{M}$ corresponds to solving the Helmholtz equation that arises in wave propagation and electromagnetism, in which $\mathbf{M}$ plays a role similar to the identity matrix—$\mathbf{M}$ is the mass matrix discretized by FEM (Allaire, 2007) with the same sparsity pattern as $\mathbf{L}$.

## 3. Schur Involution for Parallel Elimination

We demonstrate a procedure to convert the solution of the linear system $\mathbf{x} \leftarrow (\mathbf{A}, \mathbf{b})$ to the "involution" of the tensors $\boldsymbol{\chi} \leftarrow (\boldsymbol{\alpha}, \boldsymbol{\beta})$, by slicing and inverting many small dense matrices in batches to leverage modern GPU hardware.

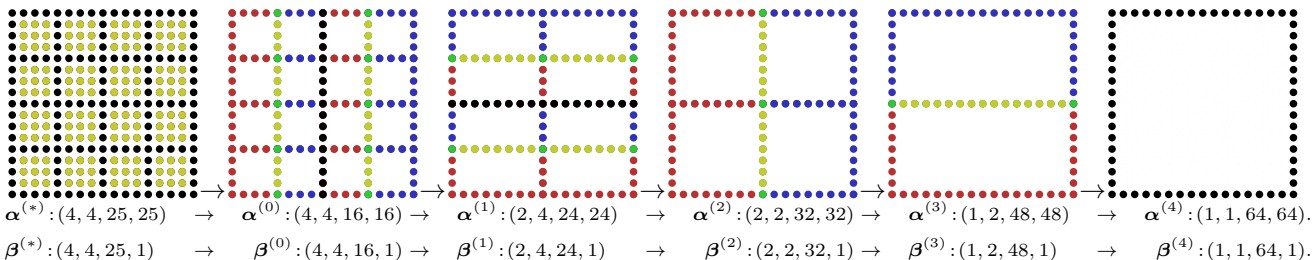

$\boldsymbol{\alpha}^{(*)} : (4, 4, 25, 25) \quad \rightarrow \quad \boldsymbol{\alpha}^{(0)} : (4, 4, 16, 16) \quad \rightarrow \quad \boldsymbol{\alpha}^{(1)} : (2, 4, 24, 24) \quad \rightarrow \quad \boldsymbol{\alpha}^{(2)} : (2, 2, 32, 32) \quad \rightarrow \quad \boldsymbol{\alpha}^{(3)} : (1, 2, 48, 48) \quad \rightarrow \quad \boldsymbol{\alpha}^{(4)} : (1, 1, 64, 64).$

$\boldsymbol{\beta}^{(*)} : (4, 4, 25, 1) \quad \rightarrow \quad \boldsymbol{\beta}^{(0)} : (4, 4, 16, 1) \quad \rightarrow \quad \boldsymbol{\beta}^{(1)} : (2, 4, 24, 1) \quad \rightarrow \quad \boldsymbol{\beta}^{(2)} : (2, 2, 32, 1) \quad \rightarrow \quad \boldsymbol{\beta}^{(3)} : (1, 2, 48, 1) \quad \rightarrow \quad \boldsymbol{\beta}^{(4)} : (1, 1, 64, 1).$

*Figure 2.* Schwarz–Schur involution on a $17{\times}17$ image. Our method (Algorithm 1) applies *batched linear algebras* to *parallelize* Gaussian elimination in a prescribed order—pixels marked in color correspond to nodes to be eliminated in that step. As the initialization, the Schwarz step converts the image into a wire-frame by canceling the interior pixels of each patch. Multiple Schur steps are applied to progressively simplify the wire-frame, until only pixels at the border are left. A Schur step merges every two adjacent subdomains by removing pixels on their shared "edges." We compact all subdomains' left-/right-hand sides in the tensors $\boldsymbol{\alpha}^{(j)}, \boldsymbol{\beta}^{(j)}$ whose shapes evolve as shown. The solutions $\boldsymbol{\chi}^{(j)}$, with the same shape as $\boldsymbol{\beta}^{(j)}$, flow reversely: $\boldsymbol{\chi}^{(*)} \leftarrow \boldsymbol{\chi}^{(0)} \leftarrow \boldsymbol{\chi}^{(1)} ... \leftarrow \boldsymbol{\chi}^{(3)} \leftarrow \boldsymbol{\chi}^{(4)} := (\boldsymbol{\alpha}^{(4)})^{-1} \boldsymbol{\beta}^{(4)}$.

### 3.1. A motivating example: sparse solvers too slow?

A $4096{\times}4096$ image can be divided into a $1024{\times}1024$ array of $4{\times}4$ patches. If adjacent patches overlap by sharing one layer of pixels, then we have a similar division: a $4097{\times}4097$ image (or $17{\times}17$ as shown in Figure 2, or $9{\times}5$ in Figure 3) can be divided into a $1024{\times}1024$ (or $4{\times}4$, or $2{\times}1$, resp.) array of small patches of size $5{\times}5$, since boundary pixels have duplicated representations. On this image, SciPy solver takes more than 20 minutes to solve a sparse system $\mathbf{A} \in \mathbb{R}^{n \times n}$, $n = 16785409$. However, inverting one million $9{\times}9$ matrices in a batch only takes $0.008$ seconds using PyTorch.

```
1  # alpha.shape is (1024,1024,9,9)
2  torch.linalg.inv(alpha) #0.008 sec.
3  # A: sparse matrix
4  scipy.sparse.linalg.spsolve(A,b) #1200 sec.
```

The batched matrix inversion can be viewed as the first step of Gaussian elimination to remove the 9 pixels in the interior $3 \times 3$ block of each patch, and we already make major progress by removing approximately $9/16 \approx 56.25\%$ variables from the problem, while being $10^5 \times$ faster than SciPy to solve the entire problem! This is strong evidence that the speed of sparse linear solvers has been significantly underestimated and parallelism has been under exploited.

### 3.2. Parallel block Gaussian elimination

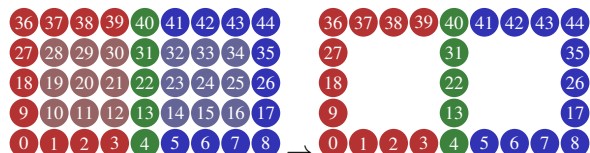

*Figure 3.* Schwarz step removes the interior pixels for each patch.

We introduce a procedure that we term as "Schwarz–Schur involution." For simplicity of illustration, let us first consider a simple case: with only two subdomains, the original problem (4) is first reduced to (5) by the Schwarz step, and then via (9) to (12) by a Schur step. Readers unfamiliar with numerical algebra should refer to §E.

### 3.2.1. SCHWARZ STEP: DECOMPOSE & INITIALIZE DTN

As shown in Figure 3, let $P, Q$ be two subdomains and divide all pixels in the image domain into five (disjoint) groups $\mathbf{r}, \mathbf{s}, \mathbf{t}, \mathbf{a}, \mathbf{b}$ that: $(\mathbf{r}, \mathbf{s})$ is the boundary of $P$ and $(\mathbf{s}, \mathbf{t})$ is the boundary of $Q$—$\mathbf{s}$ is the boundary shared between $P$ and $Q$, and the interior of $P$ and $Q$ are $\mathbf{a}$ and $\mathbf{b}$, resp. [1] The original sparse system $\mathbf{Au} = \mathbf{v}$, $\mathbf{A} \in \mathbb{R}^{n \times n}$ can be written:

$$\begin{bmatrix} \mathbf{A_{rr}} & \mathbf{0} & \mathbf{A_{rs}} & \mathbf{A_{ra}} & \mathbf{0} \\ \mathbf{0} & \mathbf{A_{tt}} & \mathbf{A_{ts}} & \mathbf{0} & \mathbf{A_{tb}} \\ \mathbf{A_{sr}} & \mathbf{A_{st}} & \mathbf{A_{ss}} & \mathbf{A_{sa}} & \mathbf{A_{sb}} \\ \mathbf{A_{ar}} & \mathbf{0} & \mathbf{A_{as}} & \mathbf{A_{aa}} & \mathbf{0} \\ \mathbf{0} & \mathbf{A_{bt}} & \mathbf{A_{bs}} & \mathbf{0} & \mathbf{A_{bb}} \end{bmatrix} \begin{bmatrix} \mathbf{u_r} \\ \mathbf{u_t} \\ \mathbf{u_s} \\ \mathbf{u_a} \\ \mathbf{u_b} \end{bmatrix} = \begin{bmatrix} \mathbf{v_r} \\ \mathbf{v_t} \\ \mathbf{v_s} \\ \mathbf{v_a} \\ \mathbf{v_b} \end{bmatrix} \quad (4)$$

in which there are $\mathbf{0}$ blocks because the interface $\mathbf{s}$ separates the pixels into non-adjacent groups, and $\mathbf{A_{ss}} = \mathbf{A_{ss}^{(P)}} + \mathbf{A_{ss}^{(Q)}}$, $\mathbf{v_s} = \mathbf{v_s^{(P)}} + \mathbf{v_s^{(Q)}}$ can be divided into contributions from $P$ and $Q$, resp. [2] Note that matrix $\mathbf{A}$ and system (4) are only for illustration purposes and never allocated as an actual matrix in our algorithm (see §E). (4) further becomes:

$$\begin{bmatrix} \mathbf{P^{rr}} & \mathbf{0} & \mathbf{P^{rs}} \\ \mathbf{0} & \mathbf{Q^{tt}} & \mathbf{Q^{ts}} \\ \mathbf{P^{sr}} & \mathbf{Q^{st}} & \mathbf{P^{ss} + Q^{ss}} \end{bmatrix} \begin{bmatrix} \mathbf{u_r} \\ \mathbf{u_t} \\ \mathbf{u_s} \end{bmatrix} = \begin{bmatrix} \mathbf{p^r} \\ \mathbf{q^t} \\ \mathbf{p^s + q^s} \end{bmatrix}, \quad (5)$$

by exercising Gaussian elimination to cancel $\mathbf{u_a}, \mathbf{u_b}$, where

$$\begin{bmatrix} \mathbf{P^{rr}} & \mathbf{P^{rs}} \\ \mathbf{P^{sr}} & \mathbf{P^{ss}} \end{bmatrix} := \begin{bmatrix} \mathbf{A_{rr}} - \mathbf{A_{ra}A_{aa}^{-1}A_{ar}} & \mathbf{A_{rs}} - \mathbf{A_{ra}A_{aa}^{-1}A_{as}} \\ \mathbf{A_{sr}} - \mathbf{A_{sa}A_{aa}^{-1}A_{ar}} & \mathbf{A_{ss}^{(P)}} - \mathbf{A_{sa}A_{aa}^{-1}A_{as}} \end{bmatrix}$$

$$\begin{bmatrix} \mathbf{Q^{tt}} & \mathbf{Q^{ts}} \\ \mathbf{Q^{st}} & \mathbf{Q^{ss}} \end{bmatrix} := \begin{bmatrix} \mathbf{A_{tt}} - \mathbf{A_{tb}A_{bb}^{-1}A_{bt}} & \mathbf{A_{ts}} - \mathbf{A_{tb}A_{bb}^{-1}A_{bs}} \\ \mathbf{A_{st}} - \mathbf{A_{sb}A_{bb}^{-1}A_{bt}} & \mathbf{A_{ss}^{(Q)}} - \mathbf{A_{sb}A_{bb}^{-1}A_{bs}} \end{bmatrix}$$

$$\begin{bmatrix} \mathbf{p^r} \\ \mathbf{p^s} \end{bmatrix} := \begin{bmatrix} \mathbf{v_r} - \mathbf{A_{ra}A_{aa}^{-1}v_a} \\ \mathbf{v_s^{(P)}} - \mathbf{A_{sa}A_{aa}^{-1}v_a} \end{bmatrix}, \quad \begin{bmatrix} \mathbf{q^t} \\ \mathbf{q^s} \end{bmatrix} := \begin{bmatrix} \mathbf{v_t} - \mathbf{A_{tb}A_{bb}^{-1}v_b} \\ \mathbf{v_s^{(Q)}} - \mathbf{A_{sb}A_{bb}^{-1}v_b} \end{bmatrix}.$$

---

[1] Concretely speaking, $\mathbf{s} = [4, 13, 22, 31, 40]$,

$\mathbf{r} = [0, 1, 2, 3, 39, 38, 37, 36, 27, 18, 9]$, $\mathbf{a} = [10, 11, 12, 19, 20, 21, 28, 29, 30]$,

$\mathbf{t} = [5, 6, 7, 8, 17, 26, 35, 44, 43, 42, 41]$, $\mathbf{b} = [14, 15, 16, 23, 24, 25, 32, 33, 34]$.

[2] The values of $\mathbf{A_{ss}^{(P)}}$ and $\mathbf{A_{ss}^{(Q)}}$ can be arbitrary, as long as their summation is the correct $\mathbf{A_{ss}}$. See §E for details.

using the last two equations in §4 to eliminate $\mathbf{u}_a, \mathbf{u}_b$:

$$\begin{bmatrix} \mathbf{u}_a \\ \mathbf{u}_b \end{bmatrix} = \begin{bmatrix} \mathbf{A}_{aa}^{-1} & \mathbf{0} \\ \mathbf{0} & \mathbf{A}_{bb}^{-1} \end{bmatrix} \left( \begin{bmatrix} \mathbf{v}_a \\ \mathbf{v}_b \end{bmatrix} - \begin{bmatrix} \mathbf{A}_{ar} & \mathbf{0} & \mathbf{A}_{as} \\ \mathbf{0} & \mathbf{A}_{bt} & \mathbf{A}_{bs} \end{bmatrix} \begin{bmatrix} \mathbf{u}_r \\ \mathbf{u}_t \\ \mathbf{u}_s \end{bmatrix} \right). \tag{6}$$

Let
$$\mathbf{P} :\overset{\mathcal{P}}{\leftarrow} \begin{bmatrix} \mathbf{P}^{rr} & \mathbf{P}^{rs} \\ \mathbf{P}^{sr} & \mathbf{P}^{ss} \end{bmatrix} \quad \mathbf{p} :\overset{\mathcal{P}}{\leftarrow} \begin{bmatrix} \mathbf{p}^r \\ \mathbf{p}^s \end{bmatrix} \tag{7}$$

$$\mathbf{Q} :\overset{\mathcal{P}}{\leftarrow} \begin{bmatrix} \mathbf{Q}^{tt} & \mathbf{Q}^{ts} \\ \mathbf{Q}^{st} & \mathbf{Q}^{ss} \end{bmatrix} \quad \mathbf{q} :\overset{\mathcal{P}}{\leftarrow} \begin{bmatrix} \mathbf{q}^t \\ \mathbf{q}^s \end{bmatrix} \tag{8}$$

The symbol $\mathbf{G} :\overset{\mathcal{P}}{\leftarrow} \mathbf{H}, \mathbf{g} :\overset{\mathcal{P}}{\leftarrow} \mathbf{h}$ means matrix $\mathbf{G}$ comes from entries of $\mathbf{H}$ but after some row/column-wise permutation $\mathbf{F} \in \mathcal{P}^{n \times n}$, so $\mathbf{G} := \mathbf{F}^{-1}\mathbf{H}\mathbf{F}, \mathbf{g} := \mathbf{F}^{-1}\mathbf{h}$, such that rows/-columns of $\mathbf{P}, \mathbf{Q}$ list boundary pixels counterclockwise in the ordering of Figure 4. See §E for details. The rationale behind dividing the domain is that once values at the boundary wire-frame $\mathbf{r}, \mathbf{s}, \mathbf{t}$ are known, the sub-problems to solve for $\mathbf{u}_a$ and $\mathbf{u}_b$ become independent: constructions of reduced systems $\mathbf{P}, \mathbf{Q}$ are independent of each other, concurrently computed in a size-2 batch.

### 3.2.2. SCHUR STEP: MERGE ADJACENT DtN

A Schur step merges two subdomains P and Q into a joint domain D, while converting their left-hand and right-hand sides $(\mathbf{P}, \mathbf{p})$ and $(\mathbf{Q}, \mathbf{q})$ into a new left-hand and right-hand sides $(\mathbf{D}, \mathbf{d})$. Divide the nodes in P into *contiguous* subsets $\alpha, \beta, \gamma, \delta, \epsilon$, such that $\alpha = [0, 1, 2, 3]$, $\beta = [4]$, $\gamma = [5, 6, 7]$, $\delta = [8]$, $\epsilon = [9, 10, 11, 12, 13, 14, 15]$. Divide the nodes in Q into contiguous subsets $\kappa, \lambda, \mu, \nu$, such that: $\kappa = [0]$, $\lambda = [1, 2, 3, 4, 5, 6, 7, 8, 9, 10, 11]$, $\mu = [12]$, $\nu = [13, 14, 15]$. Under the indexing in Figure 4, D's boundary consists of the nodes corresponding to $\alpha, \beta, \lambda, \delta, \epsilon$. After merging, $\beta, \kappa$ represent the same node , so are $\delta, \mu$; $\gamma$ represents the same set of nodes as $\nu$ but reversely ordered.

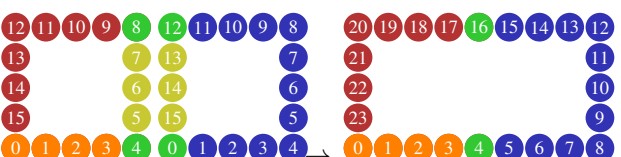

*Figure 4.* Schur step collapses subdomains P and Q into D.

The node grouping implies partitioning the matrix $\mathbf{P}, \mathbf{Q}$ into submatrices, by dividing the rows and columns into subsets.

$$\begin{bmatrix} \mathbf{P}_{\alpha\alpha} & \mathbf{P}_{\alpha\beta} & \mathbf{P}_{\alpha\gamma} & \mathbf{P}_{\alpha\delta} & \mathbf{P}_{\alpha\epsilon} \\ \mathbf{P}_{\beta\alpha} & \mathbf{P}_{\beta\beta} & \mathbf{P}_{\beta\gamma} & \mathbf{P}_{\beta\delta} & \mathbf{P}_{\beta\epsilon} \\ \mathbf{P}_{\gamma\alpha} & \mathbf{P}_{\gamma\beta} & \mathbf{P}_{\gamma\gamma} & \mathbf{P}_{\gamma\delta} & \mathbf{P}_{\gamma\epsilon} \\ \mathbf{P}_{\delta\alpha} & \mathbf{P}_{\delta\beta} & \mathbf{P}_{\delta\gamma} & \mathbf{P}_{\delta\delta} & \mathbf{P}_{\delta\epsilon} \\ \mathbf{P}_{\epsilon\alpha} & \mathbf{P}_{\epsilon\beta} & \mathbf{P}_{\epsilon\gamma} & \mathbf{P}_{\epsilon\delta} & \mathbf{P}_{\epsilon\epsilon} \end{bmatrix} := \mathbf{P}, \quad \begin{bmatrix} \mathbf{p}_\alpha \\ \mathbf{p}_\beta \\ \mathbf{p}_\gamma \\ \mathbf{p}_\delta \\ \mathbf{p}_\epsilon \end{bmatrix} := \mathbf{p},$$

$$\begin{bmatrix} \mathbf{Q}_{\kappa\kappa} & \mathbf{Q}_{\kappa\lambda} & \mathbf{Q}_{\kappa\mu} & \mathbf{Q}_{\kappa\nu} \\ \mathbf{Q}_{\lambda\kappa} & \mathbf{Q}_{\lambda\lambda} & \mathbf{Q}_{\lambda\mu} & \mathbf{Q}_{\lambda\nu} \\ \mathbf{Q}_{\mu\kappa} & \mathbf{Q}_{\mu\lambda} & \mathbf{Q}_{\mu\mu} & \mathbf{Q}_{\mu\nu} \\ \mathbf{Q}_{\nu\kappa} & \mathbf{Q}_{\nu\lambda} & \mathbf{Q}_{\nu\mu} & \mathbf{Q}_{\nu\nu} \end{bmatrix} := \mathbf{Q}, \quad \begin{bmatrix} \mathbf{q}_\kappa \\ \mathbf{q}_\lambda \\ \mathbf{q}_\mu \\ \mathbf{q}_\nu \end{bmatrix} := \mathbf{q}.$$

(9) is the linear system for the joint domain D. Since $\beta, \kappa$ represent the same node, they correspond to the same row/-column; the same applies to $\delta, \mu$. Now we assemble the new system $\mathbf{D}$ by Schur "involuting" the sub-systems $\mathbf{P}, \mathbf{Q}$. When eliminating the "wire-frame" nodes $\gamma$ (i.e. $\nu$ in reverse order), introduce the symbols to simplify notation:

$$\mathbf{X} := \begin{bmatrix} \mathbf{P}_{\alpha\alpha} & \mathbf{P}_{\alpha\beta} & \mathbf{0}_{\alpha\lambda} & \mathbf{P}_{\alpha\delta} & \mathbf{P}_{\alpha\epsilon} \\ \mathbf{P}_{\beta\alpha} & \mathbf{P}_{\beta\beta} + \mathbf{Q}_{\kappa\kappa} & \mathbf{Q}_{\kappa\lambda} & \mathbf{P}_{\beta\delta} + \mathbf{Q}_{\kappa\mu} & \mathbf{P}_{\beta\epsilon} \\ \mathbf{0}_{\lambda\alpha} & \mathbf{Q}_{\lambda\kappa} & \mathbf{Q}_{\lambda\lambda} & \mathbf{Q}_{\lambda\mu} & \mathbf{0}_{\lambda\epsilon} \\ \mathbf{P}_{\delta\alpha} & \mathbf{P}_{\delta\beta} + \mathbf{Q}_{\mu\kappa} & \mathbf{Q}_{\mu\lambda} & \mathbf{P}_{\delta\delta} + \mathbf{Q}_{\mu\mu} & \mathbf{P}_{\delta\epsilon} \\ \mathbf{P}_{\epsilon\alpha} & \mathbf{P}_{\epsilon\beta} & \mathbf{0}_{\epsilon\lambda} & \mathbf{P}_{\epsilon\delta} & \mathbf{P}_{\epsilon\epsilon} \end{bmatrix}$$

$$\mathbf{Y} := \begin{bmatrix} \mathbf{P}_{\alpha\gamma} \\ \mathbf{P}_{\beta\gamma} + \mathbf{Q}_{\kappa\nu}\mathbf{J} \\ \mathbf{Q}_{\lambda\nu}\mathbf{J} \\ \mathbf{P}_{\delta\gamma} + \mathbf{Q}_{\mu\nu}\mathbf{J} \\ \mathbf{P}_{\epsilon\gamma} \end{bmatrix} \quad \mathbf{y} := \begin{bmatrix} \mathbf{p}_\alpha \\ \mathbf{p}_\beta + \mathbf{q}_\kappa \\ \mathbf{q}_\lambda \\ \mathbf{p}_\delta + \mathbf{q}_\mu \\ \mathbf{p}_\epsilon \end{bmatrix} \quad \mathbf{x} := \begin{bmatrix} \mathbf{u}_\alpha \\ \mathbf{u}_\beta \\ \mathbf{u}_\lambda \\ \mathbf{u}_\delta \\ \mathbf{u}_\epsilon \end{bmatrix}$$

$$\mathbf{Z} := \begin{bmatrix} \mathbf{P}_{\gamma\alpha} & \mathbf{P}_{\gamma\beta} + \mathbf{J}^\mathsf{T}\mathbf{Q}_{\nu\kappa} & \mathbf{J}^\mathsf{T}\mathbf{Q}_{\nu\lambda} & \mathbf{P}_{\gamma\delta} + \mathbf{J}^\mathsf{T}\mathbf{Q}_{\nu\mu} & \mathbf{P}_{\gamma\epsilon} \end{bmatrix}$$

$$\mathbf{W} := \begin{bmatrix} \mathbf{P}_{\gamma\gamma} + \mathbf{J}^\mathsf{T}\mathbf{Q}_{\nu\nu}\mathbf{J} \end{bmatrix} \quad \mathbf{w} := \begin{bmatrix} \mathbf{p}_\gamma + \mathbf{J}^\mathsf{T}\mathbf{q}_\nu \end{bmatrix}$$

where $\mathbf{J} \equiv \mathbf{J}^\mathsf{T}$ is the reverse permutation matrix (see §E), whose action is to reverse the rows (or columns) of a matrix when being multiplied with from left (or right). By plugging in our new definitions, (9) becomes:

$$\begin{bmatrix} \mathbf{X} & \mathbf{Y} \\ \mathbf{Z} & \mathbf{W} \end{bmatrix} \begin{bmatrix} \mathbf{x} \\ \mathbf{u}_\gamma \end{bmatrix} = \begin{bmatrix} \mathbf{y} \\ \mathbf{w} \end{bmatrix} \tag{10}$$

Schur eliminate $\mathbf{u}_\gamma$ using:

$$\mathbf{u}_\gamma = \mathbf{W}^{-1}(\mathbf{w} - \mathbf{Z}\mathbf{x}) \quad \text{(back-fill)}, \tag{11}$$

the system (4) to (5) to (9) finally becomes $\mathbf{Dx} = \mathbf{d}$, where:

$$\mathbf{D} := (\mathbf{X} - \mathbf{YW}^{-1}\mathbf{Z}) \quad \mathbf{d} := \mathbf{y} - \mathbf{YW}^{-1}\mathbf{w} \tag{12}$$

In summary, to solve the original problem (4) in the special case of two subdomains, after the Schwarz step that constructs the system matrices $\mathbf{P}, \mathbf{Q}$, our final algorithm of the Schur step first solves for the values $\mathbf{x} = \mathbf{D}^{-1}\mathbf{d}$, then recovers $\mathbf{u}_\gamma$—the solution at the interface—using (11), and finally recovers $\mathbf{u}_a, \mathbf{u}_b$—values in the interior of each patch from $\mathbf{u}_r, \mathbf{u}_t, \mathbf{u}_s$—values at the patch's boundary.

### 3.2.3. GENERAL CASES

Summarized in Figure 2, when there are $2^k$ patches of $5 \times 5$, we apply the Schwarz step once, and the Schur step $k$ times to recursively glue every two subdomains together. These steps play a role similar to LU factorization (Davis, 2006). See full details in §C, §E and Algorithm 1.

The Schwarz step to eliminate interior nodes in the $3 \times 3$ block is basically the same as introduced before: for each patch P the interior nodes $a$ are eliminated and the system matrix for the remaining nodes $r$ becomes $(\mathbf{A}_{rr}^{(P)} - \mathbf{A}_{ra}^{(P)}\mathbf{A}_{aa}^{-1}\mathbf{A}_{ar}^{(P)})$.

$$\left(\begin{bmatrix}\begin{bmatrix}\mathbf{P}_{\alpha\alpha} & \mathbf{P}_{\alpha\beta} & 0_{\alpha\lambda} & \mathbf{P}_{\alpha\delta} & \mathbf{P}_{\alpha\epsilon} \\ \mathbf{P}_{\beta\alpha} & \mathbf{P}_{\beta\beta} & 0_{\beta\lambda} & \mathbf{P}_{\beta\delta} & \mathbf{P}_{\beta\epsilon} \\ 0_{\lambda\alpha} & 0_{\lambda\beta} & 0_{\lambda\lambda} & 0_{\lambda\delta} & 0_{\lambda\epsilon} \\ \mathbf{P}_{\delta\alpha} & \mathbf{P}_{\delta\beta} & 0_{\delta\lambda} & \mathbf{P}_{\delta\delta} & \mathbf{P}_{\delta\epsilon} \\ \mathbf{P}_{\epsilon\alpha} & \mathbf{P}_{\epsilon\beta} & 0_{\epsilon\lambda} & \mathbf{P}_{\epsilon\delta} & \mathbf{P}_{\epsilon\epsilon} \\ [\mathbf{P}_{\gamma\alpha} & \mathbf{P}_{\gamma\beta} & 0_{\gamma\lambda} & \mathbf{P}_{\gamma\delta} & \mathbf{P}_{\gamma\epsilon}]\end{bmatrix} \begin{bmatrix}\mathbf{P}_{\alpha\gamma} \\ \mathbf{P}_{\beta\gamma} \\ 0_{\lambda\gamma} \\ \mathbf{P}_{\delta\gamma} \\ \mathbf{P}_{\epsilon\gamma} \\ [\mathbf{P}_{\gamma\gamma}]\end{bmatrix} \\ \end{bmatrix} + \begin{bmatrix}\begin{bmatrix}0_{\alpha\alpha} & 0_{\alpha\beta} & 0_{\alpha\lambda} & 0_{\alpha\delta} & 0_{\alpha\epsilon} \\ 0_{\beta\alpha} & \mathbf{Q}_{\kappa\kappa} & \mathbf{Q}_{\kappa\lambda} & \mathbf{Q}_{\kappa\mu} & 0_{\beta\epsilon} \\ 0_{\lambda\alpha} & \mathbf{Q}_{\mu\kappa} & \mathbf{Q}_{\lambda\lambda} & \mathbf{Q}_{\lambda\mu} & 0_{\lambda\epsilon} \\ 0_{\delta\alpha} & \mathbf{Q}_{\mu\kappa} & \mathbf{Q}_{\mu\lambda} & \mathbf{Q}_{\mu\mu} & 0_{\delta\epsilon} \\ 0_{\epsilon\alpha} & 0_{\epsilon\beta} & 0_{\epsilon\lambda} & 0_{\epsilon\delta} & 0_{\epsilon\epsilon} \\ [0_{\gamma\alpha} & \mathbf{J}^\intercal\mathbf{Q}_{\nu\kappa} & \mathbf{J}^\intercal\mathbf{Q}_{\nu\lambda} & \mathbf{J}^\intercal\mathbf{Q}_{\nu\mu} & 0_{\gamma\epsilon}]\end{bmatrix} \begin{bmatrix}0_{\alpha\gamma} \\ \mathbf{Q}_{\kappa\nu}\mathbf{J} \\ \mathbf{Q}_{\lambda\nu}\mathbf{J} \\ \mathbf{Q}_{\mu\nu}\mathbf{J} \\ 0_{\epsilon\gamma} \\ [\mathbf{J}^\intercal\mathbf{Q}_{\nu\nu}\mathbf{J}]\end{bmatrix} \\ \end{bmatrix}\right)\begin{bmatrix}\mathbf{u}_\alpha \\ \mathbf{u}_\beta \\ \mathbf{u}_\lambda \\ \mathbf{u}_\delta \\ \mathbf{u}_\epsilon \\ \mathbf{u}_\gamma\end{bmatrix} = \begin{bmatrix}\mathbf{p}_\alpha \\ \mathbf{p}_\beta + \mathbf{q}_\kappa \\ \mathbf{q}_\lambda \\ \mathbf{p}_\delta + \mathbf{q}_\mu \\ \mathbf{p}_\epsilon \\ \mathbf{p}_\gamma + \mathbf{J}^\intercal\mathbf{q}_\nu\end{bmatrix}$$
(9)

*Figure 5.* The joint linear system that merges two subdomains.

The step transforms the domain to a wire-frame with $2^k$ hollow subdomains, each of which initially has one patch and is associated with a system matrix (like $\mathbf{P}, \mathbf{Q}, ...$) and a right-hand side ($\mathbf{p}, \mathbf{q}, ...$). Again, this stage takes a small fraction of overall runtime but removes the majority of pixels.

A Schur step further simplifies the wire-frame by eliminating pixels on some "edges" along the $x$ (or $y$) direction for $j$ that is odd (or even). The $j$-th Schur step ($j = 1, ..., k$) starts with $2^{k-j+1}$ subdomains that each consists of $2^{j-1}$ patches, and ends with $2^{k-j}$ subdomains that each has $2^j$ patches. To merge each pair of adjacent subdomains that are marked in red and blue, we gather the system matrices for them, also denoted as $\mathbf{P}, \mathbf{Q}$, and apply the formula (12).

**Tensorized representations of batched reduced systems** are adopted as our method's data structure: we never maintain the global system $\mathbf{A}$ as a sparse matrix like all other direct solvers. Instead, in the $j$-th Schur step of our algorithm, the linear system is a 4-dim tensor $\boldsymbol{\alpha}^{(j)}$ that $\boldsymbol{\alpha}^{(j)}[k, l, :, :]$ stores the left-hand side of the $(k, l)$-th subdomain. For $j = (2i + 1)$, the $j$-th Schur step takes as input $\boldsymbol{\alpha}^{(j)}$ of size $(2^{k/2-i}, 2^{k/2-i}, 16 \cdot 2^i, 16 \cdot 2^i)$, and outputs $\boldsymbol{\alpha}^{(j+1)}$ of size $(2^{k/2-i-1}, 2^{k/2-i}, 24 \cdot 2^i, 24 \cdot 2^i)$; $\boldsymbol{\alpha}^{(j)}$ contains all reduced systems; $\mathbf{P}, \mathbf{Q}$ are fetched batch-wise from $\boldsymbol{\alpha}^{(j)}$ for all red and blue subdomains, e.g., in PyTorch syntax:

```
1 P = a{j}[0::2, :, :, :] # even subdomains
2 Q = a{j}[1::2, :, :, :] #  odd subdomains
```

**Batched dense linear algebras** are employed for automatic parallelisms in PyTorch. For superior performance, it is critical to batch matrix operations including inversion, multiplication, sum, slicing into rows and columns, etc. For example, $\mathbf{W}, \mathbf{X}, \mathbf{Y}, \mathbf{Z}$ are 4-dim tensors obtained by batch-wise slicing into the last two dimensions of $\mathbf{P}, \mathbf{Q}$, e.g., $\mathbf{W} := \mathbf{P}[:, :, \gamma, \gamma] + \mathbf{J}^\intercal\mathbf{Q}[:, :, \nu, \nu]\mathbf{J}$, and the following code computes the merged system across all subdomains:

```
1 D = X - Y * torch.linalg.inv(W) * Z
```

### 3.3. Discussion

While our method is simple to describe—recursively applying the Schur complement formula in batches, fully understanding the motivation and appreciating the superiority in performance and numerical behavior, empirically demonstrated in §4, requires familiarity with references in numerical methods and elliptic PDEs. We highlight a few aspects.

One question is why the matrices $\mathbf{D}, \mathbf{P}, \mathbf{Q}$ are numerically well-behaving and the sum of their sub-blocks $\mathbf{W}$ is invertible. When the system $\mathbf{A}$ comes from discretizing an elliptic PDE, including the "Laplacian matrix" considered in graph

theory and computer vision, which means $\mathbf{A}$ is positive semidefinite (PSD) and $\mathbf{A}_{ii} = -\sum_{j:j\neq i} \mathbf{A}_{ij}$, the Schur complements $\mathbf{D}, \mathbf{P}, \mathbf{Q}$ are also PSD, and discretizing the Dirichlet-to-Neumann (DtN) operator, which is well-defined and enjoys numerous nice theoretical properties. See Wang et al. (2018); Levitin et al. (2023) and references therein.

DtN also plays a central role in the study of domain decomposition in numerical methods (Quarteroni & Valli, 1999). Consider the classic example of solving Laplace's equation under Dirichlet boundary condition. After dividing the image domain into subdomains, however, the usual problem arises: Dirichlet data at the interface between subdomains are *unknown*, preventing naïve divide-and-conquer. But whatever the Dirichlet data at the interface is, the solution is linear to it (through the Green's function), since we work with a linear PDE: the DtN operator encodes the solution operator that yields the solution for whatever Dirichlet data—a simple intuition why DtN is central in domain composition (Quarteroni & Valli, 1999). Unlike common practices in larger problems that implicitly realize DtN's matrix-vector-product (Liu et al., 2016), our algorithm maintains the DtN/solution operators *explicitly as dense matrix*. Unlike general-purpose sparse solvers, we do not aim at a solver with minimal floating-point operations or memory consumption and reducing fill-in sparsity during the solving, for very large-scale problems with billions of variables. Instead, we focus on runtime for common image sizes, such as 1024×1024 with around 1 million variables. As a limitation: our method is more demanding in memory (see §D.4).

**A note on novelty.** We remark that key ingredients of our algorithm exist in the literature. We essentially perform Gaussian elimination with a particular pivoting ordering induced by the nested dissection (George, 1973). Similar procedures were applied in early studies of distributing FEM solves among multiple processors, e.g., in civil (Farhat & Wilson, 1987; Farhat et al., 1987; Farhat & Roux, 1991) and ocean engineering (Rakowsky, 1999). However, our setting is different from scientific computing communities, which focus on much larger problems where inter-processor communication rises as a central consideration. Without recent advances in GPUs both in terms of high throughput and memory size, stirred by the needs of deep learning, applying the Schur formula to invert a massive amount of small dense matrices on a single device is either infeasible or uncompetitive compared to alternative direct solvers. Solving a massive amount of dense systems on a single device—without needing cross-device communication, is made efficiently only recently, to rise as the best practice.

*Table 2.* Uncertain runtime of iterative solvers. When solving an anisotropic or isotropic Laplacian (Dirichlet) system, our solver is not affected by anisotropy, while Ruge-Steuben AMG (Bell et al., 2022) can be much slower compared to solving isotropic systems.

| | anisotropic Laplacian | | | | | | | isotropic Laplacian | | |
|---|---|---|---|---|---|---|---|---|---|---|
| Example | CUDA | SciPy | ours | ours-64 | AMG (1e-4) | AMG (1e-7) | AMG (1e-10) | AMG (1e-4) | AMG (1e-7) | AMG (1e-10) |
| $2049^2$ | 21354 | 143100 | 158.5 | OOM | 12205 | 49487 | 53877 | 6559 | 7757 | 9202 |
| $1025^2$ | 4710 | 16512 | 36.45 | 175.6 | 2678 | 8112 | 13446 | 1388 | 1754 | 2236 |
| $513^2$ | 1036 | 2051 | 10.90 | 31.81 | 832.2 | 1658 | 2353 | 321.9 | 387.2 | 515.3 |
| $257^2$ | 234 | 355 | 5.82 | 7.46 | 203.0 | 385.2 | 556.0 | 91.0 | 117.3 | 347.8 |

# 4. Results and Validations

We compare our method with direct and iterative solvers and demonstrate significant advantages in runtime. The speedup from our approach can potentially be much more significant than what is reported here. Compared to standard solvers that are already highly optimized, our algorithm is naïvely prototyped in PyTorch—re-implemented as low-level operators or optimization at the hardware level like CUDA operators will result in further acceleration. Additionally, as a direct solver, our back-substitution variant enjoys extra speedups in repetitive solves by reusing numerical factorization—impossible for iterative solvers: Table 7.

**Comparison with direct solvers.** As the performance of direct solvers, especially our method, is less affected by $\mathbf{A}$, Table 1 can represent the timing comparison for whatever $\mathbf{A}$. Our method supports both single and double floating-point numbers (float-32 and float-64), while existing direct solvers only support float-64, or support float-32 without major speedups (cuDSS). Implemented in float-64, our method often achieves a relative error of $10^{-16}-10^{-12}$ even better than SciPy (Table 9). Our method can use float-32 for further speedup, in which case the error range from $10^{-6}-10^{-5}$, a typical error tolerance that iterative solvers use, sufficient for many tasks. We evaluate the accuracy of the solution using the standard metric—the (relative) error tolerance:

$$\text{tol}(\mathbf{x}) := \|\mathbf{Ax} - \mathbf{b}\|_2 / \|\mathbf{b}\|_2 \tag{13}$$

which is the stop criterion for most iterative methods. For direct solvers, we compare with SciPy (Virtanen et al., 2020) and CUDA (Nickolls et al., 2008)—the cuDSS LDU method which defaults to the METIS (nested dissection) ordering. For iterative solvers, we compare with the algebraic multi-grid solver (Ruge-Steuben) from PyAMG (Bell et al., 2022). For SciPy, we use the "spsolve" function, which defaults to the COLAMD ordering and is backended by SuperLU (Li, 2005). While CUDA can be faster than SciPy, it is still on the same order of magnitude and cannot achieve interactive rates like ours. Our experiments are collected on an Nvidia GPU A6000 Ada 48GB and an Intel CPU Xeon w9-3475X.

**Comparison with iterative solvers.** §F.2 extensively discusses why iterative solvers are problematic modules in deep learning pipelines. Although we compare with iterative methods and demonstrate advantages, direct methods remain our primary theoretical point of comparison. Direct solvers are reliable and robust with consistent runtime;

in contrast, iterative solvers, heavily dependent on $\mathbf{A}$, in challenging cases significantly slow down or explode.

*Table 3.* Runtime of a single linear solve on a few $513^2$ problems, to the relative tolerance of $10^{-6}$ for iterative solvers. No iterative solvers out-of-the-box can perform well on all tasks. "(s)": solvers benefit from the symmetry assumption when it holds, while other methods do not. "$\times k$": time for iterative solvers should be multiplied by an extra factor of $2 \sim 20+$ in some applications, since iterative solvers need to run 2 times to differentiate, and $20+$ times in eigen solving, while direct solvers do the factorization once and reuse it. "$\times 2$": should be multiplied by an extra 2 due to transposed solve to differentiate. "NaN": the solver diverges or fails to yield a small error. "*" indicates stopping at a larger tolerance due to different stopping criteria; those $> 10^{-5}$ are: a: 1.19e-5, b: 1.16e-4, c: 1.41e-4, d: 1.41e-4, e: 1.41e-4.

| | Lap 1e-2 | Lap 1e-4 | Hel 0.01 | Hel 0.1 | Hel 1.0 | R-Deconv |
|---|---|---|---|---|---|---|
| Our-32 | 12.3 | 12.2 | 12.3 | 12.2 | 12.2 | NaN |
| Our-64 | 35.6 | 35.4 | 35.6 | 35.5 | 35.5 | NaN |
| SciPy-LU | 2208 | 2487 | 2333 | 2687 | 2133 | 3145 |
| CUDSS-LDU | 1011 | 1029 | 1040 | 1037 | 1020 | 1038 |
| Our-64-backsub | 3.0 | 3.0 | 3.0 | 3.0 | 3.0 | 3.0 |
| (s) CG | 65.1 | 459 | 563 | 4142 | 6442 | NaN |
| (s) MINRES | 97.3 | 1319 | (*) 1595 | (*) 4051 | (*) 11156 | NaN |
| LSMR | 7288 | 426499 | 76928 | 103569 | 80617 | 12282 |
| LSQR | (*a) 2896 | (*b) 215758 | (*c) 84442 | (*d) 20718 | (*e) 16158 | (*) 3290 |
| biCGstab | 299 | 1898 | NaN | NaN | NaN | NaN |
| CGS | 58.3 | 922 | NaN | NaN | NaN | NaN |
| DIOM | 228 | 12716 | 1484 | 4433 | 13221 | NaN |
| FOM | 1157 | 499930 | 61344 | 574742 | NaN | NaN |
| QMR | 70.8 | 570 | 463 | 1500 | 6849 | NaN |
| BILQ | 86.5 | 634 | 635 | 2087 | 6878 | NaN |
| GMRES | 954 | 31017 | 56478 | 526975 | 4675551 | NaN |
| DQGMRES | 172 | 1029 | 1391 | 4323 | 12885 | NaN |
| FGMRES | 933 | 30793 | 56665 | 537350 | NaN | NaN |
| FGMRES+AMG | 1815 | 12917 | (*)3610256 | (*)7840145 | (*)13619368 | NaN |
| GMRES+AMG | 563 | 502 | NaN | NaN | NaN | NaN |
| (s) PCG+AMG | 408 | 303 | 509 | (*) 26741 | NaN | NaN |

In Table 3, we test all methods on solving the image matting Laplacian $\mathbf{A} = \mathbf{L} + \lambda\mathbf{M}$ (the $\mathbf{L}$ in §B.3, which is highly anisotropic, $\lambda = 10^{-2}, 10^{-4}$), the Helmholtz equation $\mathbf{A} = \mathbf{L} - \kappa^2\mathbf{M}$ under wave numbers $\kappa^2 = 1/64\{0.01, 0.1, 1.0\}$ (Figure 10), and random $\mathbf{A}$—a sparse matrix whose nonzero entries $\mathbf{A}_{ij}$ are i.i.d., uniformly drawn from $[-1, 1]$. We compare with out-of-the-box GPU linear solvers from CuPy (Okuta et al., 2017) and Julia (Bezanson et al., 2017), using the CUDA implementation in the Krylov.jl package (Montoison & Orban, 2023). We are primarily interested in indefinite and nonsymmetric square linear solvers including: LSQR, LSMR, CGS, biCGstab, DIOM. FOM, QMR, BILQ, GMRES, DQGMRES, and FGMRES. While we also report results on CG and MINRES, please note that the comparison may be slightly unfair: they benefit from the symmetry assumption that other methods do not. These out-of-the-box iterative solvers can slow down by many

orders of magnitude or fail to converge for harder problems. PCG+AMG: using Nvidia AMGX (Naumov et al., 2015), we compare with the preconditioned conjugate gradient (PCG) with AMG preconditioning; See §F for details of other AMGX solvers. No iterative numerical scheme works well for all problems—changing the parameters that generate the problem slows them down significantly. AMG preconditioners improve the CG/PCG and GMRES for elliptic PDEs but hurt for out-of-scope uses; see Figure 10. In contrast, our solver is problem-independent.

**Anisotropic Laplacian.** We solve Poisson equation with random or constant right-hand sides, with an isotropic or anisotropic kernel. As shown in Table 2, iterative solvers can slow down significantly with anisotropic kernels, while direct solvers are less affected. Table 2 reports the time to solve Laplace equation with Dirichlet condition with either the isotropic kernel $\mathbf{K}_1$, or the anisotropic $\mathbf{K}_2$ in Figure 9. The number of iterations increases for anisotropic kernels by a factor of $6\times$, with the same number of non-zeros (0s in $\mathbf{K}_1$ are treated as non-zero entries with value 0.0).

**Parabolic PDEs and high-precision diffusion kernels.** The parabolic PDE $\frac{du(x,t)}{dt} = \Delta u(x,t)$ is solved by our method. The standard practice to approximate $u(x,t)$ is the implicit Euler scheme that solves the system $\mathbf{A} \leftarrow t\mathbf{L} + \mathbf{M}$. Heat geodesics is a situation in which the **high precision of direct solvers within a tolerance of** $10^{-10}$ becomes critical. Crane et al. (2017) approximate geodesic distance from a point $x_0$, using the solution under the initial condition $u(x,0) = \delta(x - x_0)$. The right-hand side is a delta function that is nonzero only at one pixel. The solution $u(x,t)$, known as the heat kernel, is a Gaussian-like function centered at $x_0$, with infinite support. Then geodesics are recovered from $\log u(x,t)$—logarithm is of direct interests. $\mathbf{x}$ has to be computed within a tolerance of $10^{-10}$ or even $10^{-15}$ to produce non-negative heat kernel values to apply the heat method (Crane et al., 2017). Requiring a tiny tolerance makes iterative methods incompetent, while direct solvers can be simply applied. Figure 8 shows the result of our method. Our method (float-64) solves the system to an error tolerance of $10^{-16}$, smaller than that of SciPy. Our method can succeed under extreme numerics even when SciPy fails, by decreasing the stepsize $t \to 0$.

### 4.1. A zero-shot baseline of efficient PDE solvers

Learned PDE solvers emerge in scientific tasks (Lu et al., 2019; 2021; Raissi et al., 2019; Lagaris et al., 1998). In addition to surrogate modeling (*e.g.*, when the governing PDEs are unknown and must be inferred from data), a major advantage of learned solvers over conventional numerical methods is speed: the former can be 3 orders-of-magnitude faster, the main inspiration for our work. Surprisingly, we demonstrate a *zero-shot* approach also capable of improving

runtime by only leveraging GPUs, without needing training.

A successful line of research achieves orders-of-magnitude speedup in solving the PDE (2) which is also known as a Darcy flow, by learning the *coefficient-to-solution* operator (Li et al., 2020; Wang et al., 2021; Li et al., 2024). Our solver can implement FEM to realize the same operator, which becomes the mapping: $\mathbf{c} \to \mathbf{x} = \mathbf{A}^{-1}\mathbf{b}$ where $\mathbf{A} := \mathbf{G}^\mathsf{T}\mathrm{diag}([\mathbf{c}, \mathbf{c}])\mathbf{G}$. We compare with the state-of-the-art learning solver on the task of Darcy flows. We test on 1024 Darcy flow examples available with the Python package "neuraloperator" from Li et al. (2024) resampled from $421^2$ to $257^2$. Our method as a direct solver does not need any training and solves the linear system within a tolerance of $10^{-12}$ (float 64) on all examples. For our method the relative "errors" are $0.869\% \pm 0.0927\%$, smaller than that of $1.56\%$ in Li et al. (2024). Note that the "error" of our method, i.e., differences between our result and the ground truth, is due solely to the use of different numerical schemes. Our method, if used as the simulation engine, *is the ground truth*, achieving comparable "inference" speed without any training. Ours at $257^2$ has a running time of $5 - 7$ ms, comparable with many neural architectures—Darcy flows take a few milliseconds for some recent methods (Li et al., 2025).

The results suggest the suitability of our method as an efficient solver-in-the-loop importable by existing architectures, such as solution super-resolution (Ren et al., 2023). As learning-based solvers increasingly import modules from numerical PDEs, it is promising to generalize our involution to procedures with learnable parameters for data-driven discretizations (Wang et al., 2019; Bar-Sinai et al., 2019).

## 5. Applications

Featured in Figure 1, due to the foundational role of the linear solver, numerous applications immediately benefit from our method, removing the need to design problem-specific solvers. For $\mathbf{A}$ coming from different applications, our solver consistently demonstrates speedup over other direct solvers to a level similar to the case of solving PDEs in §4. Examples of sparse $\mathbf{A}$ encompass:

**Mathematical optimization.** When $\mathbf{A}, \mathbf{b}$ are the Hessian and gradient, our solver implements the **Newton's method**. See §B.1 for detailed discussions and experiments. Despite theoretical superiority, Newton's method has very limited applications in practice due to the expensive Hessian solves, an obstacle that our orders-of-magnitude speedup removes. Similar use cases include shape optimization, Gaussian processes, and inverse rendering (Nicolet et al., 2021).

**The Laplacian paradigms** for graph theory (Chung, 1997), spectral clustering, manifold learning, image/geometry processing, and vision have been extremely successful; see Wang & Solomon (2019); Chen et al. (2021b) for a sur-

vey. Early examples include: Laplace equations in optical flow (Horn & Schunck, 1981), semi-supervised learning with harmonic label propagation (Zhu et al., 2003), Perona–Malik model with anisotropic diffusion (Perona & Malik, 1990), and normalized cuts for segmentation (Shi & Malik, 2000). These are examples where PDE or linear algebraic solvers play a central role and $\Delta^{-1}$ or $(t\Delta + \mathbf{I})^{-1} \approx e^{-t\Delta}$, instead of the forward operator $\Delta$, is of primary interest. Sparse solvers play an even more prominent role in geometry modeling, where almost every algorithm solves some form of Laplacian systems (Solomon et al., 2014).

**Image processing: matting, segmentation, and editing.** $\mathbf{A}$ can be a graph or matting Laplacian $\mathbf{L}$ that $\mathbf{L}_{ij}$ encodes the similarity between adjacent pixels. In this setting, $\mathbf{L}$ can be viewed as discretizing an anisotropic Laplace equation. Our solver $\mathbf{L}^{-1}\mathbf{b}$ can be used for spectral segmentation (Shi & Malik, 2000): see §B.4. Often a constraint term is added to the linear system $\mathbf{A} := \mathbf{L} + \lambda \mathbf{R}\mathbf{R}^\intercal$, in image matting, and Poisson image editing (Pérez et al., 2003). While the constraint term $\mathbf{R}\mathbf{R}^\intercal$ makes the problem's difficulty unevenly distributed across the image and a large $\lambda$ makes the system ill-conditioned, our method in double precision consistently yields an error smaller than SciPy. See details in §B.3.

**Diffeomorphic image registration.** Diffeomorphisms, or smooth bijective maps between image domains (Younes, 2010), are not only a central concept in differential manifolds and geometry, but also foundational in image registration (Beg et al., 2005). Recent results in variational quasi-harmonic maps (Wang et al., 2023) demonstrate that the map $\mathbf{x} \rightarrow (u(\mathbf{x}), v(\mathbf{x}))$ is diffeomorphic if and only if $\nabla \cdot [\mathbf{C}(\mathbf{x})\nabla u(\mathbf{x})] = 0$ (resp. $v(\mathbf{x})$) for some $\mathbf{C}(\mathbf{x})$, with appropriate boundary conditions. Intuitively, this condition states that each pixel must be placed about some weighted average of its neighbors' positions, which is enforced by a sparse linear system. To compute diffeomorphic image registration in Figure 7, we use Wang et al. (2023), which recursively solves anisotropic Laplacian systems. Replacing their linear solver with ours as the backend again leads to orders-of-magnitude speedups. Registering a pair of images of size $1024 \times 768$ requires solving 150 sparse systems sequentially and takes 5 seconds with our solver, compared to 8 minutes in Wang et al. (2023) if using a SciPy backend.

**Geometry algorithms.** §4 shows how to reduce geometric PDEs to anisotropic PDEs: evaluate $\mathbf{L}, \mathbf{M}$ using a curved surface's metric, pushed back to the plane. §B.5 handles non-disk topology with appropriate boundary conditions.

**Generalized deconvolution.** As shown in Figure 6, we recover an image from its convolution with a spatially varying kernel in the form (1), which reduces to solving a linear system (Hansen et al., 2006). To our knowledge, the only algorithm applicable to this setting is (equivalent to) using linear solvers (since solving PDE reduces to this problem).

Our method is again orders-of-magnitude faster than SciPy: on $257^2$ images, SciPy has an average runtime of 494.4ms, slowing down compared to solving the Laplacian in Table 2, while it takes 7.51ms for ours (float-64). As expected, PyAMG fails for kernel $a_2^{(x,y)}$ since it is not symmetric.

**(Differentiable) physics simulation.** When $\mathbf{A}$ comes from discretizing a physics-based energy, our method immediately accelerates the physical simulator. §B.2 shows an example. Robot learning engines (Du et al., 2021; Hu et al., 2019) and scientific machine learning (Sanchez-Gonzalez et al., 2020) are increasingly relying on solver-in-the-loop (Amos & Kolter, 2017) using CUDA sparse solvers, which can be replaced with ours.

**Handling different boundary conditions.** Beyond handling the usual Dirichlet or Neumann boundary condition, we even introduce a midpoint reflective boundary condition to handle no-disk topology like spheres. See §C,§D.5.

# 6. Conclusion and Future Work

It is observed that in many areas—from medical image analysis to learning for solving PDEs—traditional optimization-based methods can be orders-of-magnitude slower compared to deep learning. We identify that this is partially because the implementation of conventional methods is not specially tailored to a new setting of HPC (high performance computing) on a single workstation, enabled by GPUs. In interactive vision and graphics, considerable effort has been devoted to designing numerical schemes that avoid direct linear solvers, with the assumption that exact linear solvers are too slow. Whether or not the computation time is within tens of milliseconds makes an *essential* difference for the deployability of algorithms in video conferencing, self-driving cars, and virtual reality environment. For example, Bro-Nielsen & Cotin (1996) note *"speed is everything"* in virtual surgery. Our method eliminates the need for customizing solvers, and revives methods that used to be slow due to (repetitive) linear solvers in a straightforward manner. Our method can serve as a strong baseline for learning-based PDEs, or an efficient solver/simulator-in-the-loop in robot learning or vision pipelines (Barron & Poole, 2016).

Due to the foundational role of sparse linear solvers, our method can potentially impact a broad range of applications beyond what is presented. Many classical algorithms can be viewed as one or more steps of the Schwarz–Schur involution—solving a linear Laplacian-like system (Barron & Poole, 2016; Pérez et al., 2003; Germer et al., 2020; Shi & Malik, 2000). For future work, we plan to design neural architectures based on the "involution," generalizing optimization-based Laplacian paradigms, and extend our method to work with an arbitrary mesh or graph. Our exact solver can be converted to an approximate solver by downsampling the DtN and integrated with iterative schemes.

## Impact Statement

This paper presents work whose goal is to advance the field of Machine Learning, Optimization, Computer Vision, and Scientific Computing. We call attention to the fact that even for a fundamental and well-studied task—solving linear systems, speedups of up to $1000\times$ are still achievable, demonstrating the potential for substantial performance gains by developing novel methods like our solver to best leverage hardware capacity. We suspect that perhaps the advances in GPUs have been underutilized in terms of improving conventional methods, with or without relying on deep learning. Possible societal impacts of our work include *improved energy efficiency* and reduced carbon footprint: significantly reducing computational time, methods like ours lower power consumption and contribute to a more *sustainable and environmentally friendly computing* landscape.

## Acknowledgements

We thank Justin Solomon and Mike Taylor for insightful discussions. We thank the anonymous reviewers, especially for their comments on improving clarity, self-containment, and accessibility to the broader ICML community. YB is supported by a fellowship from the Royal Society (NIF-R1-232460). Support for this research was provided in part by the BRAIN Initiative Cell Atlas Network (BICAN) grants U01MH117023 and UM1MH130981, the Brain Initiative Brain Connects consortium (U01NS132181, 1UM1NS132358-01), the National Institute for Biomedical Imaging and Bioengineering (1R01EB023281, R21EB018907, R01EB019956, P41EB030006), the National Institute on Aging (R21AG082082, 1R01AG064027, R01AG016495, 1R01AG070988), the National Institute of Mental Health (UM1MH130981, R01 MH123195, R01 MH121885, 1RF1MH123195), the National Institute for Neurological Disorders and Stroke, (1U24NS135561-01, R01NS070963, 2R01NS083534, R01NS105820, R25NS125599), and was made possible by the resources provided by Shared Instrumentation Grants 1S10RR023401, 1S10RR019307, and 1S10RR023043. Much of the computation resources required for this research was performed on computational hardware generously provided by the Massachusetts Life Sciences Center (https://www.masslifesciences.com/). In addition, BF is an advisor to DeepHealth, a company whose medical pursuits focus on medical imaging and measurement technologies. BF's interests were reviewed and are managed by Massachusetts General Hospital and Partners HealthCare in accordance with their conflict of interest policies. Support for this research was provided in part by the National Institute of Diabetes and Digestive and Kidney Diseases (1-R21-DK-108277-01).

---

**Algorithm 1 Schwarz–Schur Involution.**
(Overall solve: numerical factorization + back substitution.)

---

**Require:** $\boldsymbol{\alpha}^{(*)} \in \mathbb{R}^{2^{k/2} \times 2^{k/2} \times 25 \times 25}$
**Require:** $\boldsymbol{\beta}^{(*)} \in \mathbb{R}^{2^{k/2} \times 2^{k/2} \times 25 \times 1}$
**Require:** $\boldsymbol{\chi}^{(k)} \in \mathbb{R}^{1 \times 1 \times b \times 1}$ only for Dirichlet condition.
($b := 2H + 2W - 4$ is the number of boundary pixels.)
The Schwarz forward step—Algorithm 2:

$$\boldsymbol{\alpha}^{(0)}, \boldsymbol{\beta}^{(0)} \leftarrow \boldsymbol{\alpha}^{(*)}, \boldsymbol{\beta}^{(*)}$$

**for** $i$ in range($k/2$): **do**
    The Schur forward step (horizontal)—Algorithm 4:

$$\boldsymbol{\alpha}^{(2i+1)}, \boldsymbol{\beta}^{(2i+1)} \leftarrow \boldsymbol{\alpha}^{(2i)}, \boldsymbol{\beta}^{(2i)}$$

    The Schur forward step (vertical):

$$\boldsymbol{\alpha}^{(2i+2)}, \boldsymbol{\beta}^{(2i+2)} \leftarrow \boldsymbol{\alpha}^{(2i+1)}, \boldsymbol{\beta}^{(2i+1)}$$

**end for**
**if** Dirichlet boundary condition **then**
    $\boldsymbol{\chi}^{(k)}$ is given

$$\boldsymbol{\chi}^{(k)} \leftarrow ...$$

**else**
    either use the improved implementation of Dirichlet or other non-disk topology boundary conditions (§C.2)

$$\boldsymbol{\chi}^{(k)} \leftarrow ...$$

    or the naïve slow approach: $\boldsymbol{\chi}^{(k)}$ is solved by matrix inversion using: $\boldsymbol{\alpha}^{(k)} \in \mathbb{R}^{1 \times 1 \times b \times b}, \boldsymbol{\beta}^{(k)} \in \mathbb{R}^{1 \times 1 \times b \times 1}$

$$\boldsymbol{\chi}^{(k)} \leftarrow (\boldsymbol{\alpha}^{(k)})^{-1} \boldsymbol{\beta}^{(k)}$$

**end if**
**for** $i$ in range($k/2, 0, -1$) **do**
    The Schur backward step (vertical):

$$\boldsymbol{\chi}^{(2i-1)} \leftarrow \boldsymbol{\chi}^{(2i)}$$

    The Schur backward step (horizontal)—Algorithm 5:

$$\boldsymbol{\chi}^{(2i-1)} \leftarrow \boldsymbol{\chi}^{(2i)}$$

**end for**
The Schwarz backward step—Algorithm 3:

$$\boldsymbol{\chi}^{(*)} \leftarrow \boldsymbol{\chi}^{(0)}$$

**return** $\boldsymbol{\chi}^{(*)} \in \mathbb{R}^{2^{k/2} \times 2^{k/2} \times 25 \times 25}$.

---

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

# Table of Contents

# A. Extended Discussions on Related Work

We focus on direct solvers—exact solutions that are computed up to numerical errors accumulated due to machine precisions, rather than iterative ones that only yield approximate solutions under a given error bound. The performance of direct solvers is more stable, whereas iterative solvers can greatly slow down for difficult $\mathbf{A}$ and may fail to converge if used without care. In contrast to direct solvers, iterative methods such as multigrid solvers only yield solutions within a prescribed error tolerance. Even so, they are still much slower than ours in scenarios that they are specifically designed for, and many orders of magnitude slower on challenging examples. Direct solvers are praised for their robustness and reliability, while being much slower. We implement a direct solver to achieve both accuracy/reliability and speed, combining the best of both worlds.

Despite a well-studied problem, general-purpose sparse direct solvers, introduced decades ago and having remained stable since, are not specifically designed for today's vision and learning applications. We focus on problems that fit on a single GPU, that nonetheless encompass common image sizes, avoiding considerations in cross-device communication that arise in larger scale problems in scientific computing. In addition, advances in GPUs sparked by deep learning shift the best practice towards algorithms exploiting parallelisms and that modern GPU BLAS kernels are extremely good at solving a large number of small problems. We observe that the high throughput offered by modern GPUs has been under-exploited in direct sparse solvers at the common scale of problems in vision.

**Sparse linear solvers in vision & graphics.** In computer vision and graphics, previous developments have focused on iterative solvers, and we are not aware of a prior effort to design *direct* solvers for Laplacian-like systems—a missing setting we address. A major component of contributions in optimization-based methods is to develop problem-specific iterative solvers. Our approach can bypass the need for iterative schemes in many tasks.

Hierarchical approaches or multigrid methods are popular especially when $\mathbf{A}$ arises from elliptic PDEs (Allaire, 2007). However, in situations where the solution is non-smooth, such as the Helmholtz equation, common preconditioned iterative solvers can work very poorly (Ernst & Gander, 2011). Incomplete LU factorizations are another family of preconditioning schemes. For positive semi-definite systems, incomplete Cholesky factorization can be applied (Chen et al., 2021a). Depending on the properties of $\mathbf{A}$, popular choices of iterative methods include Jacobi methods, Gauss–Seidel, Krylov subspace including (preconditioned) conjugate gradients (PCG) for positive-definite $\mathbf{A}$, and generalized minimal residual (GMRES) for un-symmetric $\mathbf{A}$ (Solomon, 2015).

Bolz et al. (2003) pioneer the application of GPUs to accelerate iterative solvers for computer graphics applications. Barron & Poole (2016) apply preconditioned conjugate gradients for fast bilateral filtering; their solvers operate in a lower dimensional bilateral space, unlike our method which aims for pixel-space solvers. Krishnan et al. (2013) use a Schur complement formula to construct a Laplacian preconditioning scheme. For very small-scale problems, it is well-known that dense direct solvers can be faster (Bro-Nielsen & Cotin, 1996). Problem-specific iterative solvers have been designed for image processing (Jeschke et al., 2009), physical simulation (Horvath & Geiger, 2009; Liu et al., 2016), and geometry processing (Krishnan et al., 2013). Direct solvers such as CHOLMOD have been used in differentiable rendering and inverse geometry design (Nicolet et al., 2021).

Li et al. (2023) apply a preconditioner learned from data, and Arisaka & Li (2023) develop learning-based acceleration of iterative methods under a meta-learning framework.

**Domain decomposition.** Our approach conceptually follows the divide-and-conquer strategy, dating back to the celebrated Schwarz alternating method in domain decomposition (Quarteroni & Valli, 1999). The Dirichlet-to-Neumann (DtN) operator is a standard object used to "glue" solutions between subdomains (Quarteroni & Valli, 1999). Typically, it is implicitly maintained through its matrix-vector product, realized by solving sparse systems on subdomains (Liu et al., 2016). In contrast, we make a different design choice to explicitly maintain the discrete DtNs as dense matrices.

**Sparse solvers: missing in differentiable programming.** Differentiable sparse linear or eigen solvers have been as popular requested feature in deep learning packages: please refer to §F.3 for a list of examples of community requests and discussions on sparse linear or eigen solvers. In a related but different effort, recent works (Grementieri & Galeone, 2022; Négiar et al., 2023; Vladymyrov et al., 2024) learn to solve linear systems, for which our method can serve as a strong baseline or an efficient training engine. Our analysis and discussion of FEM in §2 suggest that the learning-based PDE solvers and learning-based linear solvers are in fact attacking the *same* underlying problem—yet they are currently studied as separate fields with little cross-referencing.

# B. Visualization, Applications, and Experiments

Note that our goal in this section is *not* to compete on every task with state-of-the-arts, but to pick classic methods already requiring a solution to some Laplacian-like systems, collecting sparse matrices $\mathbf{A}$ covering a broad array of cases to test our methods with. Details on the timing are provided in §B.6.

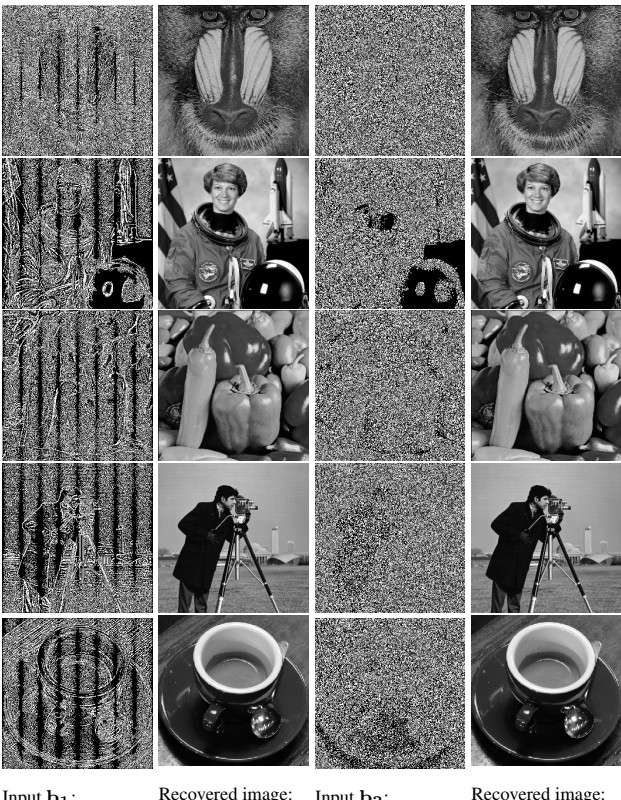

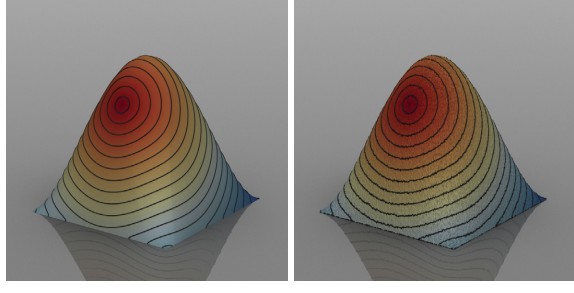

The kernel on a smooth surface.     The kernel on a noise surface.

*Figure 8.* Our solver accelerates by orders of magnitude the computation of geodesic distances. The heat kernel computed by our method is shown (in log scale). This is a task where iterative methods struggle: often the kernel solution must be within a tolerance of $10^{-10}$ for accurate geodesic distances (Crane et al., 2017).

Input $\mathbf{b_1}$:   Recovered image:   Input $\mathbf{b_2}$:   Recovered image:
convolved image.   $\mathbf{x_1} = \mathbf{A}_1^{-1}\mathbf{b_1}$.   convolved image.   $\mathbf{x_2} = \mathbf{A}_2^{-1}\mathbf{b_2}$.

*Figure 6.* Our solver is applied to image deconvolution with $\mathbf{A}$. We test with two cases of spatially varying kernels: $a_1^{(x,y)} = \sin(8\pi x)^2\mathbf{K}_1 + y^3\mathbf{K}_2$, $\mathbf{K}_1, \mathbf{K}_2 \in \mathbb{R}^{3\times3}$; $a_2^{(x,y)}$ is randomly drawn at $(x,y)$. where $\mathbf{K}_1, \mathbf{K}_2$ are defined in Figure 9.

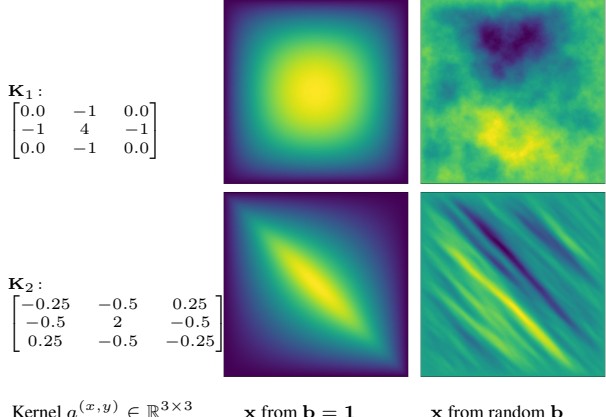

$\mathbf{K}_1$:
$$\begin{bmatrix} 0.0 & -1 & 0.0 \\ -1 & 4 & -1 \\ 0.0 & -1 & 0.0 \end{bmatrix}$$

$\mathbf{K}_2$:
$$\begin{bmatrix} -0.25 & -0.5 & 0.25 \\ -0.5 & 2 & -0.5 \\ 0.25 & -0.5 & -0.25 \end{bmatrix}$$

Kernel $a^{(x,y)} \in \mathbb{R}^{3\times3}$    $\mathbf{x}$ from $\mathbf{b} = \mathbf{1}$    $\mathbf{x}$ from random $\mathbf{b}$

*Figure 9.* Our solutions to isotropic and anisotropic Poisson. For fair comparison, 0s in $\mathbf{K}_1$ are treated as non-zero entries with value 0.0 in all solvers.

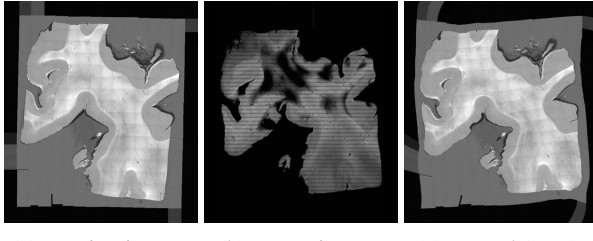

(a) Moving image.   (b) Target image.   (c) Moved (ours).

*Figure 7.* For computing diffeomorphisms and image registration, previous optimization-based methods can take a few minutes on high-res images. Switching the backend to our solver immediately reduces the runtime to a few seconds.

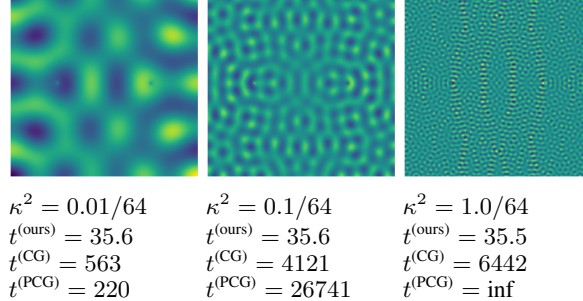

$\kappa^2 = 0.01/64$    $\kappa^2 = 0.1/64$    $\kappa^2 = 1.0/64$
$t^{\text{(ours)}} = 35.6$    $t^{\text{(ours)}} = 35.6$    $t^{\text{(ours)}} = 35.5$
$t^{\text{(CG)}} = 563$    $t^{\text{(CG)}} = 4121$    $t^{\text{(CG)}} = 6442$
$t^{\text{(PCG)}} = 220$    $t^{\text{(PCG)}} = 26741$    $t^{\text{(PCG)}} = \text{inf}$

*Figure 10.* The solutions to the Helmholtz equation under different wave number $\kappa$, computed using our method. This is the example used in Table 3. Unlike our method whose runtime is consistent, the performance of both CG and PCG (with AMG preconditioning as in §F) significantly degrades under higher frequency (larger $\kappa$) or even fail to converge (inf). Without a problem-specific preconditioner, PCG can be less efficient than CG.

### B.1. Newton's method and interactive graphics

As an immediate benefit, our fast solver potentially unlocks the capacity of Newton's method (Nocedal & Wright, 1999) relying on Hessian solves for applications in interactive physics, computer vision, and graphics.

When $\mathbf{A} = \mathbf{H}$ is the Hessian and $\mathbf{b} = \mathbf{g}$ is the gradient of some energy of a function on the image domain, the solution

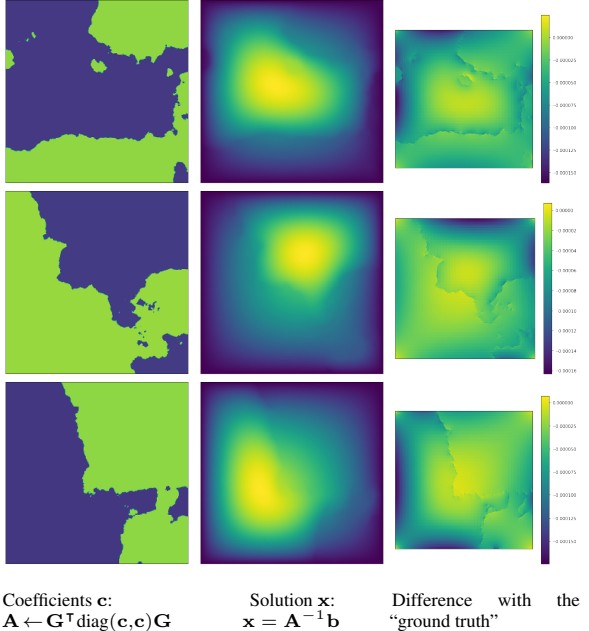

Coefficients **c**:
$\mathbf{A} \leftarrow \mathbf{G}^{\mathsf{T}} \mathrm{diag}(\mathbf{c},\mathbf{c}) \mathbf{G}$

Solution **x**:
$\mathbf{x} = \mathbf{A}^{-1} \mathbf{b}$

Difference with the "ground truth"

*Figure 11.* Our solvers realize the FEM coefficient-to-solution map.

$\mathbf{H}^{-1}\mathbf{g}$ yields the descent direction in Newton's method. It is very common that $\mathbf{H}$ and $\mathbf{g}$ come from an energy in which pixels interact locally with their neighbors, rather than all other pixels in the image. In this setting, the Hessian $\mathbf{H}$ has the same sparsity as the Laplacian matrix to apply our method; in fact, for some physics-based energy, $\mathbf{H}$ equals exactly to an anisotropic Laplacian, $\mathbf{H} \leftarrow \mathbf{L}$. In this case, the sparse linear solver is the sole bottleneck in applying Newton's method: constructing entries in $\mathbf{H}, \mathbf{g}$ involves only per-element/pixel computation, which can happen in parallel.

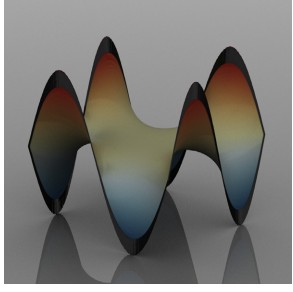

*Figure 12.* The minimal surface problem searches for the surface whose area is minimized under a prescribed boundary, having been a prior in, e.g., shape completion. The minimal surface computed by Newton's method, which is accelerated by $400\times$ with our Hessian solver compared to the SciPy backend.

**$400\times$ faster Hessian solvers for minimal surfaces.** As an application, in Figure 12, we apply our solver to Newton's method to compute the surface that minimizes the area with

a fixed boundary—the minimal surface problem. The surface is parameterized as a height field $z = u(x,y)$, and the goal is to find the one minimizing the total area, which is a nonlinear objective. Minimal surface area can be used as a prior in shape completion. Newton's method sequentially solves $\mathbf{H}^{-1}\mathbf{g}$. Thanks to the second-order convergence, using Newton's method with our solver as the backend, it converges within 20 iterations, while gradient descent using L-BFGS takes more than 1000 iterations to get to the same accuracy. Replacing SciPy in Newton's method with our solver immediately yields a speedup of $400\times$.

**Newton solver made efficient.** Despite superiority in the convergence rate, we find that Newton's method has very limited applications in computer vision. Perhaps it is partially because of the expensive Hessian solves, a barrier that our method removes.

This phenomenon is even more prominent in interactive computer graphics. In the seminal work Projective Dynamics (PD) (Bouaziz et al., 2014), as the central assumption when it comes to fast simulators, it is made explicit that Newton's method requires orders-of-magnitude fewer iterations while solving the Hessian in each iteration is expensive, making it less competitive.

The assumption that linear solve is expensive necessitates the development of alternative methods. A common pattern of the alternative fast methods is to reuse the factorization of the Laplacian matrix and run tens of thousands of such cheaper iterations, while Newton's method can converge in tens of iterations. This assumption also governs the best practice in differentiable simulation, e.g., in robot learning (Du et al., 2021).

Our method challenges this long-standing assumption, potentially unlocks the capacity of Newton's method, and eases the design of efficient algorithms.

### B.2. Physical simulation and shape optimization

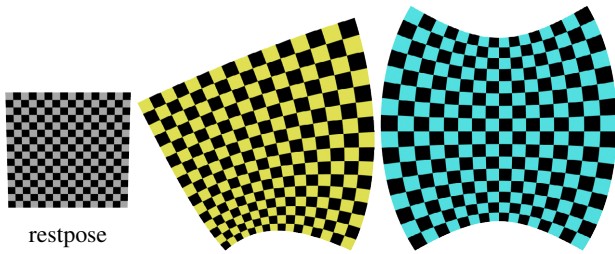

restpose

*Figure 13.* Some shape deformation produced by our method.

Physical simulation already can benefit from our method: Laplace equation governs physical phenomena in electromagnetism, gravity, heat diffusion, fluid dynamics, and shape deformation. Incorporating the coefficients $\mathbf{C}(\mathbf{x})$

allows pushing forward the PDEs in some irregular geometric domain onto a canonical domain, and accounts for the distortion induced by the geometric mapping—like how we compute distances on surfaces in §4. In addition, a straightforward way to utilize our solver is to write down some quadratic objective whose minimizer yields a linear system solve. The quadratic objective may come from the linearization of some nonlinear energy, such as the Hessian. Our solver currently only supports one variable per pixel; we plan to generalize it to multiple variables per pixel, e.g., two variables can represent the $x, y$ coordinates of a deformation. This can be done by generalizing the "DtN matrix" to be twice as large and viewing our method simply as a block Gaussian elimination. Instead, we present a simple trick to directly leverage our current solver in the setting of two variables per pixel. We put both the $x$-,$y$- components in a complex number $u_z = u_x + iu_y$. For instance, we can squeeze the $2n \times 2n$ block matrix arising in a conformal deformation energy (Mullen et al., 2008) into a single $n \times n$ complex-valued Hermitian matrix that $\mathbf{A} = \text{conj}(\mathbf{A}^\intercal)$. Without modifying the code, our solver applies to complex-valued linear systems. Figure 13 shows the deformation obtained in this way.

### B.3. Image matting and segmentation

In image matting (Grady et al., 2005; He et al., 2010; Levin et al., 2007), the matrix $\mathbf{A}$ encodes both an affinity matrix $\mathbf{L}$, a Laplacian with values decided by pixels' color, and a soft constraint term. It typically minimizes for an alpha function $\mathbf{x}$ that is smooth as measured by $\mathbf{L}$, and constrained to be 0 or 1 (stored in $\mathbf{k}$) in the foreground or background, respectively. $\mathbf{R}$ is a binary matrix selecting those constrained pixels.

$$\mathbf{x}^\intercal \mathbf{L}\mathbf{x} + \lambda \|\mathbf{R}^\intercal \mathbf{x} - \mathbf{k}\|^2 + \text{constant} \qquad (14)$$

In this case, $\mathbf{A} := \mathbf{L} + \lambda \mathbf{R}\mathbf{R}^\intercal, \mathbf{b} := \lambda \mathbf{R}\mathbf{k}$. Most of the change in $\mathbf{x}$ is restricted to the band between two layers of the user-provided masks (often called the trimap). We validate using the linear system obtained by PyMatting on the examples shown in Figure 14. We use the image matting framework PyMatting (Germer et al., 2020) and its implementation of the random walk Laplacian (Grady et al., 2005) with a small $3 \times 3$ stencil as $\mathbf{L}$. The error of our method in double precision is on the scale of $10^{-16}$, and is slightly smaller than SciPy in every image.

We choose to test it with our method for a particular reason: the difficulty of solving concentrates at the band, not evenly distributed across the image. The problem can be converted to an easy task for geometry-aware PDEs if solving only within the band region with Dirichlet condition at known pixels. But (14) represents a common way to incorporate the constraint softly: add a penalty term weighted by a very large coefficient $\lambda$. It makes some entries $\mathbf{A}_{ij}$ at constrained pixels very large compared to unconstrained pixels. Uncare-

ful direct solvers potentially follow a numerically unstable order when performing Gaussian elimination.

A purpose of this validation is to emphasize that viewed as a Gaussian elimination, our method cancels variables in an order that *is dependent* on the entries in $\mathbf{A}$, avoiding many numerical issues. A common misperception of our method is that it performs a Gaussian elimination that is blind to data $\mathbf{A}$. This is not the case. Our method does prescribes some block structures, which are independent of $\mathbf{A}$. However, within each block, the canceling order of nodes are still permuted, leading to robust numerical behaviors. This is a critical detail hidden in how to compute matrix inversions—we use "torch.linalg.inv" function which is backended by cuBLAS. Thus, when inverting a batch of dense matrices, it performs careful numerical schemes on each matrix individually to avoid usual pitfalls in numerical operations.

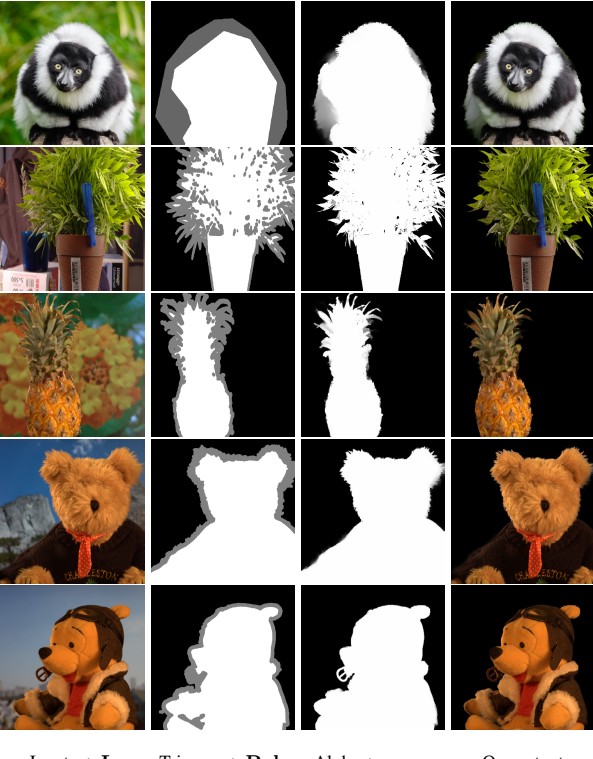

Input → $\mathbf{L}$    Trimap → $\mathbf{R}, \mathbf{k}$    Alpha ← our $\mathbf{x}$    Our output

*Figure 14.* The input image is used to generate an affinity matrix $\mathbf{A}$, the trimap for $\mathbf{A}, \mathbf{b}$, and the output alpha is $\mathbf{x} = \mathbf{A}^{-1}\mathbf{b}$.

### B.4. Fast backends for eigen solvers in spectral methods

For many practical applications, often it is the low-frequency components of the Laplacian that are relevant, rather than the high-frequency ones: $\mathbf{x} \leftarrow \mathbf{A}^{-1}\mathbf{x}$ magnifies the low-frequency components, while $\mathbf{x} \leftarrow \mathbf{A}\mathbf{x}$ does the opposite. This observation makes our method relevant for many com-

puter vision tasks.

In many vision applications, matrix $\mathbf{A}$ comes from pixel values and can be considered as a surrogate representation of an image, but it is not clear what constitutes a relevant $\mathbf{b}$. Then, the eigenvalue problem $\mathbf{A}\mathbf{x} = \lambda\mathbf{x}$ becomes the right computational model to extract information from $\mathbf{A}$.

We demonstrate how common eigen solvers can directly benefit from the fact that linear solvers can be made much more efficient than what people thought previously. Designing a standalone eigen solver is beyond the scope of this paper. The concurrent work (Wang, 2025) uses our efficient linear solver as the inverse iteration oracle to implement a fast eigen solver, reducing the design of fast eigen solvers to that of linear solvers. Our linear solver can serve as the atomic linear operator $\mathbf{x} \leftarrow \mathbf{A}^{-1}\mathbf{x}$ for iterative eigen solvers. For example, our solver can immediately speed up the spectral image segmentation method, normalized cuts (Shi & Malik, 2000). In this case, we choose the matrix $\mathbf{A}$ as $(\mathbf{I} - \hat{\mathbf{L}})$, where $\mathbf{L}$ is the matting Laplacian with the $3 \times 3$ kernel in §B.3 that becomes $\hat{\mathbf{L}}$ after having its rows and columns normalized, following Shi & Malik (2000).

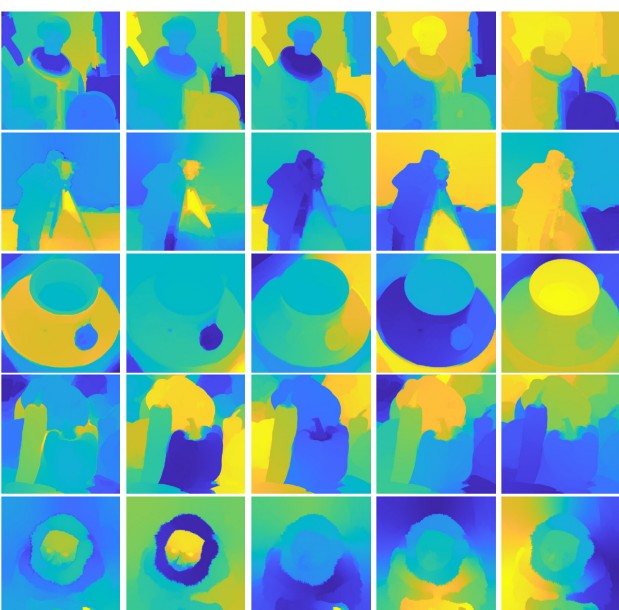

Figure 15. Our linear solver, when being used by a concurrent work to realize the inverse iteration oracle in iterative eigen solvers, is immediately transferred into an orders-of-magnitude faster eigen solver—455× faster than MATLAB's sparse eigen solver for $513 \times 513$ images, with applications to spectral image segmentation (Shi & Malik, 2000). The top 5 eigenfunctions of the astronaut, cameraman, coffee, pepper, and lemur images, computed by 20 calls to $\mathbf{A}^{-1}\mathbf{x}$, are shown.

For examples shown in Figure 15, iterative eigen solvers require only $13 \sim 20$ *calls* to $\mathbf{A}^{-1}\mathbf{x}$, to compute the top 6 eigenvectors within the tolerance of $10^{-3} \sim 10^{-2}$. For an image of size $513^2$, it takes 25 seconds to solve the top-6

eigenvalue problem for MATLAB, on which the code of Shi & Malik (2000) relies, while it takes as short as 55 milliseconds if using our linear solver, yielding an acceleration of **455×**. This is also many orders of magnitude faster compared to the Nyström methods that only approximate the spectrum (Vladymyrov & Carreira-Perpinan, 2016; Li et al., 2011; Belongie et al., 2002). This eigenvalue problem requires 1 step of numerical factorization (11 ms for our solver), and tens of the back substitution steps.

Recall that the back substitution can be much faster than the numerical factorization since it skips calculating any new left-hand sides. Thus, our linear solver can immediately accelerate by orders of magnitude the spectral segmentation in the *full* pixel space. This is different from previous methods like Barron & Poole (2016), where a major source of speedup comes from reducing the computation to a lower-dimensional space. Future work might combine our approach with the dimension-reduction strategy for further improvement, and leverage our solver as a parameter-free differentiable layer similarly to Barron & Poole (2016).

### B.5. PDEs on domains with a non-disk topology

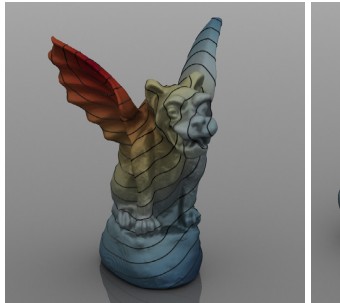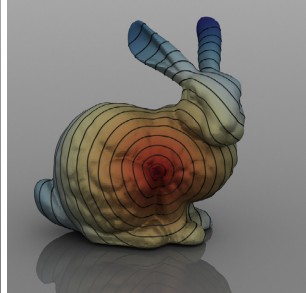

Figure 16. Our solver applies to shape analysis and geometry processing for curved surfaces as well. The heat kernel computed on two surfaces with the spherical topology are shown, accelerated by orders of magnitude using our method. Using a corner reflective boundary condition and an octahedral parameterization (Praun & Hoppe, 2003), our method can solve a geometric PDE that is defined on a surface with the spherical topology (§B.5).

We can solve PDEs on a surface with a non-disk topology, by leveraging the surface parameterization techniques, such as the octahedral parameterization (Praun & Hoppe, 2003) to cut and map a spherical topology surface onto an image domain. Our method can easily incorporate a midpoint reflective boundary condition induced by the octahedral parameterization (Praun & Hoppe, 2003), by modifying the last step to fill in the Dirichlet boundary condition in the Neumann solver ( §C.2). It allows us to process surfaces with spherical parameterization. Figure 16 shows the heat kernel computed on curved surfaces that have the spherical topology using our method.

Adding the ability to handle curved surfaces also makes our solver applicable to shape analysis and geometric deep learning, such as spectral methods for shape correspondence (Ovsjanikov et al., 2012; Litany et al., 2017; Yi et al., 2017), Laplacian eigenvalues for shape classification (Reuter et al., 2006), and many other applications (Solomon et al., 2014).

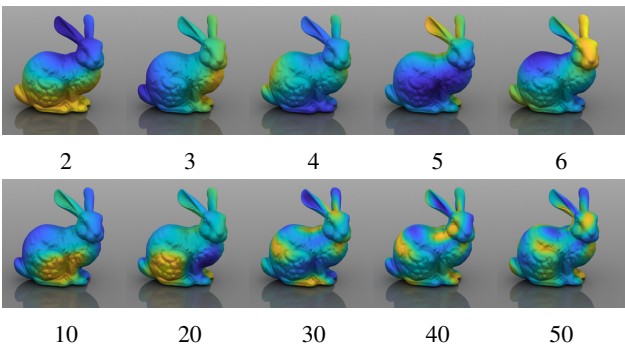

Figure 17. Our linear solver with the midpoint reflective boundary condition can be used as the reverse iteration for computing eigen functions of a surface with the the octahedral parameterization (Praun & Hoppe, 2003). Some Laplace-Beltrami eigen functions of the Stanford bunny are shown. Eigen functions of the Laplace-Beltrami operator are foundational for manifold data analysis and geometric deep learning.

## B.6. Timing details

We provide details on the timing comparison. Again, unlike the baseline solvers that are already highly optimized with official support from Nvidia, our method is simply prototyped in PyTorch with great potential for further acceleration. We have used torch.compile to reduce the overhead of PyTorch, which makes PyTorch code $1.2$ to $1.8\times$ faster. In addition to Table 2, Table 8 and 9 report the average runtime (in milliseconds) and the error tolerance to solve an isotropic or anisotropic system with the Dirichlet condition, respectively. The runtime of direct solvers including CUDA and our method are not affected by the anisotropic coefficients that make the problem much harder for iterative solvers as demonstrated in Table 2. SciPy does slow down for the anisotropic coefficients, but is still comparable to the isotropic case. In all experiments, we do not treat any matrix as symmetric even though they are.

Table 9 and 10 compare the average runtime when solving the Dirichlet or Neumann boundary condition. Since we reuse the Dirichlet solver to the Neumann problem, it can take slightly more time for the Neumann problem, but still comparable to the Dirichlet case. Future work may implement §C.2 for a better Neumann solver.

Surprisingly, the error tolerance of our method in float 64 is consistently smaller than that of SuperLU from SciPy

when solving elliptic PDEs, though our method does not optimize for accuracy like the SuperLU method does. We had suspected that the more careful pivoting in SuperLU might help to achieve a better precision, which nonetheless is not observed for solving Laplacian systems.

Table 11 reports the runtime when solving the matting system. Interestingly, SuperLU from SciPy significantly slows down in the matting example, due to the unbalanced concentration in spatial difficulty, as we have discussed in §B.3. However, our method (float 64) still has an error smaller than SciPy.

Table 4. The experiments same to the ones in Table 3 but for $1025 \times 1025$ images.

| | Lap 1e-2 | Lap 1e-4 | Hel 1e-2 | Hel 1.0 | R-Deconv | |
|---|---|---|---|---|---|---|
| Ours-64 | 175 | 175 | 175 | 175 | 175 | |
| Ours-32 | 36.4 | 36.3 | 36.5 | 36.4 | 36.4 | ×2 |
| SciPy-LU | 16001 | 15882 | 15899 | 16008 | 23175 | |
| CUDSS-LDU | 4566 | 4592 | 5095 | 4554 | 4687 | |
| Ours-64-back-sub | 3.1 | 3.1 | 3.1 | 3.1 | 3.1 | |
| (s) CG | 92.1 | 682 | 2280 | 21473 | NaN | |
| (s) MINRES | 113 | 1292 | 5037 | 33864 | NaN | |
| LSMR | 8244 | 1042012 | 749481 | 600585 | 1871 (*) | |
| LSQR | 4520 | NaN | NaN | 118307 (*) | 827 (*) | |
| biCGstab | 356 | 823 | NaN | NaN | NaN | |
| CGS | 95 | 538 | NaN | NaN | NaN | |
| DIOM | 2263 | 17549 | 39340 | 347352 | NaN | ×k |
| FOM | 5934 | 866361 | 667058 | NaN | NaN | |
| QMR | 147 | 4268 | 2444 | NaN | NaN | |
| BILQ | 150 | 4228 | 2694 | NaN | NaN | |
| GMRES | 1028 | 279031 | 285348 | NaN | NaN | |
| DQGMRES | 279 | 13018 | 5172 | 49156 | NaN | |
| FGMRES | 1051 | 40096 | NaN | NaN | NaN | |

Table 5. The error as measured by relative tolerance corresponding to Table 4. Note that random deconvolution (R-Deconv) may not be a very well-defined task to evaluate accuracy, since the matrix **A** can be close to singular due to randomness.

| Error | Lap 1e-2 | Lap 1e-4 | Hel 1e-2 | Hel 1.0 | R-Deconv |
|---|---|---|---|---|---|
| Ours-32 | 1.04E-05 | 1.21E-05 | 5.33E-03 | 3.57E-04 | NaN |
| Ours-64 | 1.89E-14 | 2.26E-14 | 1.37E-11 | 8.83E-13 | 1.89E-09 |
| SciPy-LU | 2.75E-14 | 2.82E-14 | 2.16E-13 | 4.32E-13 | 1.78E-15 |
| CUDSS-LDU | 2.03E-14 | 2.05E-14 | 7.37E-14 | 9.86E-13 | 5.41E-13 |

Table 6. The error as measured by relative tolerance corresponding to Table 3. The back-substitution variant of our solver shares the same error with (the full solve variant of) our method.

| Error | Lap 1e-2 | Lap 1e-4 | Hel 1e-2 | Hel 1.0 | R-Deconv |
|---|---|---|---|---|---|
| Ours-32 | 1.52E-05 | 1.71E-05 | 1.94E-04 | 2.30E-03 | NaN |
| Ours-64 | 2.71E-14 | 3.03E-14 | 5.45E-13 | 4.30E-12 | 3.39E-08 |
| SciPy-LU | 4.06E-14 | 4.19E-14 | 7.22E-13 | 1.03E-12 | 2.06E-15 |
| CUDSS-LDU | 2.97E-14 | 3.04E-14 | 1.56E-13 | 2.86E-12 | 6.24E-12 |

Table 7. The back substitution time for our method to solve a Neumann problem. Surprisingly, the speed for a $1025^2$ image is similar to that of a $257^2$ image, suggesting that there are probably a large room for low-level optimization for the scale of $257^2$.

| | Resolution | Time (milliseconds) |
|---|---|---|
| Ours (float-64) | 257 | 2.56 |
| Ours (float-64) | 513 | 3.01 |
| Ours (float-64) | 1025 | 3.13 |

*Table 8.* Average runtime (in milliseconds) to solve an isotropic Laplacian with Dirichlet condition.

| | time | | | | error tolerance | | |
|---|---|---|---|---|---|---|---|
| Example | CUDA | SciPy | ours | ours-64 | SciPy | ours-64 | ours |
| $2049^2$ | 21297 | 138219 | 158.5 | OOM | 1.92e-14 | 1.07e-14 | 6.86e-6 |
| $1025^2$ | 4663 | 15526 | 37.4 | 172.9 | 1.27e-14 | 7.56e-15 | 5.28e-6 |
| $513^2$ | 1053 | 2223 | 10.9 | 31.9 | 8.52e-15 | 5.43e-15 | 3.87e-6 |
| $257^2$ | 235 | 334 | 5.11 | 7.55 | 5.49e-15 | 3.79e-15 | 2.53e-6 |

*Table 9.* Average runtime (in milliseconds) to solve an anisotropic system with Dirichlet condition.

| | time | | | | error tolerance | | |
|---|---|---|---|---|---|---|---|
| Example | CUDA | SciPy | ours | ours-64 | SciPy | ours-64 | ours |
| $2049^2$ | 21354 | 143100 | 159.3 | OOM | 3.43e-14 | 2.03e-14 | 1.07e-5 |
| $1025^2$ | 4710 | 16512 | 37.5 | 171.3 | 2.54e-14 | 1.57e-14 | 7.97e-6 |
| $513^2$ | 1036 | 2051 | 10.9 | 31.77 | 1.36e-14 | 8.85e-15 | 4.44e-6 |
| $257^2$ | 234 | 355 | 5.82 | 7.43 | 2.24e-16 | 1.57e-16 | 2.45e-6 |

*Table 10.* Average runtime (in milliseconds) to solve an anisotropic system with Neumann condition.

| | time | | | | error tolerance | | |
|---|---|---|---|---|---|---|---|
| Example | CUDA | SciPy | ours | ours-64 | SciPy | ours-64 | ours |
| $2049^2$ | 21396 | 186231 | 171.6 | OOM | 2.41e-15 | 1.47e-15 | 6.73e-7 |
| $1025^2$ | 4779 | 17511 | 42.2 | 187.4 | 2.40e-15 | 1.47e-15 | 6.74e-7 |
| $513^2$ | 1038 | 2174 | 12.2 | 35.4 | 2.37e-15 | 1.47e-15 | 6.61e-7 |
| $257^2$ | 235 | 389 | 6.47 | 8.82 | 2.28e-15 | 1.47e-15 | 6.56e-7 |

*Table 11.* Average runtime (in milliseconds) to solve the matting system with Neumann condition.

| | time | | | error tolerance | |
|---|---|---|---|---|---|
| Example | CUDA | SciPy | ours-64 | SciPy | ours-64 |
| $2049^2$ | 22601 | 244125 | OOM | 2.89e-16 | 1.80e-16 |
| $1025^2$ | 4721 | 16969 | 187.6 | 2.73e-16 | 1.77e-16 |
| $513^2$ | 1046 | 6223 | 35.8 | 2.50e-16 | 1.65e-16 |
| $257^2$ | 232 | 1045 | 8.79 | 2.24e-16 | 1.57e-16 |

## B.7. Numerical stability

Our method prescribes a structure of pivoting and elimination ordering for embarrassingly parallel elimination, which means that the row- and column-wise pivoting is limited to switching rows and columns corresponding to nodes within the same sub-domain. Theoretically, such restriction on the pivoting scope could compromise the numerical stability for some problems. However, we consistently observe that when solving Laplacian systems or elliptic PDEs, our method even has a smaller error consistently than that of SuperLU or cuDSS (METIS ordering), which employ more conservative pivoting strategies. These should be related to the fact that the sub-systems that our method explicitly maintains are discrete Dirichlet-to-Neumann operators which are

well-defined objects in some sense. It will be interesting for future work to study the numerical behavior of our method theoretically. For Helmholtz equations that are not elliptic, our errors in float-64 are still comparable to the baseline methods. For the deconvolution with a random kernel, our method can have a larger error; it can fail if using float-32, but note that this is not a well-defined task, since under random spatial kernels with large probability the sub-system in some patch can be quite close to singular to make the global system **A** also close to singular, so it is expected that SuperLU with global pivoting will be the most numerical safe option. Still, in numerically challenging situations, our method with float-64 can offer accuracy that is sufficient for deep learning systems.

The row-wise and column-wise pivoting in the LU factorization is meant to improve the numerical stability of the right solve, not for the left solve $\mathbf{c}^\intercal \mathbf{A}^{-1}$, or equivalently, the transposed solve $\mathbf{A}^{-\intercal}\mathbf{c}$, for the back propagation step. Our Dirichlet-to-Neumann factorization can be applied directly to both the left and right solves, without the need of a separate factorization for $\mathbf{A}^\intercal$.

### B.8. Complexity analysis

To emphasize, unlike the standard numerical analysis, in our setting, we are less concerned about the asymptotic complexity of the numerical algorithm, but rather the actual runtime on common image sizes. These are very small-scale problems by the standard of scientific computing, so having a small constant in the complexity can be equally or even more important than the asymptotic rate. The constant and thus the best algorithm are hardware dependent, so the actual runtime as we have reported in the paper should be the only criterion, in addition to which we still provide an asymptotic complexity analysis.

The time complexity of our algorithm is dominated by the parallel matrix inversion in the Schur steps:

$$\sum_{j=1}^{\log n} ((2^j)^{1/2})^3 = n^{3/2} + \frac{1}{2^{1.5}} n^{3/2} + \left(\frac{1}{2^{1.5}}\right)^2 n^{3/2} + \cdots$$
$$= \mathcal{O}(n^{1.5})$$

(15)

where we have assumed the cubic cost of inverting dense matrices and that the parallelized inversions take the same time as inverting one matrix of the same size.

## C. Method: Extended Discussions

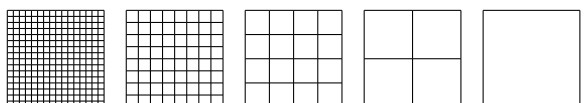

*Figure 18.* After the Schwarz step that eliminates interior nodes/pixels for each of the subdomains, the domain becomes a "wire-frame" that is progressively simplified in the Schur steps. The Schur step recursively collapses adjacent hollow subdomains by applying a Schur formula to merge two DtN matrices.

**Condense sparsity and restrict the number of variables.** For a $10^3 \times 10^3$ image, its discrete Laplacian $\mathbf{A}$ is a sparse matrix of $10^6 \times 10^6$, which is much too large to deal with as a dense matrix. If we can instead somehow work with one (dense) $10^3 \times 10^3$ matrix and a few smaller ones, then we can leverage the efficient dense linear algebraic kernels BLAS (Dongarra et al., 1988; 2018). Indeed, the largest dense matrix we have to invert is at the scale of $10^3 \times 10^3$. Our algorithm is a procedure to assemble the final

solution from many small dense matrices are very efficiently inverted.

In summary, our "Schwarz–Schur involution" recursively merges subdomains and eliminates nodes/pixels at the interface between subdomains, and updates the maintained system matrix by applying a Schur complement formula.

Following the nested dissection hierarchy (George, 1973), we recursively divide the domain into two subdomains and reduce the degrees of freedom: nodes at the interface between subdomains have duplicated copies in each subdomain. This yields a quad-tree in 2D: as shown in Figure 18, recursively applying this step reduces the number of subdomains following the sequence: $(512, 512) \rightarrow (256, 256) \rightarrow (128, 128) \rightarrow (64, 64)... \rightarrow (1, 1)$. This procedure visually corresponds to Figure 18 but in the reverse order.

### C.1. Dirichlet-to-Neumann factorization

We call the Schwarz and Schur steps as the *Dirichlet-to-Neumann factorization*, a numerical factorization procedure similar to the LU factorization. The role of the dense matrices we saved in $\boldsymbol{\alpha}^{(j)}$ and the overall hierarchy is analogous to the L and U factors in standard LU factorization (Davis, 2006). In the Schur step, small patches are recursively merged, during which the new left-hand side—the DtN matrices for the merged subdomains are constructed by inverting sub-block of the DtN matrices $\mathbf{P}, \mathbf{Q}$ at subdomains using the equations in §3.2.2; every matrix there will be saved for later usage—in the back substitution stage to apply to the right-hand side.

The output in the last Schur step is the tensor $\boldsymbol{\alpha}^{(k)}$ of size $(1, 1, b, b)$, where again $b$ is the number of pixels on the border of the image domain. Denoting $\boldsymbol{\alpha}^{(k)}[0, 0, :, :]$ as the dense matrix $\mathbf{D} \in \mathbb{R}^{b \times b}$, the original linear system $\mathbf{Ax} = \mathbf{b}$ at the last level of the hierarchy becomes $\mathbf{Du} = \mathbf{d}$, where $\mathbf{D} \in \mathbb{R}^{b \times b}$ and $\mathbf{d} \in \mathbb{R}^{b \times 1}$.

**Back substitution** is the standard step to fill in solutions for all rows of $\mathbf{x}$. For Dirichlet boundary condition, we directly fill in rows that correspond to boundary pixels, $\mathbf{x}|_{1:b} \leftarrow \mathbf{g}$. Then values at the interface can be recursively backed filled by solving (11), so the values in $\mathbf{x}$ at the wire-frame of the image domains can be obtained. Finally, we recover the values in $\mathbf{x}$ that correspond to the interior $3 \times 3$ block of each patch.

### C.2. Solvers for Neumann boundary condition

For Neumann boundary condition, there are multiple ways to directly apply our Dirichlet solvers: 1) solve some DtN system for the final domain to fill in the missing Dirichlet data and call the Dirichlet solver (§C.2.1); 2) the actual option we use in the paper for better performance: modify the first Schwarz step to also eliminate pixels at the boundary

of the domain (§C.2.3).

We use the Dirichlet boundary value problem to explain our method as it arises naturally: our method is a hierarchical construction of DtN matrices that are directly applicable under the known Dirichlet condition. The linear condition (11) that we enforced at the interface says that the Dirichlet data should be chosen so that the resulting Neumann data on either side of the interface must match each other—flux into the interface should equal to flux outward on the other side of the interface.

Most tasks in computer vision have the natural (Neumann) boundary condition. The Neumann boundary value problem can be solved in a similar fashion. In principle, one can recursively work with the *Neumann-to-Dirichlet* matrix instead of the Dirichlet-to-Neumann matrix, which are (pseudo) inverses of each other. Then we can fill in the missing Neumann boundary condition in a hierarchical fashion. Instead, we present a simple solution that directly leverages the Dirichlet solver. The goal is to minimize effort and reuse Dirichlet solvers.

### C.2.1. SOLUTION 1

The original linear system $\mathbf{A}\mathbf{x} = \mathbf{b}$ at the last level of the hierarchy becomes $\mathbf{D}\mathbf{u} = \mathbf{d}$, where $\mathbf{D} \in \mathbb{R}^{b \times b}$ and $\mathbf{d} \in \mathbb{R}^{b \times 1}$. From the Gaussian elimination viewpoint, the DtN matrix $\mathbf{D}$ is nothing more than the linear system matrix with the interior nodes in the image eliminated. Thus, the Dirichlet boundary condition that is missing in order to apply the Dirichlet solver for the Neumann problem can be simply found by solving $\mathbf{D}^{-1}\mathbf{d}$.

Indeed, this immediately gives a Neumann solver. However, there is one drawback with this strategy of reusing the Dirichlet solver in a Neumann problem. The system matrix $\mathbf{D}$ is $16\times$ as large as the largest matrix that one has to solve in the Dirichlet problem. ($\mathbf{D}$ is $4\times$ as large in rows and columns so a total of 16). This makes the Neumann solver for large-scale problem noticeably slower than the Dirichlet solver, even though in principle they should be equally difficult.

### C.2.2. SOLUTION 2

As shown in Figure 19, to avoid the need to invert a larger matrix, we design a pre-processing step to further eliminate nodes at the beginning.

In general, it is better to eliminate nodes as early as possible, since that can happen in parallel, rather than deferring it to the final stage to increase the size of the large dense matrix. In some sense, the Neumann boundary value problem is easier than the Dirichlet counterpart, since the former allows node elimination at an earlier stage.

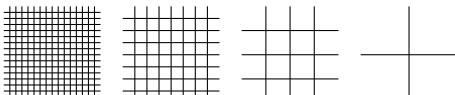

Figure 19. The wire-frame structure resulted in the Neumann elimination, obtained by modifying the Dirichlet elimination in Figure 18.

### C.2.3. SOLUTION 3

While Solution 2 should be the theoretically optimal way in this paper, it requires implementing a new edge collapse procedure and handling boundary patches differently. In

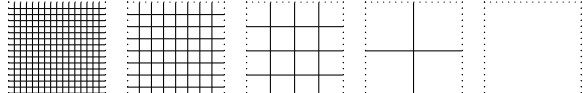

Figure 20. The final hierarchy for Neumann solver.

order to reuse the Dirichlet solver like Solution 1 does, we propose a further simplification: at the boundary of the image domain we only eliminate wire-frames without two end points. The nodes at the boundary of the image domain are still kept, though any rows and columns corresponding to them in the left-hand side are zeros, and any rows corresponding to them in the right-hand side are also zeros. Then, we can reuse the same equations in §3.2.2. This solution has the advantage that at the last level, the system to solve will be smaller than $\mathbf{D}$: while $\mathbf{D}$ has the same size, we know that most rows and cols in $\mathbf{D}$ are zeros and can be thrown away. So we only need to solve for, e.g., $\mathbf{D}^{-1}_{1:4:b,1:4,b}\mathbf{d}_{1:4:b}$ when patch size is $4 \times 4$ enlarged to $5 \times 5$ so $(\cdot)_{1:4:b}$ selects one from every 4 pixels along the border. The new system matrix $\mathbf{D}^{-1}_{1:4:b,1:4,b} \in \mathbb{R}^{b/4 \times b/4}$ records the interplay among the scatter points on the border of the image domain.

### C.3. Differentiable linear solvers and derivatives

Differentiable linear solvers are useful in many scientific applications (Hovland & Hückelheim, 2024). Following notations of Wang & Solomon (2021), let us derive the closed-form formulas for the partial derivatives $\mathbf{x} = \mathbf{A}^{-1}\mathbf{b}$: we need to obtain $\partial \mathbf{x}/\partial \mathbf{b}$ and $\partial \mathbf{x}/\partial \mathbf{A}$.

It is convenient to introduce the notation to flatten the matrix $\mathbf{A}$ into a vector:

$$\mathbf{a} := \mathrm{FLATTEN}(\mathbf{A}) \in \mathbb{R}^{n^{(nz)}} \tag{16}$$

which stores the the nonzero entries of $\mathbf{A}$ in a vector $\mathbf{a}$, where $n^{(nz)}$ is the number of nonzero entries, such that the matrix product

$$\mathbf{A} = \mathbf{J}_1^\mathsf{T} \mathrm{DIAG}(\mathbf{a}) \mathbf{J}_2 \tag{17}$$

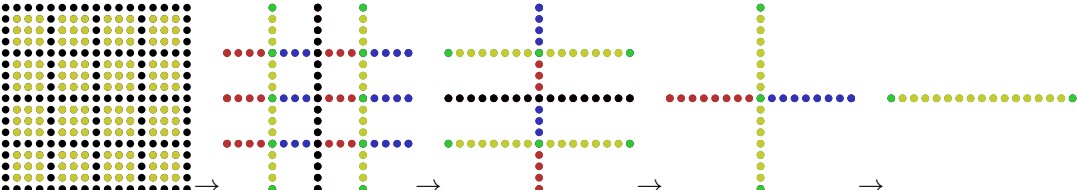

*Figure 21.* The node elimination order corresponds to Figure 19. An ideal elimination order for the Neumann boundary conditions.

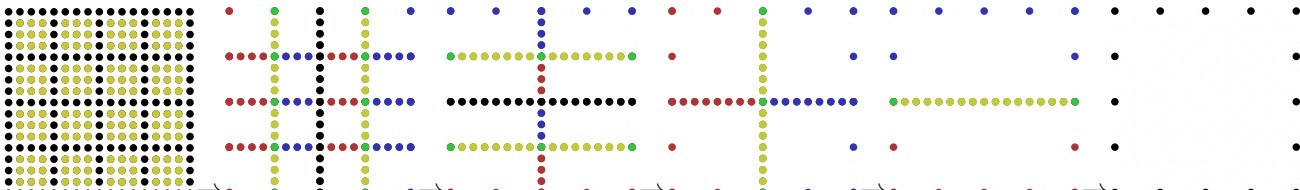

*Figure 22.* The node elimination order corresponds to Figure 20. The elimination order we actually use for the Neumann boundary conditions for the ease of implementation.

recovers the matrix $\mathbf{A}$ in the coordinate format, where $\mathbf{J}_1, \mathbf{J}_2 \in \mathbb{R}^{n^{(nz)} \times n}$. Here $\mathrm{DIAG}(\cdot)$ converts a vector into a diagonal matrix and notice that

$$\mathrm{DIAG}(\mathbf{u})\mathbf{v} = \mathrm{DIAG}(\mathbf{v})\mathbf{u} = \mathbf{u} \odot \mathbf{v} \qquad (18)$$

for two vectors $\mathbf{u}, \mathbf{v}$, where $\mathbf{u} \odot \mathbf{v}$ is the element-wise multiplication of two vectors. Namely, their $i$-th rows, $\mathbf{J}_1(i,:)$ and $\mathbf{J}_2(i,:)$, are one-hot sparse vectors that put the entry $\mathbf{a}_i$ in the correct row and column in the matrix.

Then, any infinitesimal deviation $(\delta\mathbf{A}, \delta\mathbf{b}, \delta\mathbf{x})$ from the stationary $(\mathbf{A}, \mathbf{b}, \mathbf{x})$ such that $\mathbf{A}\mathbf{x} = \mathbf{b}$ must satisfy that:

$$(\delta\mathbf{A})\mathbf{x} + \mathbf{A}\delta\mathbf{x} = \delta\mathbf{b}$$
$$\mathbf{J}_1^\mathsf{T}\mathrm{DIAG}(\delta\mathbf{a})\mathbf{J}_2\mathbf{x} + \mathbf{J}_1^\mathsf{T}\mathrm{DIAG}(\mathbf{a})\mathbf{J}_2\delta\mathbf{x} = \delta\mathbf{b}$$
$$\mathbf{J}_1^\mathsf{T}\mathrm{DIAG}(\mathbf{J}_2\mathbf{x})\delta\mathbf{a} + \mathbf{A}\delta\mathbf{x} = \delta\mathbf{b}$$

So we have:

$$\partial\mathbf{x}/\partial\mathbf{a} = -\mathbf{A}^{-1}\mathbf{J}_1^\mathsf{T}\mathrm{DIAG}(\mathbf{J}_2\mathbf{x}) \qquad (19)$$
$$\partial\mathbf{x}/\partial\mathbf{b} = \mathbf{A}^{-1} \qquad (20)$$

Thus, provided with $\nabla_\mathbf{x}E$, the gradient of the energy $E(\mathbf{x})$ w.r.t. $\mathbf{x}$, the chain rule gives the gradients $\nabla_\mathbf{a}E$ and $\nabla_\mathbf{b}E$.

$$\nabla_\mathbf{a}E = -(\mathbf{J}_2\mathbf{x}) \odot \left(\mathbf{J}_1(\mathbf{A}^\mathsf{T})^{-1}\nabla_\mathbf{x}E\right) \qquad (21)$$
$$\nabla_\mathbf{b}E = (\mathbf{A}^\mathsf{T})^{-1}\nabla_\mathbf{x}E \qquad (22)$$

Remark: to evaluate the gradients, we need to solve the system $\mathbf{A}^\mathsf{T}$ instead of $\mathbf{A}$ in general, unless $\mathbf{A}$ is symmetric. Our method is an exactly "symmetric" algorithm for $\mathbf{A}$ and $\mathbf{A}^\mathsf{T}$: For our method, it is not necessary to run the Dirichlet-to-Neumann factorization one more time for $\mathbf{A}^\mathsf{T}$, since our

algorithm already gathers all ingredients to simply apply the left multiplication $\mathbf{c}^\mathsf{T}\mathbf{A}^{-1}$ in lieu of $\mathbf{A}^{-\mathsf{T}}\mathbf{c}$. This can lead to an extra $2\times$ speedup of our method compared to other direct solvers, when differentiating linear solvers.

### C.4. Transposed solve with reused Schur complement

Schur complement reduces solving the $2 \times 2$ block matrix

$$\begin{bmatrix} \mathbf{X} & \mathbf{Y} \\ \mathbf{Z} & \mathbf{W} \end{bmatrix} \begin{bmatrix} \mathbf{x} \\ \mathbf{z} \end{bmatrix} = \begin{bmatrix} \mathbf{y} \\ \mathbf{w} \end{bmatrix} \qquad (23)$$

to the solution of a smaller matrix $\mathbf{D}\mathbf{x} = \mathbf{d}$, with the formula to update the left-hand side (LHS):

$$\mathbf{D} := \left(\mathbf{X} - \mathbf{Y}\mathbf{W}^{-1}\mathbf{Z}\right) \quad \text{(LHS update)}, \qquad (24)$$

the formula to update the right-hand side (RHS):

$$\mathbf{d} := \mathbf{y} - \mathbf{Y}\mathbf{W}^{-1}\mathbf{w} \quad \text{(RHS update)}, \qquad (25)$$

and the formula for back substitution (after receiving $\mathbf{x}$ calculated from the coarser level):

$$\mathbf{z} = \mathbf{W}^{-1}[\mathbf{w} - \mathbf{Z}\mathbf{x}] \quad \text{(back-fill)}. \qquad (26)$$

Now let us consider the transposed solve:

$$\begin{bmatrix} \mathbf{X} & \mathbf{Y} \\ \mathbf{Z} & \mathbf{W} \end{bmatrix}^\mathsf{T} \begin{bmatrix} \mathbf{x} \\ \mathbf{z} \end{bmatrix} = \begin{bmatrix} \mathbf{y} \\ \mathbf{w} \end{bmatrix} \qquad (27)$$

or

$$\begin{bmatrix} \mathbf{X}^\mathsf{T} & \mathbf{Z}^\mathsf{T} \\ \mathbf{Y}^\mathsf{T} & \mathbf{W}^\mathsf{T} \end{bmatrix} \begin{bmatrix} \mathbf{x} \\ \mathbf{z} \end{bmatrix} = \begin{bmatrix} \mathbf{y} \\ \mathbf{w} \end{bmatrix} \qquad (28)$$

Instead of directly calculating the solution using $\mathbf{x} = \tilde{\mathbf{D}}^{-1}\tilde{\mathbf{d}}$

$$\tilde{\mathbf{D}} := \left(\mathbf{X}^{\mathsf{T}} - \mathbf{Z}^{\mathsf{T}}\mathbf{W}^{-\mathsf{T}}\mathbf{Y}^{\mathsf{T}}\right) \equiv \mathbf{D}^{\mathsf{T}} \quad \tilde{\mathbf{d}} := \mathbf{y} - \mathbf{Z}^{\mathsf{T}}\mathbf{W}^{-\mathsf{T}}\mathbf{w}, \tag{29}$$

we would like to reuse $\mathbf{W}, \mathbf{D}$—the updated left-hand side for the untransposed system. Though the formula $\mathbf{D}^{-\mathsf{T}} \equiv (\mathbf{D}^{-1})^{\mathsf{T}}$ allows avoiding the second matrix inversion with a cheaper matrix transpose, we adopt an even more efficient approach that avoids transposing the dense system matrix $\mathbf{W}, \mathbf{D}$.

$$\mathbf{x} = \left[(\mathbf{y}^{\mathsf{T}} - \mathbf{w}^{\mathsf{T}}\mathbf{W}^{-1}\mathbf{Z})\mathbf{D}^{-1}\right]^{\mathsf{T}} \quad \text{(RHS update: left)} \tag{30}$$

$$\mathbf{z} = \left[(\mathbf{w}^{\mathsf{T}} - \mathbf{x}^{\mathsf{T}}\mathbf{Y})\mathbf{W}^{-1}\right]^{\mathsf{T}} \quad \text{(back-fill: left).} \tag{31}$$

These left solves are straightforward to apply since we maintain $\mathbf{W}^{-1}, \mathbf{D}^{-1}$ as dense matrices.

In summary, in the final algorithm to solve the transposed system $\mathbf{A}^{-\mathsf{T}}\mathbf{b}$, we simply replace the updating formulas (25),(26) with (30),(31), whenever a Schur complement elimination step is applied.

### C.5. Two settings in linear solvers

We clarify that there are two different settings when solving (multiple instances of) $\mathbf{x} = \mathbf{A}^{-1}\mathbf{b}$:

- $\mathbf{A}$ is unchanged and fixed. This is when the common trick can be applied, e.g. in interactive computer graphics (Bouaziz et al., 2014; Du et al., 2021): pre-factorize $\mathbf{A} = \mathbf{B}^{\mathsf{T}}\mathbf{B}$ once and at runtime reuse the fixed factorization for speedup, only performing back substitution.
- $\mathbf{A}$ is changed over iterations, solving systems with coefficients unknown in advance. This is a more common situation for nonlinear systems. For example, in Newton's method, $\mathbf{A}$ is the Hessian that changes over iterations.

In this paper, we focus on the second setup, where the system must be solved from scratch and the numerical factorization stage—which is the main bottleneck—is therefore included. Nonetheless, both settings can benefit from our method significantly.

### C.6. Involution: exact inverse convolution (spatially varying kernel)

Many classical algorithms can be viewed as a single step of Schwarz–Schur involution—solving a linear Laplacian-like system. Image filtering (Barron & Poole, 2016), editing (Pérez et al., 2003), matting (Germer et al., 2020), and segmentation (Shi & Malik, 2000) are classical examples. The linear solving in these methods can be viewed as the

generalized deconvolution with a spatially varying kernel, but they are usually not called a deconvolution task. To distinguish from the common setting of deconvolution, we term our process Schwarz–Schur involution (Sinv2D), the exact inversion of the convolution with a spatial kernel. We call the set of dense matrices that are applied across all subdomains as the *Schwarz–Schur involution kernel* at that level, which are stored in the tensor $\boldsymbol{\alpha}^{(j)}$.

Theoretically, our method can enjoy further improvements when the kernel $a^{(x,y)}$ is a spatially constant matrix (the timing reported in the paper does not take advantage of this: we always treat $a^{(x,y)}$ as spatially varying even if it is constant for fair comparisons). In this case, at each subdivision level, the S-involution kernels are the same across the domain. Thus, there is no need to maintain a dense matrix for each subdomain individually; instead, we can maintain one S-involution kernel that applies to every subdomain.

What is even better: for a prescribed kernel, the S-involution kernel has a closed-form formula that can be derived and calculated ahead of time, bypassing the need of numerical factorization. When $\mathbf{A}$ is the isotropic Laplacian, the resulting DtN matrix approximates a continuous DtN operator which has a known low-rank approximation, the structure that may be leveraged in future work (Bebendorf, 2008).

### C.7. Inverse problems, optimal control of PDEs, and PDE-constrained optimization

Our method accelerates not only the forward linear solver, but also the "backward" process of differentiating into the linear solver, which is useful in, e.g. inverse problems, optimal control of PDEs, and PDE-constrained optimization.

For $\mathbf{x} = \mathbf{A}^{-1}\mathbf{b}$, back-propagating the gradient w.r.t. $\mathbf{x}$ to that w.r.t. $\mathbf{b}$ is straightforward since $\delta\mathbf{x} = \mathbf{A}^{-1}\delta\mathbf{b}$, so we will focus on how to differentiate into the coefficients of $\mathbf{A}$. There are two ways:

- In fact, our method implemented in PyTorch, is already end-to-end differentiable w.r.t. entries of $\mathbf{A}$. However, it is generally advised against differentiating through the low-level implementation details: see Hovland & Hückelheim (2024) for empirical studies and references therein for discussions. We also observe that indeed this naïve approach is inefficient to implement the backward step to obtain gradients.
- Alternatively, as described in §C.3, without relying on the automatic differentiation feature of PyTorch, we can explicitly calculate the gradient by solving the adjoint system $\mathbf{A}^{-\mathsf{T}}\mathbf{c}$, where $\mathbf{c}$ depends on $\mathbf{x}$, similarly to the differentiable bilateral solver (Barron & Poole, 2016). When $\mathbf{A}$ is symmetric, this involves solving a linear system with the same left-hand side, see, e.g., Wang et al. (2023).

Thus, our solver immediately speeds up the optimization of some distributed parameters $\mathbf{A}$, which—written in the equivalent form $a^{(x,y)}$—can represent a convolutional kernel, local material property, or the Jacobian of a deformation field. In the literature of inverse PDEs, this is often referred to as the problem of distributed parameter identification.

In §5, we apply the differentiable solver to search for the $\mathbf{A}$ that minimizes an objective $E(\cdot)$. This is an inverse problem

$$\min_{\mathbf{A}} E(\mathbf{x})$$
$$\text{s.t. } \mathbf{x} = \mathbf{A}^{-1}\mathbf{b} \tag{32}$$

In §5, the linear system comes from FEM: $\mathbf{A} = \mathbf{G}^\mathsf{T}\mathbf{C}\mathbf{G}$ to yield a PDE-constrained optimization (Wang & Solomon, 2021; Wang et al., 2023), and the goal is to optimize $\mathbf{C}$ that encodes a deformation field that minimizes the image matching loss.

### C.8. Generalize to larger kernels

Our approaches can be generalized to a larger kernel size, e.g., to deconvolute $5 \times 5$ or $7 \times 7$ convolutional kernels, with a more complicated implementation. In the current $3 \times 3$ case, one layer of boundary pixels can separate two subdomains P, Q, and it will require two layers of boundary pixels in the case of $5 \times 5$ to separate two subdomains P,Q (or 3 layers of boundary pixels in the case of $7 \times 7$). Similarly, the boundary pixels (at the border of the whole image) in the last Schur step will have two layers of pixels. The parallel elimination procedure will be similar, but will have to account for the fact that the "wire-frame" has two layers of pixels.

In fact, in some sense, our current method **already supports** kernels larger than $3 \times 3$. The actual constraint that we have is that every pixel can only contribute to pixels in the same $5 \times 5$ patch (and recall that pixels at the patch boundary belong to multiple patches). Thus, the convolution window for a pixel can cover the entire 1/2/4 patches it belongs to: for example, pixels at the patch boundary can use a local window of $9 \times 9$ or $9 \times 5$, and it is $5 \times 5$ for an interior pixel, though the window may not be centered at it. Also, recall that the $5 \times 5$ patch size is a hyperparameter that we are free to change arbitrarily in the current method.

## D. Discussions and Implementation Details

### D.1. Details on the algorithm

A hyperparameter in our method is the size of patches, which we choose as $4 \times 4$ (enlarged to $5 \times 5$ for duplicating boundary pixels at interfaces), except for the image of size $2561^2$ where we choose the size to be $5 \times 5$ (enlarged to $6 \times 6$).

### D.2. Distributed representations of sparse systems

In our method, we never have to construct a global sparse matrix as other direct sparse solvers do, leading to extra saving in time. In the initialization step the Schur complements are stored in a distributed fashion.

### D.3. Details on experiment setups

Compared to Python, the Julia ecosystem has much better support for sparse linear algebraic CUDA APIs that are critical in scientific computing. For this reason, when comparing our method with CUDA, we use the up-to-date Julia binding that provides APIs backended by the most recently released cuDSS with cuSolver, cuSparse and CUDA. cuDSS is the state-of-the-art official CUDA library that supports sparse linear solvers including sparse LU (LDU) factorization, becoming our direct point of comparison.

### D.4. Memory complexity

We emphasize that the reported memory for our solver is the maximum GPU usage for its workspace storing all intermediate matrices and can be released after the solution is obtained: if the solver is included as a layer in the neural networks, the memory can be released after the forward pass (though storing the workspace can skip the numerical factorization stage in the back-propagation step to improve performance). For example, if the solver layer is included 10 times in the neural architecture, the peak memory usage for the model is same to including the solver only 1 time. Our solver currently treats symmetric $\mathbf{A}$ as asymmetric ones: future work can leverage the symmetry to save half of the memory usage.

*Table 12.* Memory usage (MB) comparison. Note we do not optimize our memory usage and there is likely large room for improvement. Note that if used in a differentiable solver setting, the memory of CUDA might have to be multiplied by an extra factor of 2 due to transposed solve, while there is no need for our method. (*: while it fits in memory to run the code, using "torch.compile" for an optimized runtime exceeds the limit.)

| | GPU memory usage (MB) | | |
|---|---|---|---|
| Example | CUDA | ours-32 | ours-64 |
| $4097^2$ | 33330 | OOM | OOM |
| $2049^2$ | 8104 | 16094 | 32188 (*) |
| $1025^2$ | 2234 | 3880 | 7244 |
| $513^2$ | 880 | 890 | 1736 |
| $257^2$ | 548 | 242 | 420 |

Our method achieves speed at the cost of memory usage. Table 12 compares the GPU memory usage with the CUDA solver. Note that our current implementation is generous in memory usage and we believe there is likely plenty of

room for improvement. We save every intermediate dense matrices to allow quick experimentation of ideas, and do not use any PyTorch in-place operations, which we find may cause counterintuitive behaviors for sliced matrices, making debugging challenging.

In addition, the memory usage can be significantly reduced or even eliminated, if the kernel $a^{(x,y)}$ is constant spatially; future implementation may take advantage of this. In this case, all dense matrices are the same across the domain, and even have known closed form that can be derived ahead of time.

### D.5. Midpoint reflective boundary condition

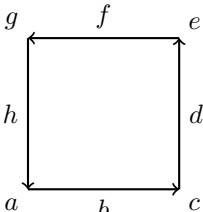

*Figure 23.* Name pixels at the boundary of the image domain in counter-clockwise ordering.

To handle spherical topology, we introduce a *midpoint reflective* boundary condition, as induced by the octahedral parameterization (Praun & Hoppe, 2003) that maps from a surface with spherical topology to the image domain. Shown in Figure 23, the boundary of the image domain is split into the chain $a \to b \to c \to d \to e \to f \to g \to h \to a$. The problem becomes

$$\min_{\mathbf{u}} \frac{1}{2}\mathbf{u}^{\mathsf{T}}\mathbf{D}\mathbf{u} - \mathbf{d}^{\mathsf{T}}\mathbf{u}$$
$$\text{s.t.} \quad \mathbf{u}_a = \mathbf{u}_c = \mathbf{u}_e = \mathbf{u}_g$$
$$\mathbf{u}_{(a\to b)} = \mathbf{u}_{(c\to b)},$$
$$\mathbf{u}_{(c\to d)} = \mathbf{u}_{(e\to d)}, \quad (33)$$
$$\mathbf{u}_{(e\to f)} = \mathbf{u}_{(g\to f)},$$
$$\mathbf{u}_{(g\to h)} = \mathbf{u}_{(a\to h)}.$$

The sub-vector $\mathbf{u}_{(a\to b)}$ does not include two endpoints $a, b$. The constraints come from how the octahedral parameterization (Praun & Hoppe, 2003) specifically sets the boundary condition to enforces the spherical topology. To solve the problem, we stack the unique values in $\mathbf{u}$ into a vector $\mathbf{v}$,

such that

$$\mathbf{u} = \begin{bmatrix} \mathbf{u}_a \\ \mathbf{u}_{(a\to b)} \\ \mathbf{u}_b \\ \mathbf{u}_{(b\to c)} \\ \mathbf{u}_c \\ \mathbf{u}_{(c\to d)} \\ \mathbf{u}_d \\ \mathbf{u}_{(d\to e)} \\ \mathbf{u}_e \\ \mathbf{u}_{(e\to f)} \\ \mathbf{u}_f \\ \mathbf{u}_{(f\to g)} \\ \mathbf{u}_g \\ \mathbf{u}_{(g\to h)} \\ \mathbf{u}_h \\ \mathbf{u}_{(h\to a)} \end{bmatrix} = \begin{bmatrix} \mathbf{u}_a \\ \mathbf{u}_{(a\to b)} \\ \mathbf{u}_b \\ \text{REV}(\mathbf{u}_{(a\to b)}) \\ \mathbf{u}_a \\ \mathbf{u}_{(c\to d)} \\ \mathbf{u}_d \\ \text{REV}(\mathbf{u}_{(c\to d)}) \\ \mathbf{u}_a \\ \mathbf{u}_{(e\to f)} \\ \mathbf{u}_f \\ \text{REV}(\mathbf{u}_{(e\to f)}) \\ \mathbf{u}_a \\ \mathbf{u}_{(g\to h)} \\ \mathbf{u}_h \\ \text{REV}(\mathbf{u}_{(g\to h)}) \end{bmatrix} \qquad \mathbf{v} = \begin{bmatrix} \mathbf{u}_a \\ \mathbf{u}_{(a\to b)} \\ \mathbf{u}_b \\ \mathbf{u}_{(c\to d)} \\ \mathbf{u}_d \\ \mathbf{u}_{(e\to f)} \\ \mathbf{u}_f \\ \mathbf{u}_{(g\to h)} \\ \mathbf{u}_h \end{bmatrix}$$

in which $\text{REV}(\cdot)$ indicates flipping a vector, i.e. the vector with the same elements appearing in the reverse order. In this case, we can reduce the number of variables using $\mathbf{u} = \mathbf{F}\mathbf{v}$. $\mathbf{F}$ is a sparse binary matrix, describing how entries in $\mathbf{u}$ can be copied from entries in $\mathbf{v}$, such that $\mathbf{1} = \mathbf{F}\mathbf{1}$.

System (33) then can be solved via $\mathbf{v} = (\mathbf{F}^{\mathsf{T}}\mathbf{D}\mathbf{F})^{-1}\mathbf{F}^{\mathsf{T}}\mathbf{d}$ and the solution can be recovered from $\mathbf{u} \leftarrow \mathbf{F}\mathbf{v}$.

# E. Detailed Method Description with Necessary Background Information

In this section, we provide a detailed description of our method along with the necessary background, supplementing the brief explanation in the main text (kept short due to space constraints). Nonetheless, understanding the internal details is typically unnecessary for most users, as our method is designed to be used as a black-box module. Readers familiar with numerical methods can skip this section.

**Plug-and-play for end users**   Just as convolution and matrix multiplication are accessed via CUDA (cuDNN/cuBLAS), we that envision our solver can be similarly exposed through low-level APIs (e.g., via a cuDSS-like interface) with further opportunities of optimization, allowing users to easily integrate it without needing to implement anything themselves.

**Notation: matrix slicing**   For a matrix $\mathbf{A} \in \mathbb{R}^{n \times n}$, we use the matrix slicing notation $\mathbf{A}_{rs}$ to denote the submatrix of $\mathbf{A}$ that takes the rows specified by $r$ and columns specified by $s$. For example, for $r = [2, 1, 3]$, $s = [6, 7]$, $\mathbf{A}_{rs}$ refers to:

$$\mathbf{A}_{rs} := \begin{bmatrix} \mathbf{A}_{2,6} & \mathbf{A}_{2,7} \\ \mathbf{A}_{1,6} & \mathbf{A}_{1,7} \\ \mathbf{A}_{3,6} & \mathbf{A}_{3,7} \end{bmatrix} \tag{34}$$

the $3 \times 2$ submatrix that is obtained by selecting the first three rows, in the order of $2, 1, 3$, and 6-th, 7-th columns from matrix $\mathbf{A}$. The lower scripted $\mathbf{A}_{rs}$ should not be confused with upper scripted $\mathbf{A}^{rs}$, which is just a variable that contains $r, s$ as part of its name and does not necessarily have anything to do with slicing.

**Reverse permutation matrix**   Define $\mathbf{J}$ so that $\mathbf{J}^\mathsf{T}\mathbf{X}$ (or $\mathbf{X}\mathbf{J}$) permutes rows (or columns) of $\mathbf{X}$ in reverse order.

$$\mathbf{J} := \begin{bmatrix} & & & 1 \\ & & \cdot^{\cdot^{\cdot}} & \\ & 1 & & \\ 1 & & & \end{bmatrix} \equiv \mathbf{J}^\mathsf{T} \equiv \mathbf{J}^{-1} \tag{35}$$

**Schur complement**   Schur complement reduces the problem of solving the $2 \times 2$ block matrix,

$$\begin{bmatrix} \mathbf{X} & \mathbf{Y} \\ \mathbf{Z} & \mathbf{W} \end{bmatrix} \begin{bmatrix} \mathbf{x} \\ \mathbf{z} \end{bmatrix} = \begin{bmatrix} \mathbf{y} \\ \mathbf{w} \end{bmatrix} \tag{36}$$

to that of solving a smaller matrix $\mathbf{D}\mathbf{x} = \mathbf{d}$, assuming an invertible $\mathbf{W}$, where

$$\mathbf{D} := \left( \mathbf{X} - \mathbf{Y}\mathbf{W}^{-1}\mathbf{Z} \right) \quad \mathbf{d} := \mathbf{y} - \mathbf{Y}\mathbf{W}^{-1}\mathbf{w} \tag{37}$$

by canceling $\mathbf{z}$ using the second line of the equation that rewrites $\mathbf{z} = \mathbf{W}^{-1}[\mathbf{w} - \mathbf{Z}\mathbf{x}]$ as a function of $\mathbf{x}$.

While equations in the paper might look a bit dense, at the high level the overall procedure is quite simple: recursively applying the Schur complement to the system many times to reduce the problem to a smaller system. There are a few extra considerations: 1) implicitly or explicitly re-order rows and columns of the matrix as necessary, so that we know the part we want to eliminate is located at the sub-block $\mathbf{W}$ (or anywhere we prescribed); 2) apply many Schur complements in parallel.

## E.1. Case study: two subdomains

We review some standard concepts before explaining our method for accessibility to a larger audience. The reader may refer to §E.6, a smaller $3 \times 7$ image for clarity of illustration.

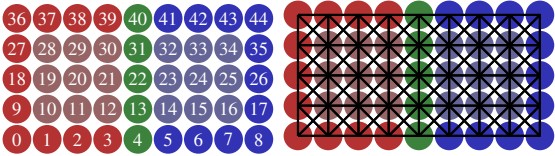

*Figure 24.* The connectivity graph resulted from the use of the $3 \times 3$ kernel size, for a $5 \times 9$ image under the lexicographic sweeping order.

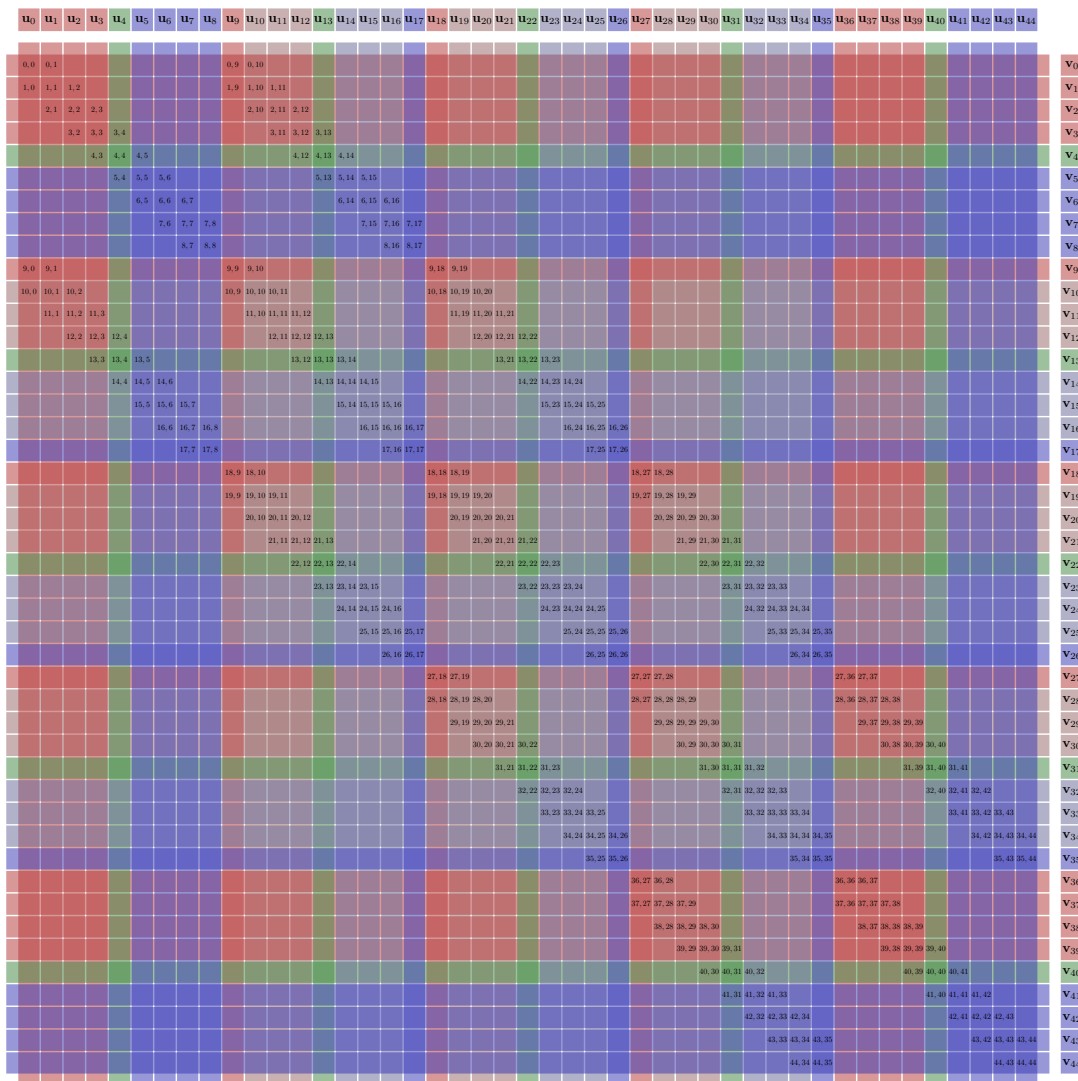

*Figure 25.* For the $5 \times 9$ image, using the pixel indexing in Figure 24, visualization of the sparsity pattern of $\mathbf{A} \in \mathbb{R}^{45 \times 45}$, for the original sparse linear system $\sum_j \mathbf{A}_{ij}\mathbf{u}_j = \mathbf{v}_i, \forall i$. As we can see, the original Laplacian-like $\mathbf{A}$ is a **banded matrix** with three banded diagonals.

For the $5 \times 9$ image shown in Figure 24, each pixel corresponds to a node in the graph and is assigned with an index $i \in \{0, 1, ..., 45-1\}$. Two nodes $i$ and $j$, where $i, j \in \{0, 1, ..., 45-1\}$, are connected by an edge if the $3 \times 3$ kernel centered at one node will cover the other node. $\mathbf{A}_{ij} = 0$ if node $i$ and node $j$ are not connected by an edge in the graph.

For the original sparse linear system, $\mathbf{A} \in \mathbb{R}^{45 \times 45}$:

$$\sum_{j=0}^{45-1} \mathbf{A}_{ij}\mathbf{u}_j = \mathbf{v}_i, \forall i \in \{0, 1, ..., 45-1\}. \tag{38}$$

Figure 25 visualizes the sparsity pattern of $\mathbf{A} \in \mathbb{R}^{45 \times 45}$. In this visualization of $\mathbf{A}$, only where there is a $(i, j)$ corresponds an entry that $\mathbf{A}_{ij} \neq 0$—namely $i$ and $j$ are connected by an edge in Figure 24, and all other entries that we omitted are zeros. The original $\mathbf{A}$ is a Laplacian-like matrix with a 9-point stencil. $\mathbf{A}$ is a **banded matrix** with three banded diagonals. Previous direct solvers have to explicitly maintain the sparse matrix $\mathbf{A}$ shown in Figure 25, while our solver does not. Instead, we work with a more compact representation that takes advantage of the regular grid structure: we start with the dense block submatrices shown in Figure 26.

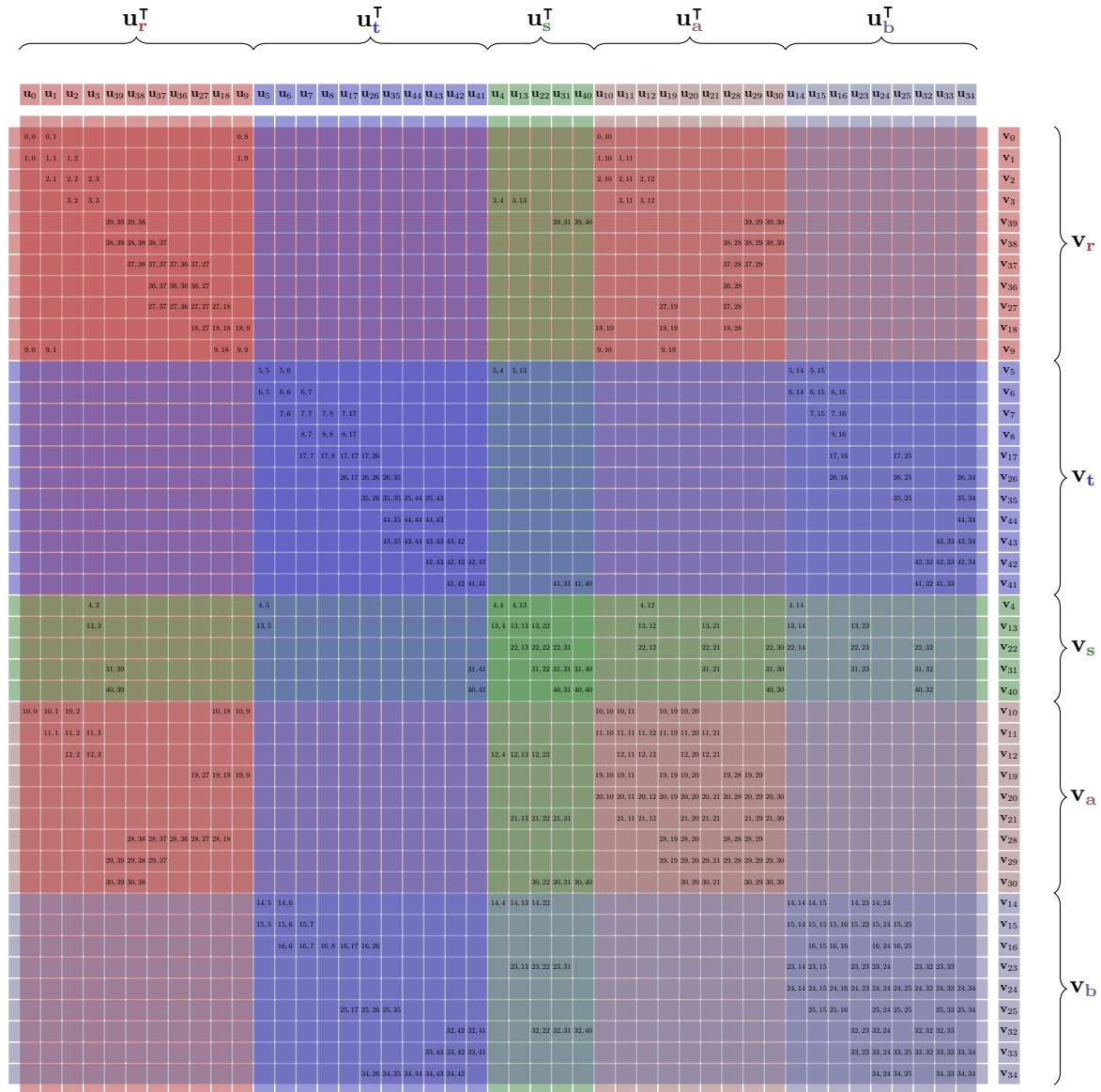

*Figure 26.* After reordering rows and columns, the original system becomes the $5 \times 5$ block system:

$$\begin{bmatrix} \mathbf{A_{rr}} & \mathbf{0} = \mathbf{A_{rt}} & \mathbf{A_{rs}} & \mathbf{A_{ra}} & \mathbf{0} = \mathbf{A_{rb}} \\ \mathbf{0} = \mathbf{A_{tr}} & \mathbf{A_{tt}} & \mathbf{A_{ts}} & \mathbf{0} = \mathbf{A_{ta}} & \mathbf{A_{tb}} \\ \mathbf{A_{sr}} & \mathbf{A_{st}} & \mathbf{A_{ss}} & \mathbf{A_{sa}} & \mathbf{A_{sb}} \\ \mathbf{A_{ar}} & \mathbf{0} = \mathbf{A_{at}} & \mathbf{A_{as}} & \mathbf{A_{aa}} & \mathbf{0} = \mathbf{A_{ab}} \\ \mathbf{0} = \mathbf{A_{br}} & \mathbf{A_{bt}} & \mathbf{A_{bs}} & \mathbf{0} = \mathbf{A_{ba}} & \mathbf{A_{bb}} \end{bmatrix} \begin{bmatrix} \mathbf{u_r} \\ \mathbf{u_t} \\ \mathbf{u_s} \\ \mathbf{u_a} \\ \mathbf{u_b} \end{bmatrix} = \begin{bmatrix} \mathbf{v_r} \\ \mathbf{v_t} \\ \mathbf{v_s} \\ \mathbf{v_a} \\ \mathbf{v_b} \end{bmatrix}.$$

By pivoting the rows and columns of $\mathbf{A}$, we arrive at the linear system in Figure 26. It is easy to verify that the pivoting does not change the problem: it only re-orders the equations and variables. From Figure 26, now it is visually clear that after the row- and column- pivoting, some submatrices are zero. We treat the nonzero block matrices as dense matrices though they can be sparse.

Now, let us start with:

$$\begin{bmatrix} \mathbf{A}_{rr} & \mathbf{0} & \mathbf{A}_{rs} & \mathbf{A}_{ra} & \mathbf{0} \\ \mathbf{0} & \mathbf{A}_{tt} & \mathbf{A}_{ts} & \mathbf{0} & \mathbf{A}_{tb} \\ \mathbf{A}_{sr} & \mathbf{A}_{st} & \mathbf{A}_{ss} & \mathbf{A}_{sa} & \mathbf{A}_{sb} \\ \mathbf{A}_{ar} & \mathbf{0} & \mathbf{A}_{as} & \mathbf{A}_{aa} & \mathbf{0} \\ \mathbf{0} & \mathbf{A}_{bt} & \mathbf{A}_{bs} & \mathbf{0} & \mathbf{A}_{bb} \end{bmatrix} \begin{bmatrix} \mathbf{u}_r \\ \mathbf{u}_t \\ \mathbf{u}_s \\ \mathbf{u}_a \\ \mathbf{u}_b \end{bmatrix} = \begin{bmatrix} \mathbf{v}_r \\ \mathbf{v}_t \\ \mathbf{v}_s \\ \mathbf{v}_a \\ \mathbf{v}_b \end{bmatrix}. \tag{39}$$

where

$$\mathbf{s} = [4, 13, 22, 31, 40],$$
$$\mathbf{r} = [0, 1, 2, 3, 39, 38, 37, 36, 27, 18, 9],$$
$$\mathbf{a} = [10, 11, 12, 19, 20, 21, 28, 29, 30], \tag{40}$$
$$\mathbf{t} = [5, 6, 7, 8, 17, 26, 35, 44, 43, 42, 41],$$
$$\mathbf{b} = [14, 15, 16, 23, 24, 25, 32, 33, 34].$$

**Patch division and patch-wise finite element methods**   Recall that

$$\mathbf{A}_{ss} = \mathbf{A}_{ss}^{(P)} + \mathbf{A}_{ss}^{(Q)}, \tag{41}$$
$$\mathbf{v}_s = \mathbf{v}_s^{(P)} + \mathbf{v}_s^{(Q)} \tag{42}$$

can be divided into contributions from $P$ and $Q$, resp. With this partition, the computations for patch P and Q are made independent from each other, and thus can be done in parallel. The values of $\mathbf{A}_{ss}^{(P)}$ and $\mathbf{A}_{ss}^{(Q)}$ can be arbitrary, as long as their summation is the correct $\mathbf{A}_{ss}$. In fact, their values are never used standalone, only their sum $\mathbf{A}_{ss}^{(P)} + \mathbf{A}_{ss}^{(Q)}$ is used (they become parts of the matrices $\mathbf{P}, \mathbf{Q}$ which are summed later).

Especially for applications like PDEs (discretized with first-order piecewise linear FEM), each patch P and Q will contribution to the value of $\mathbf{A}_{ss}$ independently: their portions of contributions $\mathbf{A}_{ss}^{(P)}$ and $\mathbf{A}_{ss}^{(Q)}$ only depend on information within the patch P and Q, respectively. This makes the computation within P and Q exactly independent from each other, easing parallel implementation.

**System partitioning**

$$\begin{bmatrix} \mathbf{A}_{rr} & \mathbf{0} & \mathbf{A}_{rs} & \mathbf{A}_{ra} & \mathbf{0} \\ \mathbf{0} & \mathbf{A}_{tt} & \mathbf{A}_{ts} & \mathbf{0} & \mathbf{A}_{tb} \\ \mathbf{A}_{sr} & \mathbf{A}_{st} & \mathbf{A}_{ss}^{(P)} + \mathbf{A}_{ss}^{(Q)} & \mathbf{A}_{sa} & \mathbf{A}_{sb} \\ \mathbf{A}_{ar} & \mathbf{0} & \mathbf{A}_{as} & \mathbf{A}_{aa} & \mathbf{0} \\ \mathbf{0} & \mathbf{A}_{bt} & \mathbf{A}_{bs} & \mathbf{0} & \mathbf{A}_{bb} \end{bmatrix} \begin{bmatrix} \mathbf{u}_r \\ \mathbf{u}_t \\ \mathbf{u}_s \\ \mathbf{u}_a \\ \mathbf{u}_b \end{bmatrix} = \begin{bmatrix} \mathbf{v}_r \\ \mathbf{v}_t \\ \mathbf{v}_s^{(P)} + \mathbf{v}_s^{(Q)} \\ \mathbf{v}_a \\ \mathbf{v}_b \end{bmatrix} \tag{43}$$

$$\begin{bmatrix} \mathbf{A}_{rr} & \mathbf{0} & \mathbf{A}_{rs} & \mathbf{A}_{ra} & \mathbf{0} \\ \mathbf{0} & \mathbf{0} & \mathbf{0} & \mathbf{0} & \mathbf{0} \\ \mathbf{A}_{sr} & \mathbf{0} & \mathbf{A}_{ss}^{(P)} & \mathbf{A}_{sa} & \mathbf{0} \\ \mathbf{A}_{ar} & \mathbf{0} & \mathbf{A}_{as} & \mathbf{A}_{aa} & \mathbf{0} \\ \mathbf{0} & \mathbf{0} & \mathbf{0} & \mathbf{0} & \mathbf{0} \end{bmatrix} \begin{bmatrix} \mathbf{u}_r \\ \mathbf{u}_t \\ \mathbf{u}_s \\ \mathbf{u}_a \\ \mathbf{u}_b \end{bmatrix} + \begin{bmatrix} \mathbf{0} & \mathbf{0} & \mathbf{0} & \mathbf{0} & \mathbf{0} \\ \mathbf{0} & \mathbf{A}_{tt} & \mathbf{A}_{ts} & \mathbf{0} & \mathbf{A}_{tb} \\ \mathbf{0} & \mathbf{A}_{st} & \mathbf{A}_{ss}^{(Q)} & \mathbf{0} & \mathbf{A}_{sb} \\ \mathbf{0} & \mathbf{0} & \mathbf{0} & \mathbf{0} & \mathbf{0} \\ \mathbf{0} & \mathbf{A}_{bt} & \mathbf{A}_{bs} & \mathbf{0} & \mathbf{A}_{bb} \end{bmatrix} \begin{bmatrix} \mathbf{u}_r \\ \mathbf{u}_t \\ \mathbf{u}_s \\ \mathbf{u}_a \\ \mathbf{u}_b \end{bmatrix} = \begin{bmatrix} \mathbf{v}_r \\ \mathbf{0} \\ \mathbf{v}_s^{(P)} \\ \mathbf{v}_a \\ \mathbf{0} \end{bmatrix} + \begin{bmatrix} \mathbf{0} \\ \mathbf{v}_t \\ \mathbf{v}_s^{(Q)} \\ \mathbf{0} \\ \mathbf{v}_b \end{bmatrix} \tag{44}$$

Instead of working with the sparse matrix $\mathbf{A}$ and a right-hand side $\mathbf{b}$, our method works with the dense blocks—that are compactly put in two tensors $\boldsymbol{\alpha}, \boldsymbol{\beta}$ as we will introduce later—that come from the nonzero parts of $\mathbf{A}$:

$$\begin{bmatrix} \mathbf{A}_{rr} & \mathbf{A}_{rs} & \mathbf{A}_{ra} \\ \mathbf{A}_{sr} & \mathbf{A}_{ss}^{(P)} & \mathbf{A}_{sa} \\ \mathbf{A}_{ar} & \mathbf{A}_{as} & \mathbf{A}_{aa} \end{bmatrix} \begin{bmatrix} \mathbf{u}_r \\ \mathbf{u}_s \\ \mathbf{u}_a \end{bmatrix} \overset{?}{=} \begin{bmatrix} \mathbf{v}_r \\ \mathbf{v}_s^{(P)} \\ \mathbf{v}_a \end{bmatrix} \tag{45}$$

$$\begin{bmatrix} \mathbf{A}_{tt} & \mathbf{A}_{ts} & \mathbf{A}_{tb} \\ \mathbf{A}_{st} & \mathbf{A}_{ss}^{(Q)} & \mathbf{A}_{sb} \\ \mathbf{A}_{bt} & \mathbf{A}_{bs} & \mathbf{A}_{bb} \end{bmatrix} \begin{bmatrix} \mathbf{u}_t \\ \mathbf{u}_s \\ \mathbf{u}_b \end{bmatrix} \overset{?}{=} \begin{bmatrix} \mathbf{v}_t \\ \mathbf{v}_s^{(Q)} \\ \mathbf{v}_b \end{bmatrix} \tag{46}$$

The symbol "$\overset{?}{=}$" indicates the above systems cannot be solved directly, since our assumption is that the values $\mathbf{A}_{ss}^{(P)}$ and $\mathbf{A}_{ss}^{(Q)}$ can be arbitrary, and only their summation is a known value $\mathbf{A}_{ss}$. In fact, the solution depends on every patch so any local computation cannot yield the correct solution. But we can still eliminate $\mathbf{u}_a, \mathbf{u}_b$ by writing their values as a function of $\mathbf{u}_r, \mathbf{u}_s, \mathbf{u}_t$.

$$
\begin{bmatrix} \mathbf{u}_a \\ \mathbf{u}_b \end{bmatrix} = \begin{bmatrix} \mathbf{A}_{aa}^{-1} & \mathbf{0} \\ \mathbf{0} & \mathbf{A}_{bb}^{-1} \end{bmatrix} \left( \begin{bmatrix} \mathbf{v}_a \\ \mathbf{v}_b \end{bmatrix} - \begin{bmatrix} \mathbf{A}_{ar} & \mathbf{0} & \mathbf{A}_{as} \\ \mathbf{0} & \mathbf{A}_{bt} & \mathbf{A}_{bs} \end{bmatrix} \begin{bmatrix} \mathbf{u}_r \\ \mathbf{u}_t \\ \mathbf{u}_s \end{bmatrix} \right)
\tag{47}
$$

and plugging it into the joint system:

$$
\begin{bmatrix} \mathbf{A}_{rr} & \mathbf{0} & \mathbf{A}_{rs} \\ \mathbf{0} & \mathbf{A}_{tt} & \mathbf{A}_{ts} \\ \mathbf{A}_{sr} & \mathbf{A}_{st} & \mathbf{A}_{ss} \end{bmatrix} \begin{bmatrix} \mathbf{u}_r \\ \mathbf{u}_t \\ \mathbf{u}_s \end{bmatrix} + \begin{bmatrix} \mathbf{A}_{ra} & \mathbf{0} \\ \mathbf{0} & \mathbf{A}_{tb} \\ \mathbf{A}_{sa} & \mathbf{A}_{sb} \end{bmatrix} \begin{bmatrix} \mathbf{u}_a \\ \mathbf{u}_b \end{bmatrix} = \begin{bmatrix} \mathbf{v}_r \\ \mathbf{v}_t \\ \mathbf{v}_s \end{bmatrix}
\tag{48}
$$

which leads to:

$$
\begin{bmatrix} \mathbf{A}_{rr} & \mathbf{0} & \mathbf{A}_{rs} \\ \mathbf{0} & \mathbf{A}_{tt} & \mathbf{A}_{ts} \\ \mathbf{A}_{sr} & \mathbf{A}_{st} & \mathbf{A}_{ss} \end{bmatrix} \begin{bmatrix} \mathbf{u}_r \\ \mathbf{u}_t \\ \mathbf{u}_s \end{bmatrix} + \begin{bmatrix} \mathbf{A}_{ra} & \mathbf{0} \\ \mathbf{0} & \mathbf{A}_{tb} \\ \mathbf{A}_{sa} & \mathbf{A}_{sb} \end{bmatrix} \begin{bmatrix} \mathbf{A}_{aa}^{-1} & \mathbf{0} \\ \mathbf{0} & \mathbf{A}_{bb}^{-1} \end{bmatrix} \left( \begin{bmatrix} \mathbf{v}_a \\ \mathbf{v}_b \end{bmatrix} - \begin{bmatrix} \mathbf{A}_{ar} & \mathbf{0} & \mathbf{A}_{as} \\ \mathbf{0} & \mathbf{A}_{bt} & \mathbf{A}_{bs} \end{bmatrix} \begin{bmatrix} \mathbf{u}_r \\ \mathbf{u}_t \\ \mathbf{u}_s \end{bmatrix} \right) = \begin{bmatrix} \mathbf{v}_r \\ \mathbf{v}_t \\ \mathbf{v}_s \end{bmatrix}
\tag{49}
$$

which further simplifies to:

$$
\begin{bmatrix} \mathbf{A}_{rr} - \mathbf{A}_{ra}\mathbf{A}_{aa}^{-1}\mathbf{A}_{ar} & \mathbf{0} & \mathbf{A}_{rs} - \mathbf{A}_{ra}\mathbf{A}_{aa}^{-1}\mathbf{A}_{as} \\ \mathbf{0} & \mathbf{A}_{tt} - \mathbf{A}_{tb}\mathbf{A}_{bb}^{-1}\mathbf{A}_{bt} & \mathbf{A}_{ts} - \mathbf{A}_{tb}\mathbf{A}_{bb}^{-1}\mathbf{A}_{bs} \\ \mathbf{A}_{sr} - \mathbf{A}_{sa}\mathbf{A}_{aa}^{-1}\mathbf{A}_{ar} & \mathbf{A}_{st} - \mathbf{A}_{sb}\mathbf{A}_{bb}^{-1}\mathbf{A}_{bt} & \mathbf{A}_{ss} - \mathbf{A}_{sa}\mathbf{A}_{aa}^{-1}\mathbf{A}_{as} - \mathbf{A}_{sb}\mathbf{A}_{bb}^{-1}\mathbf{A}_{bs} \end{bmatrix} \begin{bmatrix} \mathbf{u}_r \\ \mathbf{u}_t \\ \mathbf{u}_s \end{bmatrix}
$$
$$
= \begin{bmatrix} \mathbf{v}_r - \mathbf{A}_{ra}\mathbf{A}_{aa}^{-1}\mathbf{v}_a \\ \mathbf{v}_t - \mathbf{A}_{tb}\mathbf{A}_{bb}^{-1}\mathbf{v}_b \\ \mathbf{v}_s - \mathbf{A}_{sa}\mathbf{A}_{aa}^{-1}\mathbf{v}_a - \mathbf{A}_{sb}\mathbf{A}_{bb}^{-1}\mathbf{v}_b \end{bmatrix}
\tag{50}
$$

solving which yields the solution to the original problem.

$$
\begin{bmatrix} \mathbf{A}_{rr} - \mathbf{A}_{ra}\mathbf{A}_{aa}^{-1}\mathbf{A}_{ar} & \mathbf{0} & \mathbf{A}_{rs} - \mathbf{A}_{ra}\mathbf{A}_{aa}^{-1}\mathbf{A}_{as} \\ \mathbf{0} & \mathbf{0} & \mathbf{0} \\ \mathbf{A}_{sr} - \mathbf{A}_{sa}\mathbf{A}_{aa}^{-1}\mathbf{A}_{ar} & \mathbf{0} & \mathbf{A}_{ss}^{(P)} - \mathbf{A}_{sa}\mathbf{A}_{aa}^{-1}\mathbf{A}_{as} \end{bmatrix} \begin{bmatrix} \mathbf{u}_r \\ \mathbf{u}_t \\ \mathbf{u}_s \end{bmatrix} + \begin{bmatrix} \mathbf{0} & \mathbf{0} & \mathbf{0} \\ \mathbf{0} & \mathbf{A}_{tt} - \mathbf{A}_{tb}\mathbf{A}_{bb}^{-1}\mathbf{A}_{bt} & \mathbf{A}_{ts} - \mathbf{A}_{tb}\mathbf{A}_{bb}^{-1}\mathbf{A}_{bs} \\ \mathbf{0} & \mathbf{A}_{st} - \mathbf{A}_{sb}\mathbf{A}_{bb}^{-1}\mathbf{A}_{bt} & \mathbf{A}_{ss}^{(Q)} - \mathbf{A}_{sb}\mathbf{A}_{bb}^{-1}\mathbf{A}_{bs} \end{bmatrix} \begin{bmatrix} \mathbf{u}_r \\ \mathbf{u}_t \\ \mathbf{u}_s \end{bmatrix} =
$$
$$
\begin{bmatrix} \mathbf{v}_r - \mathbf{A}_{ra}\mathbf{A}_{aa}^{-1}\mathbf{v}_a \\ \mathbf{0} \\ \mathbf{v}_s^{(P)} - \mathbf{A}_{sa}\mathbf{A}_{aa}^{-1}\mathbf{v}_a \end{bmatrix} + \begin{bmatrix} \mathbf{0} \\ \mathbf{v}_t - \mathbf{A}_{tb}\mathbf{A}_{bb}^{-1}\mathbf{v}_b \\ \mathbf{v}_s^{(Q)} - \mathbf{A}_{sb}\mathbf{A}_{bb}^{-1}\mathbf{v}_b \end{bmatrix}
\tag{51}
$$

The derivations demonstrate that the computation can be split into contributions from each subdomain.

### E.2. Graph algorithm perspectives

Gaussian elimination, including our method, can be simply understood as a *graph algorithm* that removes nodes from a graph, while updating the edge weights accordingly. The graph algorithm perspective can make the overall algorithm significantly easier to understand.

*Weighted graph.* The matrix $\mathbf{A}$ plays the same role as the adjacency matrix in graph theory. As in Figure 21, initially each pixel in the image is a node in the graph, and two nodes are connected by an edge with weights $\mathbf{A}_{ij}$, if the pixels $i$ and $j$ are adjacent (as defined in §2).

**Node elimination**   Sequential Gaussian elimination removes nodes one by one from the graph. When removing a node $k$ from the current graph, we only need to apply a simple modification to matrix $\mathbf{A}$: for all pair of nodes $i, j$ that are both adjacent to $k$, update $\mathbf{A}_{ij}$ by subtracting the term $\mathbf{A}_{ik}\mathbf{A}_{kk}^{-1}\mathbf{A}_{kj}$ from it.

$$
\begin{aligned}
\mathbf{A}_{ij} &\leftarrow \mathbf{A}_{ij} - \mathbf{A}_{ik}\mathbf{A}_{kk}^{-1}\mathbf{A}_{kj}, \quad \forall i, j \\
\mathbf{A}_{ik} &\leftarrow 0, \quad \forall i \\
\mathbf{A}_{ki} &\leftarrow 0, \quad \forall i \\
\mathbf{A}_{kk} &\leftarrow 0.
\end{aligned}
\qquad \text{(weights updating)} \qquad (52)
$$

Since $\mathbf{A}_{k,:}$, $\mathbf{A}_{:,k}$—the row and column that correspond to node $k$ become zero after the step, we can actually delete them from the matrix; note we will need to relabel and nodes after the deletion. That is, when deleting a node $k$, the indirect influence of node $j$ on $i$ via $k$, is attributed through a direct influence of node $j$ on $i$.

While the sequential Gaussian elimination is sufficient to solve the linear system and is mathematical equivalent to our method, it is inefficient. As an improvement, we can remove multiple nodes in a single elimination, to leverage dense CUDA BLAS kernel.

**Block (multiple nodes) elimination**   The updating formula generalizes to the case when $i, j, k$ each is not a single node, but each consists of a set of nodes. Then, we have block Gaussian elimination: first divide the domain into three sets of nodes as $r$, $s$, and $t$, such that any node in $r$ is not connected to any node in $t$ as separated by $s$. The $3 \times 3$ block:

$$
\begin{bmatrix}
\mathbf{A}_{rr} & \mathbf{A}_{rs} & 0 = \mathbf{A}_{rt} \\
\mathbf{A}_{sr} & \mathbf{A}_{ss} & \mathbf{A}_{st} \\
0 = \mathbf{A}_{tr} & \mathbf{A}_{ts} & \mathbf{A}_{tt}
\end{bmatrix}
$$

becomes the $2 \times 2$ block:

$$
\begin{bmatrix}
\mathbf{A}_{ss} - \mathbf{A}_{sr}\mathbf{A}_{rr}^{-1}\mathbf{A}_{rs} & \mathbf{A}_{st} \\
\mathbf{A}_{ts} & \mathbf{A}_{tt}
\end{bmatrix}
$$

Namely, the update rule is to subtract the adjustment term $\mathbf{A}_{sr}\mathbf{A}_{rr}^{-1}\mathbf{A}_{rs}$.

$$
\begin{aligned}
\mathbf{A}_{rr} &\leftarrow \mathbf{A}_{rr} - \mathbf{A}_{sr}\mathbf{A}_{rr}^{-1}\mathbf{A}_{rs}, \\
\mathbf{A}_{rs} &\leftarrow \mathbf{0}, \\
\mathbf{A}_{sr} &\leftarrow \mathbf{0}, \\
\mathbf{A}_{rr} &\leftarrow \mathbf{0}.
\end{aligned}
\qquad \text{(weights updating)} \qquad (53)
$$

**Parallel block elimination**   For efficiency, we apply many Schur complements in parallel. In §3, we simply do two block Gaussian elimination at the same time. Then our algorithm is basically a parallel block Gaussian elimination in which many groups of nodes (marked in yellow in Figure 3) are removed by concurrently subtracting many adjustment terms. The major effort in the code is "index-tracking": carefully track what the indices of remaining nodes become after some nodes are removed.

### E.3. Parallel Schwarz elimination step

Previous discussions of the Schwarz step in §3.2.1 are only for illustrative purposes and do not match our implementation of the Schwarz step. Instead, in this section we provide full details for the Schwarz step that eliminates the interior of each patch.

**The boundary-first ordering.** As shown in Figure 27, we adopt a boundary-first ordering that uses $\eta = \{0 \sim 15\}$ to index the boundary nodes, and $\omega = \{16 \sim 24\}$ for interior nodes.

$$\eta := [0, 1, 2, ..15] \tag{54}$$
$$\omega := [16, 17, 18, ...24] \tag{55}$$

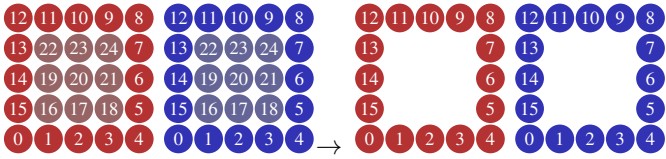

*Figure 27.* The boundary-first ordering. Schwarz step removes the interior pixels for each patch in parallel.

In contrast, Figure 28 shows the lexicographic sweeping order.

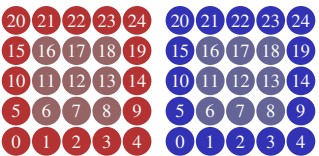

*Figure 28.* The lexicographic sweeping ordering.

Define the permutation vector $\boldsymbol{\mu}$ that is useful to map array stored in the lexicographic sweeping order to the boundary-first ordering, and its inverse permutation vector $\boldsymbol{\nu}$:

$$\boldsymbol{\mu} := [0, 1, 2, 3, 4, 9, 14, 19, 24, 23, 22, 21, 20, 15, 10, 5, 6, 7, 8, 11, 12, 13, 16, 17, 18] \tag{56}$$
$$\boldsymbol{\nu} := [0, 1, 2, 3, 4, 15, 16, 17, 18, 5, 14, 19, 20, 21, 6, 13, 22, 23, 24, 7, 12, 11, 10, 9, 8] \tag{57}$$

**Patch-wise subsystems.** This step directly constructs the matrices $\mathbf{P}, \mathbf{Q} \in \mathbb{R}^{16 \times 16}$. The input to the Schwarz step is the sparse system matrices $\mathbf{H}^{(P)}, \mathbf{H}^{(Q)} \in \mathbb{R}^{25 \times 25}$ and right-hand sides $\mathbf{h}^{(P)}, \mathbf{h}^{(Q)} \in \mathbb{R}^{25 \times 1}$. Although $\mathbf{H}^{(P)}, \mathbf{H}^{(Q)}$ are sparse, we treat them as dense matrices in our algorithms. $(\mathbf{H}^{(P)}, \mathbf{h}^{(P)})$ and $(\mathbf{H}^{(Q)}, \mathbf{h}^{(Q)})$ play the role of sub-systems in previous derivations.

$$\mathbf{H}^{(P)} :\overset{\mathcal{P}}{\leftarrow} \begin{bmatrix} \mathbf{A}_{\mathbf{rr}} & \mathbf{A}_{\mathbf{rs}} & \mathbf{A}_{\mathbf{ra}} \\ \mathbf{A}_{\mathbf{sr}} & \mathbf{A}_{\mathbf{ss}}^{(P)} & \mathbf{A}_{\mathbf{sa}} \\ \mathbf{A}_{\mathbf{ar}} & \mathbf{A}_{\mathbf{as}} & \mathbf{A}_{\mathbf{aa}} \end{bmatrix} \quad \mathbf{h}^{(P)} :\overset{\mathcal{P}}{\leftarrow} \begin{bmatrix} \mathbf{v}_{\mathbf{r}} \\ \mathbf{v}_{\mathbf{s}}^{(P)} \\ \mathbf{v}_{\mathbf{a}} \end{bmatrix} \tag{58}$$

$$\mathbf{H}^{(Q)} :\overset{\mathcal{P}}{\leftarrow} \begin{bmatrix} \mathbf{A}_{\mathbf{tt}} & \mathbf{A}_{\mathbf{ts}} & \mathbf{A}_{\mathbf{tb}} \\ \mathbf{A}_{\mathbf{st}} & \mathbf{A}_{\mathbf{ss}}^{(Q)} & \mathbf{A}_{\mathbf{sb}} \\ \mathbf{A}_{\mathbf{bt}} & \mathbf{A}_{\mathbf{bs}} & \mathbf{A}_{\mathbf{bb}} \end{bmatrix} \quad \mathbf{h}^{(Q)} :\overset{\mathcal{P}}{\leftarrow} \begin{bmatrix} \mathbf{v}_{\mathbf{t}} \\ \mathbf{v}_{\mathbf{s}}^{(Q)} \\ \mathbf{v}_{\mathbf{b}} \end{bmatrix} \tag{59}$$

In fact, our algorithm always adopts $\mathbf{H}$ as the direct representation of $\mathbf{A}$, and never explicitly constructs $\mathbf{A}$ as a matrix. Using $\mathbf{H}$ is a much more natural choice to work with parallel linear solvers.

So, our method directly constructs matrix $\mathbf{H}, \mathbf{h}$ in the following way: the entry $\mathbf{H}_{ij}^{(P)}$ represents some interaction or affinity between node $i$ and node $j$ in the subdomain $P$, under the *boundary-first ordering* shown in Figure 27. For example, it is

common that the patch-wise sub-system is given in the lexicographic sweeping order shown in Figure 28 as a matrix and vector $\mathbf{E}, \mathbf{e}$ of the same size. in this case, we should first map it to the boundary-first ordering using:

$$\mathbf{H} := \mathbf{E}[\boldsymbol{\mu}, \boldsymbol{\mu}], \quad \mathbf{h} := \mathbf{e}[\boldsymbol{\mu}, :] \tag{60}$$

The symbol ":$\overset{\mathcal{P}}{\leftarrow}$" amounts to using some permutation vector in this way.

We will also be able to map it back to the lexicographic sweeping order if necessary using the inverse permutation vector:

$$\mathbf{E} = \mathbf{H}[\boldsymbol{\nu}, \boldsymbol{\nu}], \quad \mathbf{e} = \mathbf{h}[\boldsymbol{\nu}, :] \tag{61}$$

We can store the patch-wise sub-systems in the tensors $\boldsymbol{\alpha}^{(*)} \in \mathbb{R}^{2 \times 1 \times 25 \times 25}, \boldsymbol{\beta}^{(*)} \in \mathbb{R}^{2 \times 1 \times 25 \times 1}$:

$$\boldsymbol{\alpha}^{(*)}[0, 0, :, :] := \mathbf{H}^{(P)}, \quad \boldsymbol{\beta}^{(*)}[0, 0, :, :] := \mathbf{h}^{(P)}$$
$$\boldsymbol{\alpha}^{(*)}[1, 0, :, :] := \mathbf{H}^{(Q)}, \quad \boldsymbol{\beta}^{(*)}[1, 0, :, :] := \mathbf{h}^{(Q)}$$

Then, we call Algorithm 2 for a parallel implementation of the Schwarz step:

---

**Algorithm 2** The parallel Schwarz step. Forward pass.

---

**Require:** $\boldsymbol{\alpha}^{(*)} \in \mathbb{R}^{d_1 \times d_2 \times 25 \times 25}, \boldsymbol{\beta}^{(*)} \in \mathbb{R}^{d_1 \times d_2 \times 25 \times 1}$
  $(d_1, d_2)$ is $(2^{k/2}, 2^{k/2})$, or $(2, 1)$ in the 2-patch illustrative example.

$$\mathbf{X} := \boldsymbol{\alpha}[:, :, \boldsymbol{\eta}, \boldsymbol{\eta}] \qquad \mathbf{Y} := \boldsymbol{\alpha}[:, :, \boldsymbol{\eta}, \boldsymbol{\omega}] \qquad \mathbf{y} := \boldsymbol{\beta}[:, :, \boldsymbol{\eta}, :], \tag{62}$$
$$\mathbf{Z} := \boldsymbol{\alpha}[:, :, \boldsymbol{\omega}, \boldsymbol{\eta}] \qquad \mathbf{W} := \boldsymbol{\alpha}[:, :, \boldsymbol{\omega}, \boldsymbol{\omega}] \qquad \mathbf{w} := \boldsymbol{\beta}[:, :, \boldsymbol{\omega}, :]. \tag{63}$$

$$\mathbf{D} := \left(\mathbf{X} - \mathbf{Y}\mathbf{W}^{-1}\mathbf{Z}\right) \quad \mathbf{d} := \mathbf{y} - \mathbf{Y}\mathbf{W}^{-1}\mathbf{w} \tag{64}$$

$$\boldsymbol{\alpha} := \mathbf{D}, \quad \boldsymbol{\beta} := \mathbf{d} \tag{65}$$

  **return** $\boldsymbol{\alpha}^{(0)} \in \mathbb{R}^{d_1 \times d_2 \times 16 \times 16}, \boldsymbol{\beta}^{(0)} \in \mathbb{R}^{d_1 \times d_2 \times 16 \times 1}$

---

Algorithm 2 simultaneously calculates the new sub-systems for all subdomains—$\mathbf{P}$ and $\mathbf{Q}$ in the case of two subdomains. and the sub-systems can be fetched from the tensor $\boldsymbol{\alpha}$.

$$\mathbf{P} = \boldsymbol{\alpha}[0, 0, :, :]$$
$$\mathbf{Q} = \boldsymbol{\alpha}[1, 0, :, :]$$

Algorithm 2 still applies when there is an $d_1 \times d_2$ array of patches, instead of an $2 \times 1$ array of patches that we used as the illustration example. Algorithm 2 conducts batch-wise computation that simultaneously calculates the new sub-systems for all patches.

### E.4. Details on Schur step

Now we provide details on the Schur step that merges every two adjacent sub-systems, to supplement §3.2.2.

The Schur step assembles the new system $\mathbf{D}$ by Schur "involuting" the sub-systems $\mathbf{P}, \mathbf{Q}$. The step takes as input the left-hand and right-hand sides $(\mathbf{P}, \mathbf{p})$ and $(\mathbf{Q}, \mathbf{q})$, and outputs a new left-hand and right-hand sides $(\mathbf{D}, \mathbf{d})$.

We store the sub-systems in a four dimensional tensor $\boldsymbol{\alpha}^{(j)}$ that $\boldsymbol{\alpha}^{(j)}[c, d, :, :]$ stores the system matrix at $(c, d)$ in the array of subdomains.

---

**Algorithm 3** The parallel Schwarz step. Backward pass.

---

**Require:** $\boldsymbol{\chi}^{(0)}$: tensor of size $(d_1, d_2, 16, 16)$.

$$\mathbf{x} := \mathbf{u}_{\boldsymbol{\eta}} \tag{66}$$

$$\mathbf{u}_{\boldsymbol{\omega}} := \mathbf{W}^{-1} \left( \mathbf{w} - \mathbf{Z}\mathbf{x} \right) \quad \text{(back-substitution)}, \tag{67}$$

$$\boldsymbol{\chi}^{(*)} := \text{ZEROS}(2^{k/2}, 2^{k/2}, 25 \cdot 2^i, 1). \tag{68}$$

$$\boldsymbol{\chi}^{(*)}[:, :, \boldsymbol{\eta}, :] := \mathbf{u}_{\boldsymbol{\eta}} \tag{69}$$

$$\boldsymbol{\chi}^{(*)}[:, :, \boldsymbol{\omega}, :] := \mathbf{u}_{\boldsymbol{\omega}} \tag{70}$$

**return** $\boldsymbol{\chi}^{(*)}$ tensor of size $(d_1, d_2, 25, 25)$.

---

$\boldsymbol{\alpha}^{(j)}$ contains all reduced systems. For $j = (2i + 1)$, the $j$-th Schur step takes as input $\boldsymbol{\alpha}^{(j)}$ of size $(2^{k/2-i}, 2^{k/2-i}, 16 \cdot 2^i, 16 \cdot 2^i)$, and outputs $\boldsymbol{\alpha}^{(j+1)}$ of size $(2^{k/2-i-1}, 2^{k/2-i}, 24 \cdot 2^i, 24 \cdot 2^i)$;

For $j = (2i + 1), i = 0, 1, ..., k - 1$, the $j$-th Schur step merges subdomains in the horizontal direction.

For $j = (2i + 2), i = 0, 1, ..., k - 1$, the $j$-th Schur step merges subdomains in the vertical direction.

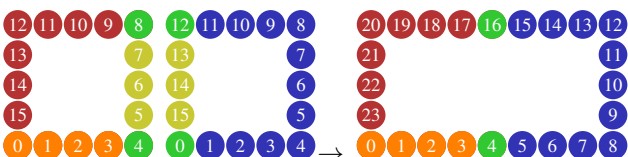

*Figure 29.* Figure 4: Schur step collapses subdomains P and Q into D.

**Horizontal merge**   Divide the nodes in P into contiguous subsets $\boldsymbol{\alpha}, \boldsymbol{\beta}, \boldsymbol{\gamma}, \boldsymbol{\delta}, \boldsymbol{\epsilon}$, such that

$$\boldsymbol{\alpha} = [0, 1, 2, 3], \boldsymbol{\beta} = [4], \boldsymbol{\gamma} = [5, 6, 7], \boldsymbol{\delta} = [8], \boldsymbol{\epsilon} = [9, 10, 11, 12, 13, 14, 15]. \tag{71}$$

Divide the nodes in Q into contiguous subsets $\boldsymbol{\kappa}, \boldsymbol{\lambda}, \boldsymbol{\mu}, \boldsymbol{\nu}$, such that:

$$\boldsymbol{\kappa} = [0], \boldsymbol{\lambda} = [1, 2, 3, 4, 5, 6, 7, 8, 9, 10, 11], \boldsymbol{\mu} = [12], \boldsymbol{\nu} = [13, 14, 15]. \tag{72}$$

Under the indexing in Figure 4, D's boundary consists of the nodes corresponding to $\boldsymbol{\alpha}, \boldsymbol{\beta}, \boldsymbol{\lambda}, \boldsymbol{\delta}, \boldsymbol{\epsilon}$. $\boldsymbol{\beta}, \boldsymbol{\kappa}$ represent the same node after merging, so are $\boldsymbol{\delta}, \boldsymbol{\mu}$.

It makes sense to define their indices in the merged domain.

$$\hat{\boldsymbol{\alpha}} = [0, 1, 2, 3], \hat{\boldsymbol{\beta}} = [4], \hat{\boldsymbol{\lambda}} = [5, 6, 7, 8, 9, 10, 11, 12, 13, 14, 15], \hat{\boldsymbol{\delta}} = [16], \hat{\boldsymbol{\epsilon}} = [17, 18, 19, 20, 21, 22, 23]. \tag{73}$$

Note that the values of these indices depend on the shape of the patches: we only give their values in the first Schur step.

The node grouping implies partitioning the matrix $\mathbf{P}, \mathbf{Q}$ into submatrices, by dividing the rows and columns into subsets.

$$
\left(
\begin{bmatrix}
\begin{bmatrix}
\mathbf{P}_{\alpha\alpha} & \mathbf{P}_{\alpha\beta} & \mathbf{0}_{\alpha\lambda} & \mathbf{P}_{\alpha\delta} & \mathbf{P}_{\alpha\epsilon} \\
\mathbf{P}_{\beta\alpha} & \mathbf{P}_{\beta\beta} & \mathbf{0}_{\beta\lambda} & \mathbf{P}_{\beta\delta} & \mathbf{P}_{\beta\epsilon} \\
\mathbf{0}_{\lambda\alpha} & \mathbf{0}_{\lambda\beta} & \mathbf{0}_{\lambda\lambda} & \mathbf{0}_{\lambda\delta} & \mathbf{0}_{\lambda\epsilon} \\
\mathbf{P}_{\delta\alpha} & \mathbf{P}_{\delta\beta} & \mathbf{0}_{\delta\lambda} & \mathbf{P}_{\delta\delta} & \mathbf{P}_{\delta\epsilon} \\
\mathbf{P}_{\epsilon\alpha} & \mathbf{P}_{\epsilon\beta} & \mathbf{0}_{\epsilon\lambda} & \mathbf{P}_{\epsilon\delta} & \mathbf{P}_{\epsilon\epsilon}
\end{bmatrix} &
\begin{bmatrix}
\mathbf{P}_{\alpha\gamma} \\
\mathbf{P}_{\beta\gamma} \\
\mathbf{0}_{\lambda\gamma} \\
\mathbf{P}_{\delta\gamma} \\
\mathbf{P}_{\epsilon\gamma}
\end{bmatrix} \\
\begin{bmatrix}
\mathbf{P}_{\gamma\alpha} & \mathbf{P}_{\gamma\beta} & \mathbf{0}_{\gamma\lambda} & \mathbf{P}_{\gamma\delta} & \mathbf{P}_{\gamma\epsilon}
\end{bmatrix} &
\begin{bmatrix}
\mathbf{P}_{\gamma\gamma}
\end{bmatrix}
\end{bmatrix}
\right.
$$

$$
\left.
+
\begin{bmatrix}
\begin{bmatrix}
\mathbf{0}_{\alpha\alpha} & \mathbf{0}_{\alpha\beta} & \mathbf{0}_{\alpha\lambda} & \mathbf{0}_{\alpha\delta} & \mathbf{0}_{\alpha\epsilon} \\
\mathbf{0}_{\beta\alpha} & \mathbf{Q}_{\kappa\kappa} & \mathbf{Q}_{\kappa\lambda} & \mathbf{Q}_{\kappa\mu} & \mathbf{0}_{\beta\epsilon} \\
\mathbf{0}_{\lambda\alpha} & \mathbf{Q}_{\lambda\kappa} & \mathbf{Q}_{\lambda\lambda} & \mathbf{Q}_{\lambda\mu} & \mathbf{0}_{\lambda\epsilon} \\
\mathbf{0}_{\delta\alpha} & \mathbf{Q}_{\mu\kappa} & \mathbf{Q}_{\mu\lambda} & \mathbf{Q}_{\mu\mu} & \mathbf{0}_{\delta\epsilon} \\
\mathbf{0}_{\epsilon\alpha} & \mathbf{0}_{\epsilon\beta} & \mathbf{0}_{\epsilon\lambda} & \mathbf{0}_{\epsilon\delta} & \mathbf{0}_{\epsilon\epsilon}
\end{bmatrix} &
\begin{bmatrix}
\mathbf{0}_{\alpha\gamma} \\
\mathbf{Q}_{\kappa\nu}\mathbf{J} \\
\mathbf{Q}_{\lambda\nu}\mathbf{J} \\
\mathbf{Q}_{\mu\nu}\mathbf{J} \\
\mathbf{0}_{\epsilon\gamma}
\end{bmatrix} \\
\begin{bmatrix}
\mathbf{0}_{\gamma\alpha} & \mathbf{J}^{\mathsf{T}}\mathbf{Q}_{\nu\kappa} & \mathbf{J}^{\mathsf{T}}\mathbf{Q}_{\nu\lambda} & \mathbf{J}^{\mathsf{T}}\mathbf{Q}_{\nu\mu} & \mathbf{0}_{\gamma\epsilon}
\end{bmatrix} &
\begin{bmatrix}
\mathbf{J}^{\mathsf{T}}\mathbf{Q}_{\nu\nu}\mathbf{J}
\end{bmatrix}
\end{bmatrix}
\right)
\begin{bmatrix}
\mathbf{u}_{\alpha} \\
\mathbf{u}_{\beta} \\
\mathbf{u}_{\lambda} \\
\mathbf{u}_{\delta} \\
\mathbf{u}_{\epsilon} \\
\mathbf{u}_{\gamma}
\end{bmatrix}
=
\begin{bmatrix}
\mathbf{p}_{\alpha} \\
\mathbf{p}_{\beta} + \mathbf{q}_{\kappa} \\
\mathbf{q}_{\lambda} \\
\mathbf{p}_{\delta} + \mathbf{q}_{\mu} \\
\mathbf{p}_{\epsilon} \\
\mathbf{p}_{\gamma} + \mathbf{J}^{\mathsf{T}}\mathbf{q}_{\nu}
\end{bmatrix}
\tag{74}
$$

is the linear system for the joint domain D. Since $\beta, \kappa$ represent the same node, they correspond to the same row/column; the same applies to $\delta, \mu$. Note that $\gamma$ represents the same set of nodes as $\nu$ but in reverse order. Let $\mathbf{J} \equiv \mathbf{J}^{\mathsf{T}}$ to be the reverse permutation matrix (see §E), whose action is to reverse the rows (or columns) of a matrix when being multiplied with from left (or right).

### E.5. Final algorithm

Algorithm 1 specifies the overall algorithm with both the numerical factorization and back substitution stages.

**The numerical factorization variant** can be implemented by only keeping the computations involving $\boldsymbol{\alpha}$, and returns and saves the values of $\boldsymbol{\alpha}^{(0)}, \boldsymbol{\alpha}^{(1)}, ..., \boldsymbol{\alpha}^{(k)}$ for the use of the back substitution step.

**The back substitution variant** can be realized by only keeping the lines involving $\boldsymbol{\beta}, \boldsymbol{\chi}$, and using the $\boldsymbol{\alpha}^{(0)}, \boldsymbol{\alpha}^{(1)}, ..., \boldsymbol{\alpha}^{(k)}$ cached in the numerical factorization step.

---

**Algorithm 4** The parallel Schur step, $j$-th step, $j = (2i + 1)$. Forward pass. Horizontal.

---

**Require:** $\boldsymbol{\alpha}^{(j)}$ of size $(2^{k/2-i}, 2^{k/2-i}, 16 \cdot 2^i, 16 \cdot 2^i)$,
**Require:** $\boldsymbol{\beta}^{(j)}$ of size $(2^{k/2-i}, 2^{k/2-i}, 16 \cdot 2^i, 1)$.
  $\mathbf{P}, \mathbf{Q}$ are fetched in batches from $\boldsymbol{\alpha}^{(j)}$ for all red and blue subdomains.

$$\mathbf{P} := \boldsymbol{\alpha}^{(j)}[0 :: 2, :, :, :], \quad \mathbf{p} := \boldsymbol{\beta}^{(j)}[0 :: 2, :, :, :] \tag{75}$$

$$\mathbf{Q} := \boldsymbol{\alpha}^{(j)}[1 :: 2, :, :, :], \quad \mathbf{q} := \boldsymbol{\beta}^{(j)}[1 :: 2, :, :, :] \tag{76}$$

$$\begin{bmatrix} \mathbf{P}_{\alpha\alpha} & \mathbf{P}_{\alpha\beta} & \mathbf{P}_{\alpha\gamma} & \mathbf{P}_{\alpha\delta} & \mathbf{P}_{\alpha\epsilon} \\ \mathbf{P}_{\beta\alpha} & \mathbf{P}_{\beta\beta} & \mathbf{P}_{\beta\gamma} & \mathbf{P}_{\beta\delta} & \mathbf{P}_{\beta\epsilon} \\ \mathbf{P}_{\gamma\alpha} & \mathbf{P}_{\gamma\beta} & \mathbf{P}_{\gamma\gamma} & \mathbf{P}_{\gamma\delta} & \mathbf{P}_{\gamma\epsilon} \\ \mathbf{P}_{\delta\alpha} & \mathbf{P}_{\delta\beta} & \mathbf{P}_{\delta\gamma} & \mathbf{P}_{\delta\delta} & \mathbf{P}_{\delta\epsilon} \\ \mathbf{P}_{\epsilon\alpha} & \mathbf{P}_{\epsilon\beta} & \mathbf{P}_{\epsilon\gamma} & \mathbf{P}_{\epsilon\delta} & \mathbf{P}_{\epsilon\epsilon} \end{bmatrix} := \mathbf{P}, \quad \begin{bmatrix} \mathbf{p}_{\alpha} \\ \mathbf{p}_{\beta} \\ \mathbf{p}_{\gamma} \\ \mathbf{p}_{\delta} \\ \mathbf{p}_{\epsilon} \end{bmatrix} := \mathbf{p}, \tag{77}$$

$$\begin{bmatrix} \mathbf{Q}_{\kappa\kappa} & \mathbf{Q}_{\kappa\lambda} & \mathbf{Q}_{\kappa\mu} & \mathbf{Q}_{\kappa\nu} \\ \mathbf{Q}_{\lambda\kappa} & \mathbf{Q}_{\lambda\lambda} & \mathbf{Q}_{\lambda\mu} & \mathbf{Q}_{\lambda\nu} \\ \mathbf{Q}_{\mu\kappa} & \mathbf{Q}_{\mu\lambda} & \mathbf{Q}_{\mu\mu} & \mathbf{Q}_{\mu\nu} \\ \mathbf{Q}_{\nu\kappa} & \mathbf{Q}_{\nu\lambda} & \mathbf{Q}_{\nu\mu} & \mathbf{Q}_{\nu\nu} \end{bmatrix} := \mathbf{Q}, \quad \begin{bmatrix} \mathbf{q}_{\kappa} \\ \mathbf{q}_{\lambda} \\ \mathbf{q}_{\mu} \\ \mathbf{q}_{\nu} \end{bmatrix} := \mathbf{q}. \tag{78}$$

$$\mathbf{X} := \begin{bmatrix} \mathbf{P}_{\alpha\alpha} & \mathbf{P}_{\alpha\beta} & \mathbf{0}_{\alpha\lambda} & \mathbf{P}_{\alpha\delta} & \mathbf{P}_{\alpha\epsilon} \\ \mathbf{P}_{\beta\alpha} & \mathbf{P}_{\beta\beta} + \mathbf{Q}_{\kappa\kappa} & \mathbf{Q}_{\kappa\lambda} & \mathbf{P}_{\beta\delta} + \mathbf{Q}_{\kappa\mu} & \mathbf{P}_{\beta\epsilon} \\ \mathbf{0}_{\lambda\alpha} & \mathbf{Q}_{\lambda\kappa} & \mathbf{Q}_{\lambda\lambda} & \mathbf{Q}_{\lambda\mu} & \mathbf{0}_{\lambda\epsilon} \\ \mathbf{P}_{\delta\alpha} & \mathbf{P}_{\delta\beta} + \mathbf{Q}_{\mu\kappa} & \mathbf{Q}_{\mu\lambda} & \mathbf{P}_{\delta\delta} + \mathbf{Q}_{\mu\mu} & \mathbf{P}_{\delta\epsilon} \\ \mathbf{P}_{\epsilon\alpha} & \mathbf{P}_{\epsilon\beta} & \mathbf{0}_{\epsilon\lambda} & \mathbf{P}_{\epsilon\delta} & \mathbf{P}_{\epsilon\epsilon} \end{bmatrix} \tag{79}$$

$$\mathbf{Y} := \begin{bmatrix} \mathbf{P}_{\alpha\gamma} \\ \mathbf{P}_{\beta\gamma} + \mathbf{Q}_{\kappa\nu}\mathbf{J} \\ \mathbf{Q}_{\lambda\nu}\mathbf{J} \\ \mathbf{P}_{\delta\gamma} + \mathbf{Q}_{\mu\nu}\mathbf{J} \\ \mathbf{P}_{\epsilon\gamma} \end{bmatrix} \quad \mathbf{y} := \begin{bmatrix} \mathbf{p}_{\alpha} \\ \mathbf{p}_{\beta} + \mathbf{q}_{\kappa} \\ \mathbf{q}_{\lambda} \\ \mathbf{p}_{\delta} + \mathbf{q}_{\mu} \\ \mathbf{p}_{\epsilon} \end{bmatrix} \tag{80}$$

$$\mathbf{Z} := \begin{bmatrix} \mathbf{P}_{\gamma\alpha} & \mathbf{P}_{\gamma\beta} + \mathbf{J}^{\mathsf{T}}\mathbf{Q}_{\nu\kappa} & \mathbf{J}^{\mathsf{T}}\mathbf{Q}_{\nu\lambda} & \mathbf{P}_{\gamma\delta} + \mathbf{J}^{\mathsf{T}}\mathbf{Q}_{\nu\mu} & \mathbf{P}_{\gamma\epsilon} \end{bmatrix} \tag{81}$$

$$\mathbf{W} := \begin{bmatrix} \mathbf{P}_{\gamma\gamma} + \mathbf{J}^{\mathsf{T}}\mathbf{Q}_{\nu\nu}\mathbf{J} \end{bmatrix} \quad \mathbf{w} := \begin{bmatrix} \mathbf{p}_{\gamma} + \mathbf{J}^{\mathsf{T}}\mathbf{q}_{\nu} \end{bmatrix} \tag{82}$$

$$\mathbf{D} := \left( \mathbf{X} - \mathbf{Y}\mathbf{W}^{-1}\mathbf{Z} \right) \quad \mathbf{d} := \mathbf{y} - \mathbf{Y}\mathbf{W}^{-1}\mathbf{w} \tag{83}$$

$$\boldsymbol{\alpha} := \mathbf{D}, \quad \boldsymbol{\beta} := \mathbf{d} \tag{84}$$

  **return** $\boldsymbol{\alpha}^{(j+1)}$ of size $(2^{k/2-i-1}, 2^{k/2-i}, 24 \cdot 2^i, 24 \cdot 2^i)$,
$\boldsymbol{\beta}^{(j+1)}$ of size $(2^{k/2-i-1}, 2^{k/2-i}, 24 \cdot 2^i, 1)$

---

---

**Algorithm 5** The parallel $j$-th Schur step, $j = (2i + 1)$. Backward pass. Horizontal.

---

**Require:** $\boldsymbol{\chi}^{(j+1)}$ of size $(2^{k/2-i}, 2^{k/2-i}, 24 \cdot 2^i, 1)$

$$
\begin{bmatrix} \mathbf{u}_{\hat{\boldsymbol{\alpha}}} \\ \mathbf{u}_{\hat{\boldsymbol{\beta}}} \\ \mathbf{u}_{\hat{\boldsymbol{\lambda}}} \\ \mathbf{u}_{\hat{\boldsymbol{\delta}}} \\ \mathbf{u}_{\hat{\boldsymbol{\epsilon}}} \end{bmatrix} := \boldsymbol{\beta}^{(j)} \text{ or } \quad
\begin{aligned}
\mathbf{u}_{\hat{\boldsymbol{\alpha}}} &:= \boldsymbol{\beta}^{(j)}[:,:,\hat{\boldsymbol{\alpha}},:] \\
\mathbf{u}_{\hat{\boldsymbol{\beta}}} &:= \boldsymbol{\beta}^{(j)}[:,:,\hat{\boldsymbol{\beta}},:] \\
\mathbf{u}_{\hat{\boldsymbol{\lambda}}} &:= \boldsymbol{\beta}^{(j)}[:,:,\hat{\boldsymbol{\lambda}},:] \\
\mathbf{u}_{\hat{\boldsymbol{\delta}}} &:= \boldsymbol{\beta}^{(j)}[:,:,\hat{\boldsymbol{\delta}},:] \\
\mathbf{u}_{\hat{\boldsymbol{\epsilon}}} &:= \boldsymbol{\beta}^{(j)}[:,:,\hat{\boldsymbol{\epsilon}},:]
\end{aligned}
\tag{85}
$$

$$
\mathbf{x} := \begin{bmatrix} \mathbf{u}_{\hat{\boldsymbol{\alpha}}} \\ \mathbf{u}_{\hat{\boldsymbol{\beta}}} \\ \mathbf{u}_{\hat{\boldsymbol{\lambda}}} \\ \mathbf{u}_{\hat{\boldsymbol{\delta}}} \\ \mathbf{u}_{\hat{\boldsymbol{\epsilon}}} \end{bmatrix}
\tag{86}
$$

$$
\boldsymbol{\chi}^{(j)} := \text{ZEROS}(2^{k/2-i}, 2^{k/2-i}, 16 \cdot 2^i, 1).
\tag{87}
$$

$$
\mathbf{u}^{\boldsymbol{\gamma}} := \mathbf{W}^{-1}(\mathbf{w} - \mathbf{Z}\mathbf{x}) \quad \text{(back-substitution)},
\tag{88}
$$

where we use upper script to emphasize that it does not correspond to slice into a vector.

$$
\boldsymbol{\chi}^{(j)}[0::2,:,:,:] := \begin{bmatrix} \mathbf{u}_{\hat{\boldsymbol{\alpha}}} \\ \mathbf{u}_{\hat{\boldsymbol{\beta}}} \\ \mathbf{u}^{\boldsymbol{\gamma}} \\ \mathbf{u}_{\hat{\boldsymbol{\delta}}} \\ \mathbf{u}_{\hat{\boldsymbol{\epsilon}}} \end{bmatrix} \text{ or }
\begin{aligned}
\boldsymbol{\chi}^{(j)}[0::2,:,\boldsymbol{\alpha},:] &:= \mathbf{u}_{\hat{\boldsymbol{\alpha}}} \\
\boldsymbol{\chi}^{(j)}[0::2,:,\boldsymbol{\beta},:] &:= \mathbf{u}_{\hat{\boldsymbol{\beta}}} \\
\boldsymbol{\chi}^{(j)}[0::2,:,\boldsymbol{\gamma},:] &:= \mathbf{u}^{\boldsymbol{\nu}} \\
\boldsymbol{\chi}^{(j)}[0::2,:,\boldsymbol{\delta},:] &:= \mathbf{u}_{\hat{\boldsymbol{\delta}}} \\
\boldsymbol{\chi}^{(j)}[0::2,:,\boldsymbol{\epsilon},:] &:= \mathbf{u}_{\hat{\boldsymbol{\epsilon}}}
\end{aligned} ,
\begin{aligned}
\boldsymbol{\chi}^{(j)}[1::2,:,\boldsymbol{\kappa},:] &:= \mathbf{u}_{\hat{\boldsymbol{\beta}}} \\
\boldsymbol{\chi}^{(j)}[1::2,:,\boldsymbol{\lambda},:] &:= \mathbf{u}_{\hat{\boldsymbol{\lambda}}} \\
\boldsymbol{\chi}^{(j)}[1::2,:,\boldsymbol{\mu},:] &:= \mathbf{u}_{\hat{\boldsymbol{\delta}}} \\
\boldsymbol{\chi}^{(j)}[1::2,:,\boldsymbol{\nu},:] &:= \mathbf{J}^{\intercal}\mathbf{u}^{\boldsymbol{\gamma}}
\end{aligned}
\tag{89}
$$

**return** $\boldsymbol{\chi}^{(j)}$ of size $(2^{k/2-i}, 2^{k/2-i}, 16 \cdot 2^i, 1)$.

---

### E.6. Illustration on a 3x7 image

**An example with detailed visualization** Let us consider a smaller $3 \times 7$ image for the convenience of illustration, as shown in Figure 24. Each pixel corresponds to a node in the graph and is assigned with an index $i \in \{0, 1, ..., 21 - 1\}$. Two nodes $i$ and $j$, where $i, j \in \{0, 1, ..., 21 - 1\}$, are connected by an edge if the $3 \times 3$ kernel centered at one node will cover the other node. $\mathbf{A}_{ij} = 0$ if node $i$ and node $j$ are not connected by an edge in the graph. Each of the color includes the following subsets of nodes: $\mathbf{r} = [0, 1, 2, 7, 14, 15, 16]$, $\mathbf{s} = [3, 10, 17]$, $\mathbf{t} = [4, 5, 6, 13, 18, 19, 20]$, $\mathbf{a} = [8, 9]$, $\mathbf{b} = [11, 12]$.

The original sparse linear system, $\mathbf{A} \in \mathbb{R}^{21 \times 21}$:

$$\sum_{j=0}^{21-1} \mathbf{A}_{ij} \mathbf{u}_j = \mathbf{v}_i, \forall i \in \{0, 1, ..., 21 - 1\}. \tag{90}$$

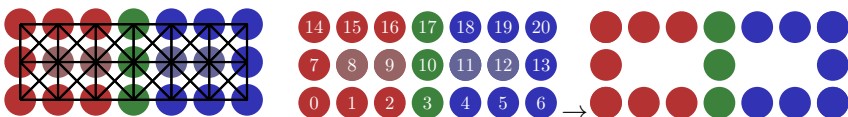

*Figure 30.* The connectivity graph resulted from the use of the $3 \times 3$ kernel size, for a $3 \times 7$ image under the lexicographic sweeping order.

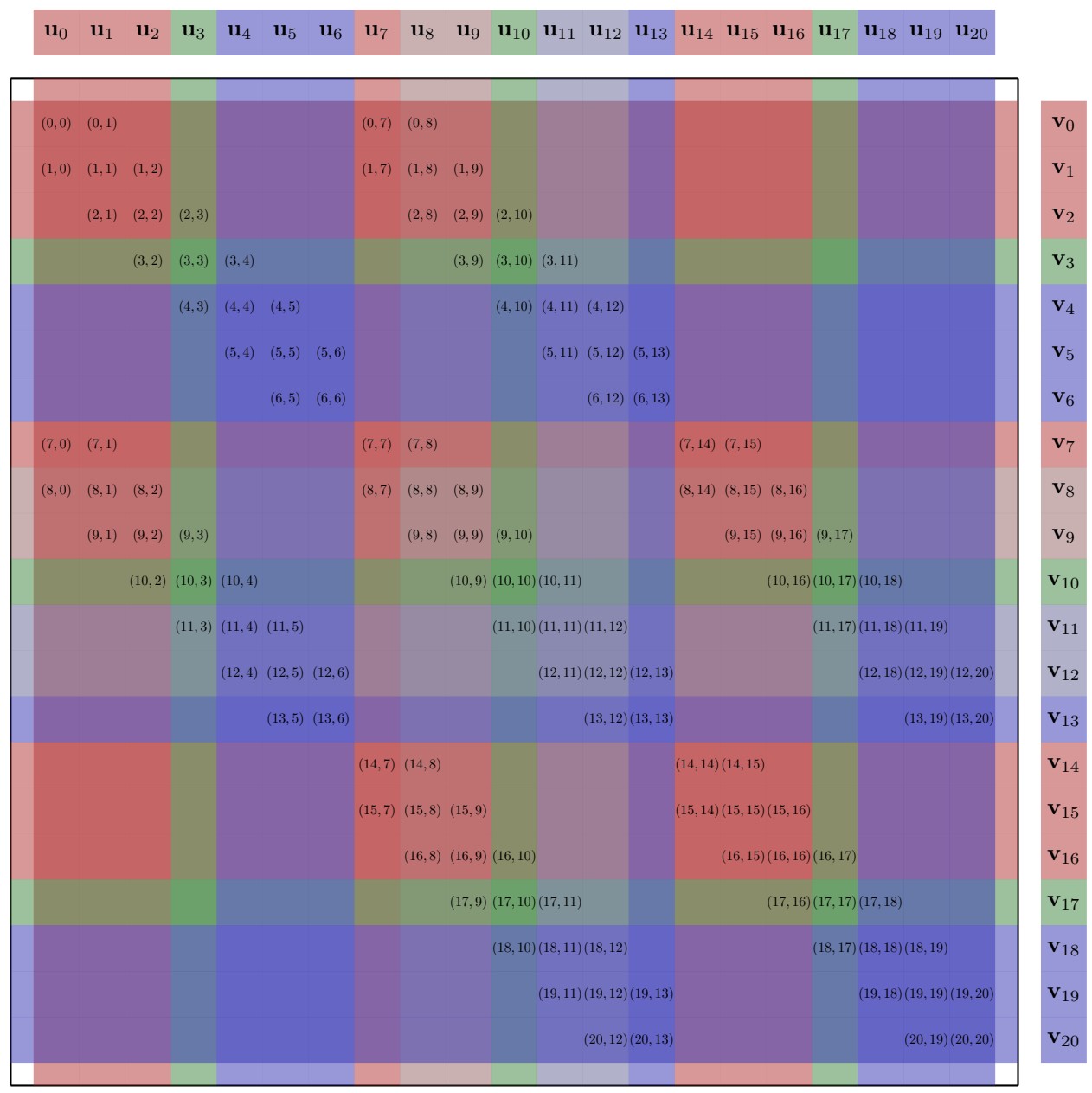

*Figure 31.* Same as Figure 25 but for a $3 \times 7$ image.

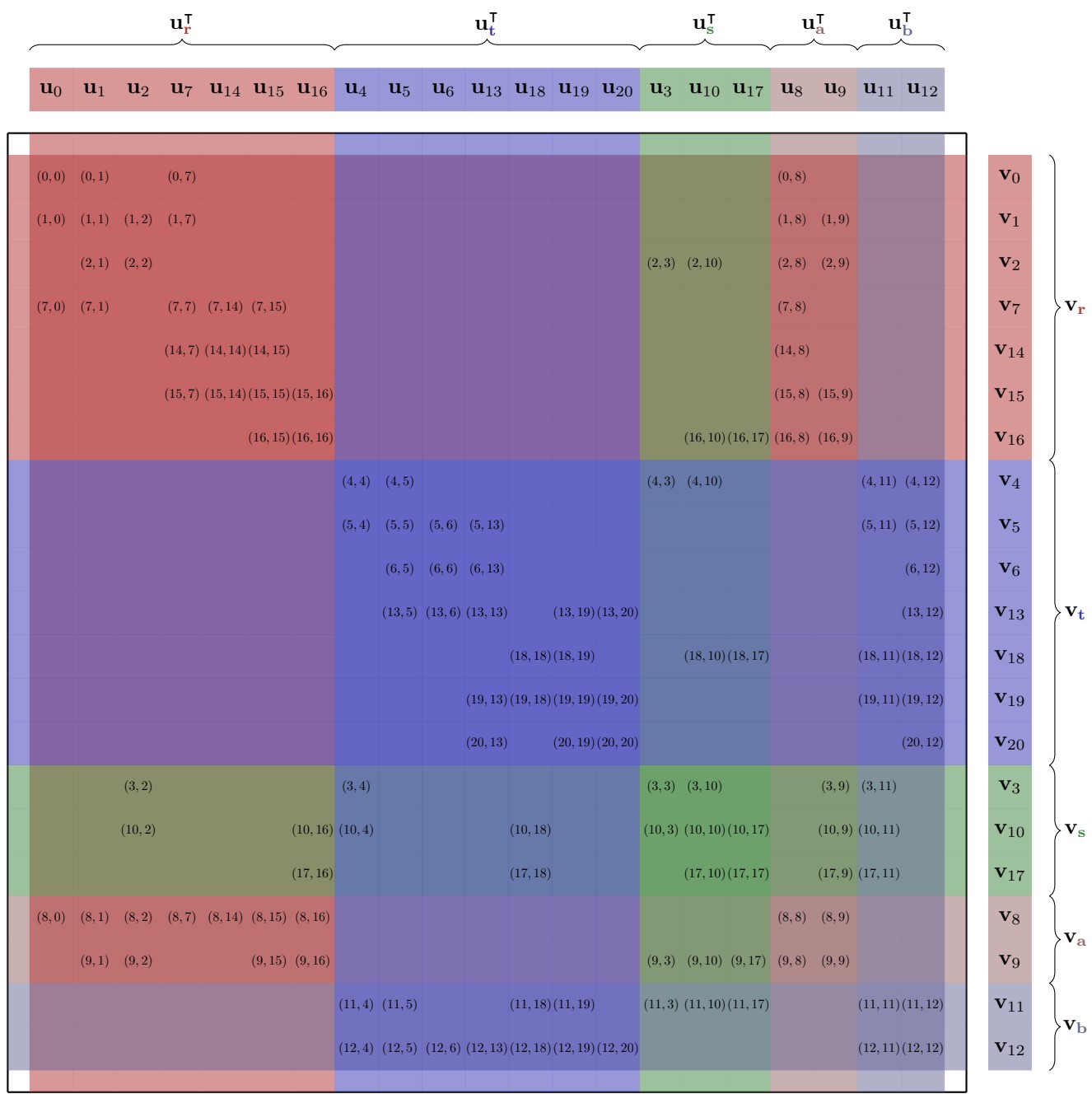

*Figure 32.* Same as Figure 26 but for a $3 \times 7$ image.

# F. Additional Details

Please note that our use of the term "involution" is not related to the mathematical concept that refers to a self-inverse function as an "involution," though deploying our solver in an eigen solver resembles similar concepts. We use "involution" to refer to the exact, generalized, and inverted convolution—generalized in the sense of spatially varying kernels. Our use of "involution" is close to its biomedical meaning—the shrinking or reduction in the size of an organ, but applied to an image.

## F.1. Details on FEM and PDE discretization: addibility of parallel linear elements

Although where the matrix $\mathbf{A}$ or the tensor $\boldsymbol{\alpha}$ comes from does not matter to apply our sparse solver (though the numerical error can depend on $\mathbf{A}$), as long as $\mathbf{A}$ is invertible with the sparsity pattern we have described, we supplement with implementation details of assembling matrix $\mathbf{A}$ for reproducibility. Despite that theoretically one could use, e.g., bilinear bases as well, in this paper, we use the first-order piecewise-linear finite element method (Wang et al., 2023) to discretize the partial differential equations. There are multiple advantages: 1) The discretized system preserved certain algebraic properties of the continuous PDEs that are particularly valuable in geometry tasks (Wang et al., 2023). 2) Most importantly, the **addibility of parallel linear elements**. It is easy to verify that if each of the patch $(i, j)$ puts in $\boldsymbol{\alpha}^{(*)}[i, j, :, :]$ the Laplacian matrix $\mathbf{L}^{(i,j)}$ that is discretized with the Neumann boundary condition, then **assembling together these patch-wise sub-matrices yields the Laplacian matrix L for the whole image domains**, under the same Neumann boundary condition. Here "assembling patch-wise sub-matrices" means summing together the sub-matrices to a large sparse matrix $\mathbf{L}$ by indexing into the rows and columns of $\mathbf{L}$ that the patch nodes correspond to. In other words, we know it holds that:

$$\mathbf{L} \equiv \sum_{(i,j)} (\mathbf{J}^{(i,j)})^{\mathsf{T}} \mathbf{L}^{(i,j)} \mathbf{J}^{(i,j)} \in \mathbb{R}^{n \times n} \tag{91}$$

where $\mathbf{L}^{(i,j)} \in \mathbb{R}^{25 \times 25}$ comes from the Laplacian matrix discretizing an elliptic PDE with the Neumann boundary condition, and $\mathbf{L} \in \mathbb{R}^{n \times n}$ comes from discretizing the same PDE & boundary condition but over the entire image domain. Here $\mathbf{J}^{(i,j)} \in \mathbb{R}^{25 \times n}$ is the binary matrix, playing the role of putting the patch sub-system $\mathbf{L}^{(i,j)}$ in the correct rows and columns of the large sparse system.

The fact can be verified by noticing that using first-order piecewise-linear FEM (Wang et al., 2023) to discretize the elliptic PDE (3) which stores the anisotropic tensor $\mathbf{C}(\mathbf{x})$ as a $2 \times 2$ matrix per triangle, each triangle will contribute to the entries of $\mathbf{A}$ independently using information only within that triangle. The fact that patch sub-systems can be "seamlessly" added together thanks to the use of the Neumann boundary condition—the *natural boundary condition*. Using certain high-order finite element methods might necessitate information exchange between adjacent patches, which is still doable but requires extra care in the implementation.

**A double covered FEM mesh layout for images.** During the tessellation of the regular grid as triangular meshes, to avoid bias in choosing the orientation of the long edge of the triangles, we simply use a double covered triangular mesh—each square has four cross-edge triangles whose edges connect pixels.

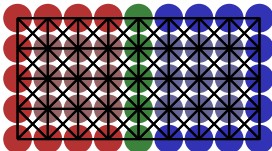

*Figure 33.* Our double-covered piecewise-linear triangle mesh layout.

## F.2. Issues with incorporating iterative solvers in deep learning

Our method is a **S**parse solver that is **C**onsistent-performance, **H**yperspeed, in-the-**W**ild, **A**ccurate, **R**obust, and **Z**ero-parameter. In this section, we explain why indirect, a.k.a. iterative, solvers fall short of these goals. We identify a key obstacle preventing neural architectures from adopting linear solvers is the lack of a method like ours.

Deploying iterative solvers appropriately comes with a tedious workflow, requiring users in the loop with PhD-level expertise in numerical analysis, PDEs, GPU optimization, or image processing. Iterative solvers require problem-specific solvers with many parameters, preconditioners which also have parameters. Although solvers such as LSMR and LSQR seem to

```
1  {
2      "config_version": 2,
3      "solver": {
4          "scope": "main",
5          "print_grid_stats": 1,
6          "store_res_history": 1,
7          "solver": "FGMRES",
8          "print_solve_stats": 1,
9          "obtain_timings": 1,
10         "preconditioner": {
11             "interpolator": "D2",
12             "print_grid_stats": 1,
13             "aggressive_levels": 1,
14             "solver": "AMG",
15             "smoother": {
16                 "relaxation_factor": 1,
17                 "scope": "jacobi",
18                 "solver": "JACOBI_L1"
19             },
20             "presweeps": 1,
21             "selector": "HMIS",
22             "coarsest_sweeps": 1,
23             "coarse_solver": "NOSOLVER",
24             "max_iters": 1,
25             "max_row_sum": 0.9,
26             "strength_threshold": 0.25,
27             "min_coarse_rows": 2,
28             "scope": "amg_solver",
29             "max_levels": 50,
30             "cycle": "V",
31             "postsweeps": 1
32         },
33         "max_iters": 100,
34         "monitor_residual": 1,
35         "gmres_n_restart": 10,
36         "convergence": "RELATIVE_INI",
37         "tolerance" : 1e-06,
38         "norm": "L2"
39     }
40 }
```

*Figure 34.* The FGMRES+AMG solver (with HMIS selector) in Table 3.

be applicable to all matrices, especially when it comes to random deconvolution, we observe that they can be inefficient compared to our method in §4.

The large diversity of possible PDEs makes it difficult for one user to master all the methods well and implement the entire arsenal of numerical PDEs. Especially, we lack an automatic pipeline to analyze the type of $\mathbf{A}$ to deploy the appropriate iterative solvers, and to automatically set the parameters, preconditioner, and the parameters of the preconditioner for good performance by analyzing entries of $\mathbf{A}$. In the setup of PDE learning, the exact form of PDE is unknown. To algorithmically search for the PDE (that is, the matrix A represents it), one will have to implement iterative solvers for *all* possible PDEs, which is an infeasible task for unstudied PDEs. Iterative solvers can perform poorly for highly singularly, stiff, anisotropic coefficients, or oscillatory solutions such as the Helmholtz equation under a relatively large frequency (Ernst & Gander, 2011).

Even restricting to solving some well-studied PDEs, so there are efficient iterative numerical schemes, the optimal values of these parameters also depend on the entries and properties of $\mathbf{A}$. The many parameters in iterative solvers must be appropriately set to yield a correct result: sigma shift values, preconditioning schemes, and parameters such as error tolerance, maximum iterations. For example, Figures 34, 35 and 36 are the FGMRES+AMG and GMRES+AMG, and PCG+AMG solvers we test in Table 3, under the Nvidia AMGX framework (Naumov et al., 2015).

```
1  {
2      "config_version": 2,
3      "solver": {
4          "preconditioner": {
5              "print_grid_stats": 1,
6              "print_vis_data": 0,
7              "solver": "AMG",
8              "smoother": {
9                  "scope": "jacobi",
10                 "solver": "BLOCK_JACOBI",
11                 "monitor_residual": 0,
12                 "print_solve_stats": 0
13             },
14             "print_solve_stats": 0,
15             "presweeps": 1,
16             "interpolator": "D2",
17             "max_iters": 1,
18             "monitor_residual": 0,
19             "store_res_history": 0,
20             "scope": "amg",
21             "max_levels": 50,
22             "cycle": "V",
23             "postsweeps": 1
24         },
25         "solver": "PCG",
26         "print_solve_stats": 1,
27         "obtain_timings": 1,
28         "max_iters": 100,
29         "monitor_residual": 1,
30         "convergence": "RELATIVE_INI",
31         "scope": "main",
32         "tolerance" : 1e-06,
33         "norm": "L2"
34     }
35 }
```

*Figure 35.* The PCG+AMG solver in Table 3.

```
1  {
2      "config_version": 2,
3      "determinism_flag": 1,
4      "exception_handling" : 1,
5      "solver": {
6          "scope": "main",
7          "print_grid_stats": 1,
8          "store_res_history": 1,
9          "solver": "GMRES",
10         "print_solve_stats": 1,
11         "obtain_timings": 1,
12         "preconditioner": {
13             "interpolator": "D2",
14             "print_grid_stats": 1,
15             "solver": "AMG",
16             "smoother": "JACOBI_L1",
17             "presweeps": 2,
18             "selector": "PMIS",
19             "coarsest_sweeps": 2,
20             "coarse_solver": "NOSOLVER",
21             "max_iters": 1,
22             "interp_max_elements": 4,
23             "min_coarse_rows": 2,
24             "scope": "amg_solver",
25             "max_levels": 50,
26             "cycle": "V",
27             "postsweeps": 2
28         },
29         "max_iters": 100,
30         "monitor_residual": 1,
31         "gmres_n_restart": 10,
32         "convergence": "RELATIVE_INI",
33         "tolerance" : 1e-06,
34         "norm": "L2"
35     }
36 }
```

*Figure 36.* The GMRES+AMG solver in Table 3.

That said, even to solve a single instance of PDE with iterative solvers, the user may have to manually repeat the trial-and-error steps until convergence: tweak the parameters and then run the iterative solver with them; if not converged or the result is not desirable, the user has to go back to change the parameters. The tedious user-in-the-loop workflows prevent training with a solver in the loop on a massive amount of data in scientific machine learning in a way similar to the practice of computer vision and natural language processing.

Unpredictable runtime: When the input problem changes, the runtime can change dramatically, which is a critical drawback for interactive applications such as self-driving cars or other time-sensitive scenarios. Conservative parameter settings significantly slow down iterative solvers, and aggressive parameter settings risk insufficient iterations or unconverged results.

Catastrophic failures can incur when applying unsuitable iterative solvers: it can obtain totally wrong results to ruin trained models, calling for user intervention: manually fixes for certain training examples and restoration from trained checkpoints.

### F.3. Differentiable sparse solvers: widely desired, yet absent

Here are links to examples of feature requests and discussions on differentiable linear solvers in the community. Despite of the frequent requests, linear solvers have only very limited support in mainstream deep learning packages, which perhaps makes sense because existing sparse linear solvers are quite limited in applicability, and inefficient compared to other differentiable modules and our method. Naïvely porting existing solvers to, e.g., PyTorch, makes them the sole bottleneck in the training pipelines. For example, JAX has a sparse solver, which is CPU-only and wraps up a naïve callback of SciPy's "spsolve" function. ~~PyTorch still does not have a sparse linear solver until Oct. 2024 (version 2.4).~~ Concurrent to the development of our work, since version 2.5, PyTorch recently starts to supporting a sparse solver backended by cuDSS (which requires compilation from the source; as of August 2025, the pip binary release of PyTorch does not ship with the cuDSS solver). However, the cuDSS solver is much slower than our method, as demonstrated throughout the paper.

https://github.com/pytorch/pytorch/issues/58828,

https://github.com/pytorch/pytorch/issues/108977,

https://github.com/pytorch/pytorch/issues/69538,

https://discuss.pytorch.org/t/linear-solver-for-sparse-matrices/200289/6,

https://github.com/tensorflow/addons/pull/2396,

https://github.com/tensorflow/addons/issues/2387,

https://github.com/tensorflow/tensorflow/issues/27380,

https://discuss.pytorch.org/t/solving-ax-b-for-sparse-tensors-preferably-with-backward/102331,

https://discuss.pytorch.org/t/differentiable-sparse-linear-solver-with-cupy-backend-unsupported-tensor-layout-sparse-in-gradcheck/141309.

https://github.com/jax-ml/jax/discussions/18452

