# OpenReview forum: "Schwarz–Schur Involution: Lightspeed Differentiable Sparse Linear Solvers"
_ICML.cc/2025/Conference — ICML 2025 poster_

### Official Review · Reviewer_QcrL · 2025-03-13

**Overall Recommendation:** 4

**Summary:**

This paper investigates efficient methods for solving sparse linear equations that commonly arise in applications related to partial differential equations (PDEs) and convolutional neural networks. The key insight of the study is the exploitation of hidden structures within convolutional kernels, allowing the division of an image into small, independent patches. Within each patch, computations can be performed independently by leveraging the concept of Gaussian elimination.

This technique follows a divide-and-conquer approach, where the problem is progressively broken down into smaller subproblems. The solution process involves solving these subproblems independently and then performing back-substitution from the final step to the initial step. By structuring the problem in this manner, the authors effectively harness the parallel computing capabilities of GPUs to handle multiple small but dense square matrices efficiently.

## update after rebuttal

I want to thank the authors for updating the codebase to include a comparison with scipy's direct solve method. I can verify scipy is significantly slower than the proposed method.

However, I also implemented indirect method using cupy+GPU, here are the preliminary results:

```
# GPU sparse solve alternative
A_cupy_sparse = cp.sparse.csr_matrix(A_scipy) # Convert SciPy sparse to CuPy sparse
lb_cupy = cp.asarray(lb.flatten()) # Keep RHS as dense CuPy array
x_cupy_sparse = None
print("Timing CuPy sparse solve (with synchronization):")
for _ in range(5):
  time_start_sparse = time.time()
  # x_cupy_sparse = cupyx.scipy.sparse.linalg.spsolve(A_cupy_sparse, lb_cupy) # method 1, 89 second, norm difference with gsol is 0.955
  # x_cupy_sparse = cupyx.scipy.sparse.linalg.splu(A_cupy_sparse).solve(lb_cupy) # method 2, 17 seconds, norm difference with gsol is 0.005
  # x_cupy_sparse = cupyx.scipy.sparse.linalg.lsqr(A_cupy_sparse, lb_cupy) # method 3, 84 second, norm difference with gsol is, nan
  x_cupy_sparse = cupyx.scipy.sparse.linalg.cg(A_cupy_sparse, lb_cupy) # method 4, 0.15 second, norm difference with gsol is 0.085
  # x_cupy_sparse = cupyx.scipy.sparse.linalg.gmres(A_cupy_sparse, lb_cupy) # method 5, 0.54 second, norm difference with gsol is 1.40
  # x_cupy_sparse = cupyx.scipy.sparse.linalg.cgs(A_cupy_sparse, lb_cupy) # method 6, 0.14 second, norm difference with gsol is 0.606
  # x_cupy_sparse = cupyx.scipy.sparse.linalg.minres(A_cupy_sparse, lb_cupy) # method 7, 0.066 seconds, norm difference with gsol is 141.08
  # x_cupy_sparse = cupyx.scipy.sparse.linalg.lsmr(A_cupy_sparse, lb_cupy) # method 8, 14 seconds, norm difference with gsol is 165.35
  cp.cuda.Stream.null.synchronize() # Wait for GPU to finish
  print(f"cupy sparse solve time: {time.time() - time_start_sparse}")
```

For comparison, the proposed method (on GPU) runs in 0.032 seconds. The scipy direct method runs in 11.12 seconds. The norm difference between gsol and x_scipy is 0.021.

Hence, we can see the proposed method is still faster than these cupy indirect solve implementation, but the speedup is now much smaller than anticipated. The accuracy for the indirect method is kind of okay in my stand, since the norm is calculated as np.linalg.norm(x_vec - y_vec), and the vector has length 263169. So, the speedup over indirect method is more around 5-10 I would say.

Also, for this current example image size, transfer A_scipy to the GPU doesn't seem to take that long. Therefore, having the claim of 1000x speedup in the title is \textit{a little bit} of an exaggeration. I would probably add a quantifier in the title saying that this impressive speedup is only for direct solve.

Personally, I still support the acceptance of this paper due to this nice observation/idea of being able to do lots of local parallelization for direct solve, although a little less excited than first reading this work (I thought this was a HUGE breakthrough). My current score is more like 3.5 than 4, but I am going to keep it as 4. For revision, I would strongly suggest the authors should compare thoroughly with indirect cupy methods (as I show above) and also modify the title and abstract to reflect that there is a place for the indirect cupy method.

If the authors can show the speedup of the proposed method over the indirect cupy method is much more significant (2 or 3 orders of magnitude) on large images (2561 x 2561), that would be wonderful. However, since I do not have that information, I cannot champion for something I haven't seen or experimented by myself.

If the authors feel strongly about the advantage of their method on large image instances, they are welcome to update the codebase in the anonymous code repo and send a message to the AC with clear evidence so that AC may check it out.

**Claims And Evidence:**

The claims made in the paper are supported by clear and compelling evidence, particularly in Section 4.2, where the authors use color-coded visualizations and mathematical derivations to illustrate key computational concepts. While the explanations are well-structured, those unfamiliar with convolutional neural networks may find some parts challenging to follow. However, after manually performing the Gaussian elimination steps from Equations (4.2) to (5) and verifying the authors' calculations, the divide-and-conquer approach became evident.

The computational results presented are particularly impressive. Compared to CUDA-based implementations, the proposed method achieves a speedup of two orders of magnitude. Additionally, when compared to SciPy implementations, the authors report a computational speedup exceeding three orders of magnitude. These results strongly support the effectiveness of the proposed approach.

**Essential References Not Discussed:**

N/A

**Experimental Designs Or Analyses:**

I have checked the experimental designs and analyses. It's mainly solving the same sparse linear systems and comparing the running times, which is very straightforward.

**Methods And Evaluation Criteria:**

Yes, the proposed methods and evaluation criteria (running time comparison) are valid for the problem and application.

**Other Comments Or Suggestions:**

Thank you very much for writing this paper. I have enjoyed reading, understanding, and learning from your proposed idea. Below are some writing suggestions:

Clarify the meaning of the color-coded subscripts for matrix A in Section 4.2.1.

Provide a step-by-step breakdown of Gaussian elimination from Equation (4.2) to (5). At least mention that $u_a = A_{aa}^{-1}(v_a - A_{ar} u_r - A_{as} u_s)$.

Improve the caption of Figure 4 to explicitly mention the kernel size (3x3) and its impact on patch selection.

Consider adding a brief explanation or visual aid to illustrate why the green dots incorporate information from all other colored dots while the brown and light blue dots do not incorporate information from all colored dots.

**Other Strengths And Weaknesses:**

The proposed method is both novel and clever, demonstrating significant potential for applications in various domains. However, one notable weakness lies in Section 4, particularly Subsection 4.2, where the methodology is discussed. The explanation lacks sufficient detail, making it difficult for readers unfamiliar with convolutional neural networks to grasp the key ideas immediately. In particular, understanding the notation and subscripts used in the matrices required considerable effort. The color-coded elements in Subsection 4.2.1 are not well explained, which adds to the confusion. Additionally, the process of Gaussian elimination from Equation (4.2) to (5) is not elaborated on sufficiently, requiring extra effort from the reader to reconstruct the steps.

Another point of concern is Figure 4, where the rationale behind dividing patches in the presented manner is unclear. A suggestion for improvement would be to clarify in the caption that the convolutional kernel is 3x3, which influences the selection of points included in the computation. Specifically, when the kernel focuses on a brown dot, it only includes nearby red and green dots while never incorporating the light blue dots. A clearer explanation would help readers understand why certain dots incorporate information while others do not. Adding such clarifications would significantly improve the paper’s clarity and accessibility.

**Questions For Authors:**

1. Regarding Figure 4, is our understanding correct that the kernel size is , and that this is why the brown dots only incorporate information from red, brown, and green dots but not from the light and dark blue dots?

2. In Line 218, what do $A_{SS}^P$ and $A_{SS}^Q$ represent? Do they both equal 1/2 of $A_{SS}$?

3. In Line 678, you discuss the running time of your method but do not mention the running time of the gradient descent method using L-BFGS. How long does L-BFGS take in comparison?

4. In the appendix, Line 749 (right column), you mention that it takes 25 seconds to solve the top-six eigenvalue problem in MATLAB. Which method is this referring to? Is it your proposed method or the baseline? If it is your method, how long does the baseline take to solve the same problem?

5. In Line 813 (right column), you state that matrices are not treated as symmetric, even when they are. Do the baseline methods exploit this prior knowledge of symmetry?

6. In Appendix Section B5, you discuss optimizing matrix A for Problem 11. Could you clarify how this optimization is related to solving the sparse linear system?

7. There is no code submission for this paper. While the overall methodology is clear on a high-level, verification through implementation would be helpful. Would it be possible to provide a sample implementation during rebuttal, specifically for the deconvolution operation shown in Figure 7? An anonymous GitHub repository or similar would greatly enhance reproducibility and allow for further validation of your approach.

**Relation To Broader Scientific Literature:**

Yes, this work is at the intersection of scientific computing and differential equations. Different branches of literature reviewers are discussed extensively in this work.

**Theoretical Claims:**

I have manually checked the Gaussian Elimination step from Equation 4 to Equation 5 and can verify that they are correct.

---

> ### Author Rebuttal · Authors · 2025-03-30
>
> Thanks for the insightful comments and efforts to verify our method & derivation!
>
> We appreciate the detailed feedback on improving clarity, and will be sure to elaborate on full details to reduce readers’ efforts to reconstruct the steps.
>
> Please also refer to Re:eGtJ for extra descriptions, and refer to the Re:D4Wa for background information and the visualizations in https://anonymous.4open.science/r/icml-rebuttal-268F/rebuttal-figures.pdf (Figures 21-24)
> # Clarifications (+D4Wa):
> > Clarify the meaning of the color-coded subscripts for matrix A in Section 4.2.1:
>
> please refer to Figures 22,23 in the anonymous link: each color indicates a group of pixels: r = [0, 1, 2, 7, 14, 15, 16], s = [3, 10, 17], t = [4, 5, 6, 13, 18, 19, 20], a = [8, 9], b = [11, 12], in the case of 3x7 image (Figure 21).
>
> A_rr, A_rs represent the 7x7 and 7x3 submatrices (since the group r, s have 7, 3 nodes, resp.):
> A_rs is the slicing into matrix A that uses r as indices to select rows and s to select columns: A_rs = A[r, s]. A_rr = A [r,r].
>
> See Figure 24 for the definitions of P_rr, P_rs:
> P_rr = Arr − Ara Aaa^{-1} Aar,
> P_rs = Ars − Ara Aaa^{-1} Aas.
>
> P, Q  are 10×10 matrices (since after eliminating nodes in a, b, each of the
> subdomains have 10 nodes remaining, as shown in Figure 21.)
>
> P_rr is a 7x7 matrix, as a submatrix of P, selecting the rows and columns that correspond to the red nodes (there are 7 of them).
> P_rs is a 7x3 matrix, similar to P_rr except that its columns correspond to the green nodes (3 of them).
>
> In Figure 24, we provide a step-by-step breakdown of Gaussian elimination from Equation (4.2) to (5), including the derived value of $u_a$ same as suggested.
> # Re: Questions
> Q1: Yes, it is exactly right that because the kernel size is 3x3, a 3x3 window centered at the a brown dot cannot incorporate information from the light and dark blue dots. We add a demonstration in Figure 21.
>
> Q2: The values of $A_{SS}^P$ and $A_{SS}^Q$ can be arbitrary, including $1/2 A_{SS}$, as long as their sum is $A_{SS}$. In fact the values of $A_{SS}^P$ and $A_{SS}^Q$ are never used standalone, only their sum $A_{SS}^P + A_{SS}^Q$ is used (they become parts of the matrices P,Q which are summed later).
>
> The terms of $A_{SS}^P$ and $A_{SS}^Q$ represent the partition of the value $A_{SS} = A_{SS}^P + A_{SS}^Q$, i.e. into contributions from the patch P and Q separately. This partition is because then the computation for patch P and Q can be done in parallel. Especially for applications like 1st-order FEM discretized PDEs, each patch P and Q will contribution to the $A_{SS}$, and their contributions $A_{SS}^P$ and $A_{SS}^Q$ only depend on information within the patch P and Q, respectively, This makes the computation within P and Q exactly independent from each other so it can be done in parallel.
>
> Q3: The overall runtime of L-BFGS is 20x slower than our method. Note that L-BFGS is a very different type of method---Newton's method using sparse Hessian solver with SciPy backend remains our primary point of comparison, which is 400x slower than our method.
>
> Q4: Only the baseline method [Shi & Malik 2000] uses MATLAB---the “eigs” function and it takes theirs/MATLAB 25s to solve the top-6 Eigs.  In contrast, it takes less than 0.3s for our method (since the major computation is sequentially calling 20 times of A^{-1} b, that each takes 0.011s using our method).
>
> Q5: For fair comparison, both ours and all baseline direct solvers (Scipy, CUDA) do not use symmetry. Theoretically, methods that use symmetry can roughly reduce half of the computation. Note the baselines AMG and (effectively) neural operators do exploit symmetry: In other words, the actual improvement of our method is more significant than what is presented in the paper. In theory, our method could use symmetry to get another 2x speedup. Some relevant discussions are in Appendix A.6, C.4.
>
> Q6: Our basic routine A^{-1} b is a forward problem: solves for x given both A,b. When A is not known, we need to search for the value of A, using the gradient of E w.r.t. entries of A, which further depends on the Jacobian of x over entries of A. As explained in B.5, our solver is already differentiable in PyTorch so no extra work needed to make it differentiable to obtain the Jacobian. Otherwise, one has to explicitly implementing the adjoint method that solves for A^{-T}, as done in [Wang et al. 2023] and also here:
> https://arxiv.org/pdf/2404.17039v2
>
> Q7: We provide sample code in the link: https://anonymous.4open.science/r/icml-rebuttal-268F  and promise the final code will be released. In the code repo, the major effort to implement our solver is “index-tracking”: i.e., in Figure 3, when some nodes are eliminated, what will be the new indices of these nodes (so that subsequent matrix slicings can find the correct submatrices). These are tedious details that can be found in the code. Actually, the 2-lines code in line 179 is a minimum validation to trace down why our solver can be much faster.

---

> > ### Comment · Reviewer_QcrL · 2025-04-04
> >
> > Thank you very much!
> >
> > I am not sure whether it is allowed to attach figures in anonymous links during rebuttal. I am generally fine since I figured out the derivation while reading the paper. However, I strongly recommend you to put in details since I see other reviewers complain about the scarcity of the mathematical derivations as well. To me, I don't understand why you left lots of the details during the submission. This lack of mathematical details doesn't match the high-quality in other parts of the paper. Were you trying to hide the details so that you could later fill them out and submit the full version to a journal??? That's my only hypothesis why this could happen. If so, please don't do that. Even if this paper gets accepted, I think it is necessary to include all details so that readers can understand what is going on and reproduce the results if they need.
> >
> > I am able to run the code provided in the anonymous repository. Is it possible for you to update the repo so that the Jupyter notebook also has a regular linear solver to solve the large system without using your trick? I want to compare the running time and confirm that the speedup is real. Right now, I only have access to the 3D representation of the image, not the 2D matrix representation of the image.
> >
> > I have an additional question. What happens if the kernel size is not 3x3? What if the kernel size is 5x5 or 7x7? Can the current method be easily extended to these large kernel sizes? How would you handle all the boundary pixels? Do we have to rewrite  a different parallel Gaussian elimination method? The pixel overlap would be different in the cases of 3x3 and 5x5, in my undertstanding.

---

> > > ### Author Response · Authors · 2025-04-07
> > >
> > > We appreciate the insightful feedback and careful examination of our method!
> > >
> > > > “…attach figures…”
> > >
> > > Yes, the official ICML guidance allows “figures, proofs, and code” in the anonymous link in the response. https://icml.cc/Conferences/2025/PeerReviewFAQ
> > >
> > > > “...put in details…”
> > >
> > > We thank the reviewer for affirming the high quality of the paper. We appreciate the desire to include additional mathematical details and we will do so in the revised version of the paper. We kept the method section brief due to space limitations as we prioritized the paper to focus on the broad impact of an efficient sparse solver in machine learning.
> > >
> > > During the rebuttal period, we made new figures and visualizations at the request of Reviewer D4Wa. We remark that readers do not unnecessarily need to follow them to reproduce our method. They provide one possible example to implement the block elimination outlined in Figure 3, under a specific node indexing---which can be arbitrary so there are many different ways to implement the elimination. We had omitted indexing-specific descriptions to keep the method general but added it to be specific. With the extra exposition, the work should appeal to a larger audience in machine learning, requiring minimal background knowledge.
> > >
> > > > “...larger kernel…”
> > >
> > > Great question (we will incorporate the discussion in our paper)! Our approaches can be generalized to a larger kernel size of 5x5 or 7x7 but the implementation will be more complicated. Currently one layer of boundary pixels can separate two subdomains P, Q; and it will require two layers of boundary pixels in the case of 5x5 to separate two subdomains P,Q (or 3 layers of boundary pixels in the case of 7x7). Similarly, the boundary pixels for the whole image at the last Schur step will have two layers of pixels. The parallel elimination procedure will be similar but have to account for the fact that the “wire-frame” has two layers of pixels.
> > >
> > > In fact, in some sense, our current method *already supports* kernels larger than 3x3. The actual constraint we have is that every pixel can only contribute to pixels in the same 5x5 patch (and recall that pixels at the patch boundary belong to multiple patches). Thus, the convolution window for a pixel can cover the entire 1/2/4 patches it belongs to (for example, pixels at the patch boundary can use a local window of 9x9 or 9x5; and it is 5x5 for an interior pixel though the window may not be centered at it). Also recall the 5x5 patch size is a hyperparameter that we are free to change arbitrarily in the current method.
> > >
> > > >  code for the baseline
> > >
> > > Thank you for taking the time to run the code of our method and verify that an expected result is produced!
> > > We have also added the code for the baseline method in the same anonymous link  https://anonymous.4open.science/r/icml-rebuttal-268F, in addition to the code for our method.
> > >
> > > For timing, we also realize we forgot to mention that in the demo.ipynb one will need to add a line like ```torch.set_default_device('cuda:0')```  before executing the code, otherwise the notebook will fall back to using the cpu. We have added this to the code repo as well.
> > >
> > > Please note this message will be our last opportunity to reply to your questions. We are absolutely certain about the reproducibility and the level of speedup of our method as reported in the paper. Our code will be released and we envision *a very broad range of users* will benefit from our method and use it to verify its effectiveness.

---

### Official Review · Reviewer_D4Wa · 2025-03-13

**Overall Recommendation:** 2

**Summary:**

The authors propose an efficient method to solve sparse linear systems. Current algorithms for solving such systems are slow, which hinders their applications in real-time scenarios such as interactive graphics. The authors propose a direct solver, which uses a divide-and-conquer strategy to efficiently solve sparse linear systems. The proposed method is differentiable, thus can be integrated into modern machine learning framework.

**Claims And Evidence:**

The authors claim their method is faster than current baselines, Table 1 shows that indeed the runtimes are orders of magnitude faster.

**Essential References Not Discussed:**

None

**Experimental Designs Or Analyses:**

The experimental analysis looks good.

**Methods And Evaluation Criteria:**

The major problem right now with the manuscript is the description of the method. It is currently hard to follow, sections 4.2.1 and 4.2.2. Although the authors tried to put visuals to make the texts easier to follow, certain symbols are introduced without enough details. For example, how do we arrive at equation 4, what are the submatrices A_rr, A_rs representing exactly. Same question for equation 5, what does P_rr, P_rs represent. It may be better if the authors start with even smaller P,Q in Figure 4, say 3x7 and show exactly how the submatrices in equation 4 are composed of. If there are too many details, some can be put into a supplementary. Some of the results can also be moved to supplementary, it is essential that the method is clear enough to understand so that it can be reproduced by interested researchers.

**Other Comments Or Suggestions:**

None

**Other Strengths And Weaknesses:**

None

**Questions For Authors:**

None

**Relation To Broader Scientific Literature:**

Solving sparse linear systems is a general problem, with wide applications to scientific domains, some of which are mentioned in the manuscript. The authors innovation on efficiently solving such sparse systems can greatly accelerate scientific computation in many domains.

**Theoretical Claims:**

None

---

> ### Author Rebuttal · Authors · 2025-03-30
>
> We appreciate the thoughtful feedback on improving our paper. Please also refer to Re:eGtJ for extra descriptions of our methods.
> # Method motivations and descriptions
> We are committed to improving the exposition and we are confident we can make the paper significantly more accessible to a broader ICML audience.
>
> We have added supporting figures in the following link:
> https://anonymous.4open.science/r/icml-rebuttal-268F/rebuttal-figures.pdf
> This includes a step-by-step illustration of our method.
>
> We realize our presentation assumed too much background in Gaussian elimination which we are happy to expand upon. To address this, we have added a brief summary and will expand it into a detailed explanation in the revised paper.
> ## Parallel Gaussian elimination
> Schur complement reduces solving the 2x2 block matrix,
> [ X  Y; Z  W ]
> to solving a smaller matrix [X - YW^{-1} Z]
>
> While equations in the paper might look a bit dense, at the high level the overall objective is quite simple and intuitive: recursively applying the Schur complement many times to reduce the problem to a smaller system.
> Two extra considerations have to be added: 1) always first re-order rows and columns of matrices, so that we know the part we want to eliminate is located at W (or anywhere we prescribed); 2) apply many Schur complements in parallel.
> ## Graph algorithm perspectives
> Our method, as Gaussian elimination, can be simply understood as a *graph algorithm* that removes nodes from a graph. The actual procedure looks involved because the data is put in a tensor to leverage BLAS for parallel computing, and we avoid explicitly constructing the graph.
>
> As in Figure 21, initially each pixel in the image is a node in the graph, and two nodes are connected by an edge with weights $A_{ij}$, if the pixels i and j are adjacent (as defined in Sec. 2). In other words, the matrix A plays the same role as the adjacency matrix in graph theory.
> Gaussian elimination removes nodes one by one from the graph: when removing a node $k$ from the current graph, the only modification we need to make to matrix A is: for all pair of  nodes i,j that are both adjacent to k, update $A_{ij}$ by subtracting the term $A_{ik}A_{kk}^{-1}A_{kj}$ from it; 2) delete the row and column correspond to node $k$.
> I.e., the indirect influence of node j on i via k, is attributed through a direct influence of node j on i, when deleting $k$.
> ## Block elimination
> The updating formula generalizes to the case when $i,j,k$ each is not a single node, but each consists of a set of nodes. For example, for  r=[1,2,3], s=[6,7], $A_{rs}$ refers to the 3x2 submatrix that selects the first three rows and 6-th, 7-th columns from matrix A.
>
> Then, we have block Gaussian elimination: first divide the domain into three sets of nodes as r, s, and t, such that any node in r is not connected to any node in t as separated by s.
> The 3x3 block:
> [Arr, Ars, Art==0;
>  Asr, Ass, Ast;
>  Atr==0,Ats, Att;]
> becomes the 2x2 block:
> [Ass - Asr Arr^{-1} Ars, Ast;
>  Ats, Att;]
> Namely, the update rule is to subtract the adjustment term: Asr Arr^{-1} Ars.
> In Sec 4, we simply do two block Gaussian elimination at the same time.
>
> Then our algorithm is basically a parallel block Gaussian elimination in which many groups of nodes (marked in yellow in Figure 3) are removed by concurrently subtracting many adjustment terms.
>
> The major effort in the code is ``index-tracking’’: carefully track what the indices of remaining nodes become after some nodes are removed.
>
> Illustrated in Figure 24 as a step-by-step breakdown diagram of algebraic manipulation: we apply a few steps of transformations: equations in Gray box are the transformed equations, and definitions in White box (including P, Q and their submatrices) are intermediate symbols we introduce to simplify the notation. The steps simply apply Schur complements many times and in parallel.
> # Re: Questions
> We add visualization, sample code, and derivation diagrams to help to understand the method in the anonymous link above.
> > how do we arrive at equation 4
>
> Please refer to Figure 21-23, for a visualization of the matrix partitioning process used to derive Equation (4), demonstrated on a 3×7 image.
>
> > what are the submatrices A_rr, A_rs, P_rr, P_rs represent
>
> Please refer to Re:QcrL-Clarifications for definitions of submatrices.
>
> # Plug-and-play for end users
> Our method is designed to be *black-box usable* for the vast majority of users—understanding the internal details is often not required (though we will clearly document them). Just as convolution and matrix multiplication are accessed via CUDA (cuDNN/cuBLAS), we envision our solver being similarly exposed through low-level APIs (e.g., via a cuDSS-like interface), allowing users to easily integrate it without needing to implement anything themselves. Our code will be released publicly. A demo and code that include all implementation details are already available at the anonymous link.

---

### Official Review · Reviewer_eGtJ · 2025-03-14

**Overall Recommendation:** 3

**Summary:**

The paper proposes a method for accelerating sparse linear and PDE solvers by transforming sparse Laplacian matrices into dense tensors. This procedure uses dense GPU BLAS kernel to batch and run such system in parallel. This method is differentiable, which can be potentially useful for machine learning pipeline integration.

**Claims And Evidence:**

The major claims of this paper include the following: First, the significant speedup compared to the existing solutions. This is supported by Table 1, where it shows the average runtime to solve Laplacian systems under the proposed method, CUDA, Scipy, and AMG. Second, the paper claims that the proposed method is applicable to more problems than PDE, which is demonstrated in examples such as image segmentation.

**Essential References Not Discussed:**

I have no comments on the essential references.

**Experimental Designs Or Analyses:**

The experimental designs in this paper cover multiple domains, including PDE solving and real-world applications in graphics and vision. In Section 5.2, “A zero-shot baseline: learn-to-solve PDEs,” the author evaluates the proposed method against a state-of-the-art neural operator on the Darcy Flow dataset. While this comparison demonstrates the method’s applicability to PDE solving, it primarily focuses on a single PDE equation. While the paper shows that the method is widely applicable, it is suggested to include experiments on additional PDE types, such as Navier-Stokes equations.

**Methods And Evaluation Criteria:**

The evaluation contains multiple PDE problems, including anisotropic and isotropic Laplacian systems, along with Darcy flow experiments compared to neural operator baseline. Another evaluation aspect is the comparison of average runtime, performed on SciPy, CUDA, and AMG. The author uses the relative error tolerance as the main evaluation metric, which is a standard measurement solution accuracy.

**Other Comments Or Suggestions:**

I do not have specific comments on this aspect as it is outside my primary research area.

**Other Strengths And Weaknesses:**

Strength:
The proposed method significantly speeds up the PDE solver and the method can be applied to extensive areas more than PDE solving.

**Questions For Authors:**

I have no further questions to the authors.

**Relation To Broader Scientific Literature:**

The paper builds on literature in sparse linear solvers, domain decomposition methods, such as the Dirichlet-to-Neumann methods operator, and the backends for direct sparse solvers.

**Theoretical Claims:**

I have no comments on the theoretical claims due to a lack of expertise in this specific area.

---

> ### Author Rebuttal · Authors · 2025-03-30
>
> # General responses
> We thank the reviewers for their careful examination of our paper. We note that all reviewers appreciate the broad impact that our method will have (eGtJ: “can be applied to extensive areas”, D4Wa: “greatly accelerate scientific computation in many domains”, QcrL: “both novel and clever”, “significant potential for applications in various domains”).
>
> We also acknowledge that our paper is rather unusual, in that it revisits an area that is often taken for granted (linear solvers), and in its current form assumes textbook knowledge about classical methods, in particular Gaussian elimination. It’s now clear that by bypassing some steps in our derivations (for reasons of readability and space), we have made the method more difficult to follow for some readers. However, our derivatives are correct (as confirmed by QcrL and numerical experiments), and we are happy to add intermediate derivations in Appendix. These extra steps, along with a more extensive graphical presentation of the problem and its specific sparsity, are available in https://anonymous.4open.science/r/icml-rebuttal-268F and will be added to Appendix.
>
> The reviewers' comments highlight a specificity of our paper, in that it identifies a problem that is *critical yet long overlooked and considered having little-room-to-improve*. It is generally accepted that existing direct solvers are close-to-optimal, and haven’t been improved in decades. However, we do show that the approach we propose accelerates the inversion of large systems with Laplace-like sparsity by several orders of magnitude. There is nothing magical though: we simply make the observation–which has already been made for *forward* operators and is critical to the success of deep networks–that extremely parallelized problems can be implemented extremely efficiently on GPUs, such that they outperform “optimal” (flop-wise) sequential algorithms. Our *main contribution* is in transforming a problem that is not parallel by construction into a form that enables the use of dense parallel operations. Rather than focusing on the mathematics (which build on simple ideas) of our method, we have chosen to focus on its potential applications and on quantifying its performance gains on an array of real-world applications.
> # Broad impact
> Due to the foundational role of linear solvers, our method can broadly impact:
> * We provide the *first method to invert convolution* efficiently and exactly (in the generalized case of spatially-varying kernels) at interactively rate, for vision & image processing.
> * Immediate improvements across many areas: PDE solvers, learning-for-PDEs, image segmentation, physical simulation, shape optimization, and so on.
> * Since solvers are used in all subareas of science and engineering, many computing, solver-in-the-loop and learning systems can benefit from ours by substituting their existing solvers.
> * Our finding---sparse solvers can be 1000x faster than current algorithms---qualifies them as a module in the neural networks; especially those involving geometry/physics, since conventional methods in e.g. physics/geometry processing heavily rely on sparse solvers.
> * Sparse solvers have been a long requested feature in e.g. PyTorch; the lack of efficient implementation hinders its adoption in neural architectures, especially for scientific ML.
> * Optimization: by making Newton’s method much faster on images.
> * Spectral methods are accelerated: vision, graphics, geometric deep learning.
> * Partially explain and reduce the performance gap between conventional methods and deep learning for many tasks.
> # Other PDEs
> We appreciate the insightful suggestions on more PDEs like the Navier–Stokes. We note that *Laplace FEM solvers are effectively sparse linear solvers*, making them ideal for evaluating our method. Many nonlinear PDEs—minimal surfaces, deformation, heat flow, optimal control of PDEs (diffeomorphisms), and eigenvalue problems—*repeatedly call Laplace solvers as subroutines*. Shown in Appendix, our method *directly accelerates these different types of PDEs*.
>
> Some classic Navier–Stokes solvers involve solving Laplace equations with *constant* kernels—these can often be handled efficiently by FFT-based methods, and do not require general-purpose sparse solvers like ours. In contrast, our method targets **spatially-varying kernels** (e.g., inhomogeneous materials so FFTs no longer apply) or **generalized deconvolution** with spatially-varying kernels. While our method *can* handle more general fluid models with varying diffusivity or anisotropy, we are not aware of widely-used PDE learning benchmarks involving such settings.
>
> We discuss in Appendix (line 962) on how future work can employ **a simplified variant** of our method to accelerate constant-kernel PDEs like regular Navier–Stokes, and we will try to implement the discussed variants and apply it to Navier–Stokes experiments if time permits. This will further extend the applicability of our method.

---

### Decision · Program_Chairs · 2025-05-01

**Decision:**

Accept (poster)

**Comment:**

The authors provide a practical method for solving sparse linear system, following a divide and conquer approach that allows parallelization and efficient use of GPU.
Reviewers agreed that the method seems to offer, in multiple settings, interesting speedups. Nevertheless reviewer `QcrL`, with additional experiments, suggested that the 1000x claim was too bold, and this needs to be toned down for the camera ready version.
As acknowledged during the rebuttal, the authors must also include in appendix additional derivations to ease understanding of the paper for the ICML audience.